# The least-control principle
# for local learning at equilibrium

**Alexander Meulemans**[*†1]**, Nicolas Zucchet**[*1]**, Seijin Kobayashi**[*†1]
**Johannes von Oswald**[1]**, João Sacramento**[2]

[1]Department of Computer Science, ETH Zürich
[2]Institute of Neuroinformatics, University of Zürich and ETH Zürich
`{ameulema, nzucchet, seijink, voswaldj, rjoao}@ethz.ch`

## Abstract

Equilibrium systems are a powerful way to express neural computations. As special cases, they include models of great current interest in both neuroscience and machine learning, such as deep neural networks, equilibrium recurrent neural networks, deep equilibrium models, or meta-learning. Here, we present a new principle for learning such systems with a temporally- and spatially-local rule. Our principle casts learning as a *least-control* problem, where we first introduce an optimal controller to lead the system towards a solution state, and then define learning as reducing the amount of control needed to reach such a state. We show that incorporating learning signals within a dynamics as an optimal control enables transmitting activity-dependent credit assignment information, avoids storing intermediate states in memory, and does not rely on infinitesimal learning signals. In practice, our principle leads to strong performance matching that of leading gradient-based learning methods when applied to an array of problems involving recurrent neural networks and meta-learning. Our results shed light on how the brain might learn and offer new ways of approaching a broad class of machine learning problems.

## 1  Introduction

The neural networks of the cortex are both layered and highly recurrent [1, 2]. Their high degree of recurrence and relatively low depth stands in contrast to the prevailing design of artificial neural networks, which have high depth and little to no recurrence. This discrepancy has triggered a recent wave of research into recurrent networks that are more brain-like and which achieve high performance in perceptual tasks [3–7]. Concurrently, another line of recent work has shown that repeating a short sequence of neural computations until convergence can lead to large gains in efficiency, reaching the state-of-the-art in various machine learning problems while reducing model size [8–10].

As we develop models that come closer to cortical networks by way of their recurrence, the precise mechanisms supporting learning in the brain remain largely unknown. While gradient-based methods currently dominate machine learning, standard methods for efficient gradient computation result in non-local rules that are hard to interpret in biological terms [11, 12]. This issue is particularly aggravated when applying these methods to complex systems involving recurrence [13]. Indeed, while multiple interesting proposals [14–23] have emerged for how to efficiently compute loss function gradients for feedforward networks in biologically-plausible ways, apart from a few notable exceptions [24, 25] much less progress has been made for recurrent networks. Furthermore, the majority of these methods requires that error feedback does not influence network activity, a property

---

*Equal contribution
†Work partly done at the Institute of Neuroinformatics, University of Zürich & ETH Zürich

36th Conference on Neural Information Processing Systems (NeurIPS 2022).

at odds with many experimental findings on activity-dependent plasticity [26]. In this paper, we focus on the problem of learning such recurrent systems using biologically-plausible, activity-dependent local rules. To make progress in this longstanding question we make the assumption that our system is at equilibrium, and formalize learning as the following optimization problem:

$$\min_\theta L(\phi^*) \quad \text{s.t.} \ f(\phi^*, \theta) = 0, \tag{1}$$

where $\theta$ are the parameters we wish to learn, $L$ is a loss function which measures performance, and $\dot{\phi} = f(\phi, \theta)$ is the system dynamics which is at an equilibrium point $\phi^*$. Our model is thus *implicitly* defined through a dynamics that is at equilibrium. Expressing our problem in this general form allows us to model a very broad class of learning systems, without being restricted to a particular type of neural network [27–29]. In particular, we cover both feedforward and recurrent neural architectures. More generally, $f$ is not even restricted to being a neural dynamics. Consider the case where $f$ defines a learning algorithm governing the dynamics of the *weights* of a network; in this case (1) becomes a meta-learning problem, where the goal is to tune the (meta-)parameters of a learning algorithm such that its performance improves.

Inspired by a recent control-based learning method for feedforward neural networks [30], we present a new principle for solving problems of the form (1) which yields learning rules that are (i) gradient-following, (ii) local in space, (iii) local in time, and in particular do not require storing intermediate states, (iv) activity-dependent by embedding error information into the network activity, and (v) not reliant on infinitesimally small learning signals. To meet all five criteria at once, we depart from direct gradient-based optimization of the loss, and we reformulate gradient-based learning within the framework of optimal control as a problem of *least-control*.

Such problems can be approached in two steps. First, an optimal controller provides additional feedback input leading the dynamical system to a least-control state: an equilibrium point in which the loss $L$ is minimized with the least amount of control. Subsequently, the parameters are changed to further reduce the amount of control at the resulting equilibria. Critically, this minimization can be achieved using a local gradient-based rule for which all the required information is available at the controlled equilibria. This should be contrasted to the ubiquitous backpropagation of error algorithm [31, 32] (and its variant for equilibrium systems [33, 34]) which yields non-local parameter updates based on the outcome of two separate phases. Importantly, our least-control problem is intimately related to the original learning problem; we make the connection between the two precise by identifying mathematical conditions under which least-control solutions correspond to minima of the original objective $L(\phi^*)$, or a bound of it.

We apply our principle to learn equilibrium neural networks featuring a variety of architectures, including both fully-connected and convolutional feedfoward networks, laterally-connected recurrent networks of leaky integrator neurons, as well as deep equilibrium models [8–10], a recent family of high-performance models which repeat until convergence a sequence of complex computations. To further demonstrate the generality of our principle, we then consider a recently studied meta-learning problem where the goal is to change the internal state of a complex synapse such that future learning performance is improved [35]. We find that our single-phase local learning rules yield highly competitive performance in both application domains. Our results extend the limits of what can be achieved with local learning rules, opening novel avenues to an old problem.

## 2 The least-control principle

### 2.1 The principle

We introduce the least-control principle for learning dynamical systems that evolve in time according to $\dot{\phi}(t) = f(\phi(t), \theta)$ until an equilibrium $\phi^*$ is reached. Instead of directly minimizing a loss evaluated at equilibrium $L(\phi^*)$ as in (1), we augment the dynamics $\dot{\phi}(t) = f(\phi(t), \theta) + \psi(t)$ with a control signal $\psi(t)$ that drives the system towards a new controlled equilibrium where the loss is minimized (c.f. Figure 1). Our principle recasts learning as minimizing the amount of control needed to reach that state:

$$\min_{\theta, \phi, \psi} \frac{1}{2} \|\psi\|^2 \quad \text{s.t.} \ f(\phi, \theta) + \psi = 0, \ \nabla_\phi L(\phi) = 0. \tag{2}$$

The two constraints in the equation above ensure that we reach a steady state of the controlled dynamics, and that the loss $L$ is minimized at this state. Without loss of generality, we consider a single data point above (c.f. Section S5 in the supplementary materials).

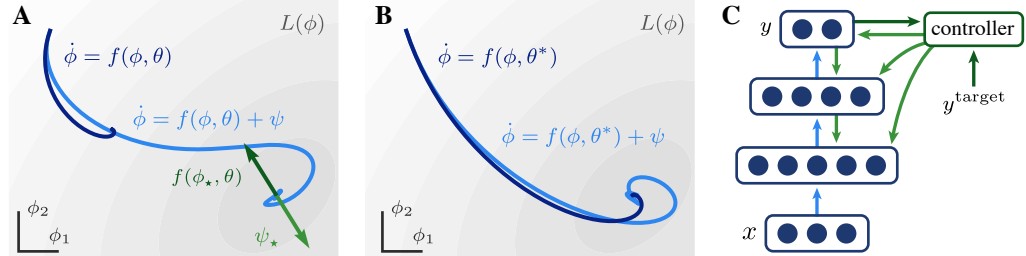

Figure 1: (A, B) Intuition behind the least-control principle. (A) During learning, the free dynamics (dark blue) is augmented with a controller (light blue) that drives the system towards an equilibrium that minimizes the loss function $L$ (grey, the darker the smaller the loss is). Our principle prescribes minimizing the amount of optimal control $\psi_\star$ (light green) needed at an equilibrium that minimizes $L$. (B) After learning, the free equilibrium coincides with the controlled equilibrium and hence minimizes the loss $L$. (C) Example of an instantiation of the principle for the supervised learning of a feedforward neural network. The controller both drives the output units $y$ of the network towards a target value $y^{\text{target}}$ and influences the rest of the network. The control signal $\psi$ can be computed directly from the output error or indirectly by leveraging internal feedback within the network.

To solve (2), we first fix the parameters $\theta$. We then find an optimal control $\psi_\star$ and controlled equilibrium $\phi_\star$ that minimize the following least-control objective, for the current setting of $\theta$,

$$\min_{\phi,\psi} \frac{1}{2}\|\psi\|^2 \quad \text{s.t. } f(\phi,\theta) + \psi = 0, \ \nabla_\phi L(\phi) = 0. \tag{3}$$

We call $\phi_\star$ the least-control state and $\psi_\star$ the least control. Here, $\psi_\star$ represents an optimal control with terminal cost $\|\psi\|^2$, and a terminal constraint enforcing that the optimally controlled equilibrium $\phi_\star$ minimizes the loss $L$. Critically, both credit assignment and system dynamics are now combined into a single phase through the optimal control $\psi_\star$. We then take a gradient-following update in $\theta$ on $\mathcal{H}(\theta) := \frac{1}{2}\|\psi_\star\|^2$, the least-control objective of Eq. 3, which implicitly depends on $\theta$ through $\psi_\star$.

Theorem 1 shows that the optimal control $\psi_\star$ contains enough information about the learning objective so that the gradient is easy to compute. We prove Theorem 1, along with all the theoretical results from Section 2, in Section S2. We use $\partial_x$ to denote partial derivatives w.r.t. the variable $x$.

**Theorem 1** (Informal). *Under some mild regularity conditions, the least-control principle yields the following gradient for $\theta$:*

$$\nabla_\theta \mathcal{H}(\theta) = -\partial_\theta f(\phi_\star,\theta)^\top \psi_\star = \partial_\theta f(\phi_\star,\theta)^\top f(\phi_\star,\theta). \tag{4}$$

To illustrate what our principle achieves, let us briefly consider the setting where $f(\phi,\theta)$ implements a feedforward neural network, typically learned by backpropagation of a supervised error signal. In this case, the variable $\phi$ corresponds to the state of the neurons (e.g. their firing rate or postsynaptic voltage) and $\theta$ to the synaptic connection weights. The least control $\psi_\star$ then drives the network to an equilibrium that minimizes the output loss, while being of minimum norm. The resulting weight update (c.f. Section S6.2) now represents a local Hebbian rule multiplying presynaptic input with the postsynaptic control signal: the neural activity implicitly encodes credit assignment information, in such a way that local learning becomes possible. By contrast, in the backpropagation algorithm, errors do not directly influence neural activity and are computed in a separate phase.

The simplicity of our parameter update becomes apparent when contrasting it with the gradient associated with the original objective (1), as the implicit function theorem (c.f. Section S3) gives

$$\nabla_\theta L(\phi^*) = -\partial_\theta f(\phi^*,\theta)^\top \left(\partial_\phi f(\phi^*,\theta)\right)^{-\top} \nabla_\phi L(\phi^*). \tag{5}$$

Calculating this gradient directly requires inverting the Jacobian $\partial_\phi f(\phi^*,\theta)$, which is intractable for large-scale systems. The standard way of dealing with this issue is to resort to the two-phase recurrent backpropagation algorithm, which estimates $\nabla_\theta L(\phi^*)$ by running a second linear dynamics until equilibrium, while holding the first equilibrium state $\phi^*$ in memory. For acyclic computation graphs, of which feedforward neural networks are a prime example, it simplifies into the error backpropagation algorithm. Those procedures are still more complicated than ours: they require implementing a specialized auxiliary dynamical system, holding intermediate states in memory, and alternating between inference and gradient computation phases.

## 2.2 A general class of dynamics leads to the same optimal control and parameter update

Our principle is agnostic to the dynamics $\dot{\psi}(t)$ of the controller; the only assumption made is that its value $\psi_*$ at the steady state is optimal, i.e. $\psi_* = \psi_\star$. We use the subscripts $_*$ to denote equilibrium states of the controlled dynamics and $_\star$ to indicate optimality on the least-control objective of Eq. 3, and the superscript $^*$ for a free (not controlled) equilibrium. In particular, the parameter update only depends on information available at the controlled equilibrium state $(\phi_*, \psi_*)$. Here, we capitalize on this important flexibility by designing different simple feedback controller dynamics that lead to the same optimal steady state. We consider feedback controllers which take the following general form:

$$\dot{\phi} = f(\phi, \theta) + Q(\phi, \theta)u, \quad \text{and} \quad \dot{u} = -\nabla_\phi L(\phi) - \alpha u, \tag{6}$$

where $u$ is a controller defined on the output units and $Q(\phi, \theta)$ determines the influence of the controller on the entire system. We note that here and throughout our paper we omit the dependence on time (above, of $\phi(t)$ and $u(t)$) from our equations to unclutter the presentation. In (6), the mapping $Q(\phi, \theta)$ can either be a direct mapping from the output controller $u$ to the entire system, or an indirect one leveraging internal feedback mechanisms in the system (c.f. Figure 1C). The controller $u$ may be seen as a leaky integral controller with leak rate $\alpha$, that uses the output error $\nabla_\phi L$ to drive the network towards a minimal-loss state (c.f. Section S2). Returning to our example of a neural network, $u$ leads the network into a state configuration where the output neurons $y$ are at the target value $y^{\text{target}}$, when $\alpha = 0$.

We use leaky integral control in (6) for explanatory purposes and as we use it in our experiments. We remark however that our theory goes beyond this form of output control; in particular, it generalizes to any controller satisfying $\alpha u_* + \nabla_\phi L(\phi_*) = 0$ at equilibrium. For example, the proportional control $u = -\beta \nabla_\phi L(\phi)$ satisfies this condition for $\alpha = \beta^{-1}$.

Theorem 2 provides general conditions for the dynamics (6) to converge to an optimal control signal needed for obtaining first-order parameter updates in Theorem 1.

**Theorem 2.** *Let $(\phi_*, u_*)$ be a steady state of the generalized dynamics* (6). *Assume that $\partial_\phi f(\phi_*, \theta)$ is invertible and that the following column space condition is satisfied at equilibrium:*

$$\text{col}\left[Q(\phi_*, \theta)\right] = \text{row}\left[[0, \text{Id}] \, \partial_\phi f(\phi_*, \theta)^{-1}\right] \tag{7}$$

*where the matrix $[0, \text{Id}]$ has $|\phi|$ columns and as many rows as the system has output units. Then, $\psi_* = Q(\phi_*, \theta)u_*$ is an optimal control $\psi_\star$ for the least-control objective* (3), *in the limit $\alpha \to 0$.*

Interestingly, Theorem 2 shows that many different feedback mappings from the output controller $u$ towards the system state $\phi$ are possible, as long as the mapping $Q(\phi_*, \theta)$ satisfies the column space condition (7) at equilibrium, in the limit $\alpha \to 0$. Furthermore, the control $\psi$ need not be instantaneous and can have its own temporal dynamics $\dot{\psi}$ alongside the output controller dynamics $\dot{u}$, as the column space condition (7) is only defined at equilibrium. We exploit this property and construct a general-purpose algorithm that computes an optimal control $\psi_\star$:

$$\dot{\phi} = f(\phi, \theta) + \psi, \quad \text{and} \quad \dot{\psi} = \partial_\phi f(\phi, \theta)^\top \psi + u \tag{8}$$

with $u$ the leaky integral controller defined in (6). This *inversion dynamics* has two special properties. First, it satisfies the conditions of Theorem 2 by construction (c.f. Section S2.3). Second, as defined above, $\dot{\psi}$ resembles the second-phase dynamics of the (recurrent) backpropagation algorithm (c.f. Section S3), or in case of feedforward networks, the layerwise backpropagation of error signals in the error backpropagation algorithm. Intuitively, in this case we find an optimally-controlled equilibrium by *simultaneously* (in a single phase) running the first- and second-phase dynamics of (recurrent) backpropagation, with the critical difference that here the dynamical equations of both phases interact through the relation $\dot{\phi} = f(\phi, \theta) + \psi$.

## 2.3 The least-control principle solves the original learning problem

Iterating our parameter update minimizes the least-control objective $\mathcal{H}(\theta)$ and not directly the original objective of learning (1). The latter is however the objective of ultimate interest, used to measure the performance of the system after learning, when it is no longer under the influence of a controller. We now show that there is a close link between the two objectives and identify a wide range of conditions under which least-control solutions also solve the original learning problem (1).

Proposition 3 shows the intuitive result that if the optimal control is minimized to zero, the equilibria of the controlled and free dynamics coincide and hence the loss $L$ is minimized at the free equilibrium as well (c.f. Figure 1.B).

**Proposition 3.** *Assuming $L$ is convex on the system output, we have that the optimal control $\psi_\star$ is equal to 0 iff. the free equilibrium $\phi^*$ minimizes $L$.*

As $\psi_\star = -f(\phi_\star, \theta)$, reaching $\psi_\star = 0$ can only be done if the model is powerful enough to perfectly fit the controlled equilibria for all data points. We show in Proposition 4, that overparameterization, i.e., having more parameters $\theta$ than system states $\phi$, indeed helps solve the original learning problem (1). The condition $\partial_\theta f(\phi_\star, \theta)$ being of full row rank, that is needed in this proposition, can only be satisfied when the dimensions satisfy $|\theta| \geq |\phi|$, with $\phi$ the concatenated states of all data points, hence providing another perspective on why overparameterization helps.

**Proposition 4.** *Assuming $L$ is convex on the system output, a local minimizer $\theta$ of the least control objective $\mathcal{H}$ is a global minimizer of the original learning problem* (1)*, under the sufficient condition that $\partial_\theta f(\phi_\star, \theta)$ has full row rank.*

Fully minimizing the amount of control to zero is not always possible, which highlights the need to understand the relation between the least-control objective $\mathcal{H}(\theta) = \frac{1}{2}\|\psi_\star\|^2$ and the loss $L(\phi^*)$ from the original learning problem (1). Proposition 5 shows that we can do so, as minimizing $\|\psi_\star\|^2$ indirectly minimizes the loss $L(\phi^*)$, under some regularity and strong convexity assumptions.

**Proposition 5.** *If $\|f\|^2$ is $\mu$-strongly convex, $L$ is $M$-Lipschitz continuous and the minimum of $L$ is 0, then*

$$L(\phi^*) \leq \frac{\sqrt{\mu}}{\sqrt{2}M}\|\psi_\star\| = \frac{\sqrt{\mu}}{\sqrt{2}M}\|f(\phi_\star, \theta)\|.$$

**Approximate equilibria.** Our least-control theory requires the system to be at equilibrium, and that the loss is minimized at the controlled equilibrium. This is almost never the case in practice, for instance because the dynamics are run for a finite amount of time, the column space condition for $Q$ is not perfectly satisfied, or because $\alpha$ is not zero in the output controller $u$ dynamics. The update we take therefore does not strictly follow the gradient of the least-control objective $\mathcal{H}(\theta)$. We formally show in Section S2.6 that our update resulting from an approximate optimally-controlled equilibrium is close to the gradient $\nabla_\theta \mathcal{H}$. More precisely, under some regularity conditions on $f$, the error in the gradient estimation is upper bounded by the distance between $\phi_\star$ and its estimate. Additionally, if we reach equilibrium but $\alpha$ is non-zero the update $-\partial_\theta f(\phi_*, \theta)^\top \psi_*$ follows the gradient of an objective very similar to the least-control one $\mathcal{H}$, under strict conditions on the feedback mapping $Q$.

## 2.4 The least-control principle as constrained energy minimization

We now provide a dual view of our least-control principle from the perspective of energy minimization. This will allow designing a second general-purpose dynamics for computing the least control $\psi_\star$, and establishing a link to another principle for learning known as the free-energy principle. To arrive at an energy function, we rewrite the least-control objective (3) by substituting the constraint $\psi = -f(\phi, \theta)$ into this objective, leading to the following constrained optimization problem:

$$\phi_\star = \arg\min_\phi \frac{1}{2}\|f(\phi, \theta)\|^2 \quad \text{s.t.} \ \nabla_\phi L(\phi) = 0 \tag{9}$$

The role of optimal control can therefore be reinterpreted as finding the state that is the closest to equilibrium under the free dynamics, among the states that minimize the loss function.

Next, we introduce the augmented energy $F(\phi, \theta, \beta) := \frac{1}{2}\|f(\phi, \theta)\|^2 + \beta L(\phi)$, which adds a nudging potential to the equilibrium energy $\|f(\phi, \theta)\|^2$, and take the *perfect control limit*, $\beta \to \infty$, where this potential dominates the energy. As we show in Section S2.5, minimizing the augmented energy with respect to the state variable $\phi$ leads to fulfilling our objective (9). Interestingly, when $\beta \to \infty$ and $f(\phi, \theta)$ implements a feedforward neural network, we recover the free-energy function which governs the predictive coding model of Whittington and Bogacz [36], which was obtained from an entirely different route based on variational expectation maximization for a probabilistic latent variable model [37] (c.f. Section S4).

**Energy-based dynamics as optimal control.** We leverage this connection to design a second general-purpose dynamics that computes an optimal control, by using the gradient flow on $F$:

$$\dot{\phi} = -\partial_\phi f(\phi, \theta)^\top f(\phi, \theta) - \beta \nabla_\phi L(\phi) \tag{10}$$

Here, the control $\psi$ is implicitly contained in the dynamics of $\phi$. Part of the dynamics, $-\beta\nabla_\phi L(\phi)$, plays the role of an infinitely-strong proportional controller on the system output; another part is in charge of optimally sending the teaching signal from the output back to the rest of the system. We refer to Section S2.5 for more details.

**Two opposing limits: perfect control vs. weak nudging.** The update $\partial_\theta f^\top f$ that we obtain in Theorem 1 has been extensively studied in the weak nudging limit $\beta \to 0$ [36, 38, 39], which sits at the opposite end of the $\beta$-spectrum of the perfect control limit studied here. A seminal result shows that as $\beta \to 0$ the gradient $\nabla_\theta L(\phi^*)$ of the original objective function is recovered [24, 40]. However, this update is known to be sensitive to noise as the value of $f$ can be very small at the weakly-nudged equilibrium [35]. Our least-control principle works at the opposite perfect control end of the spectrum ($\beta \to \infty$), and it is therefore more resistant to noise. One of our main contributions is to show that using the same update, but instead evaluated at an optimally-controlled state $\phi_\star$, still performs principled gradient-based optimization, however now on the least-control objective $\mathcal{H}(\theta)$.

# 3 Applications of the least-control principle

The least-control principle applies to a general class of dynamical systems at equilibrium. By contrast, the majority of previous work on local learning focused on designing circuits and rules tailored towards feedforward neural networks [14, 17, 23, 36, 41–45]. We now demonstrate the generality of our principle by applying it to two problems of interest in neuroscience and machine learning: deep and recurrent neural network learning, and meta-learning.[3] In both cases, our principle leads to activity-dependent local learning rules by leveraging Theorem 1, while making use of simple and flexible optimal controllers to feed back credit assignment information (c.f. Theorem 2). We test our learning and meta-learning systems on standard benchmarks and find that they perform competitively when compared to conventional gradient-based methods.

## 3.1 The least-control principle for feedforward and recurrent neural networks

As a first demonstration of learning according to our principle we consider feedforward neural networks with multiple layers of hidden neurons, the current preferred architecture for a large array of perceptual problems. Moreover, prior work on biologically-plausible alternatives to backpropagation has mostly focused on feedforward networks, making them a natural first choice to study our principle.

Motivated by the massive recurrence of cortical networks, we then turn to equilibrium recurrent neural networks as a second application of our principle, aiming to demonstrate its applicability beyond feedforward models. While there is debate over the precise functional roles of recurrent processing in the brain, there is strong experimental evidence that these connections are not limited to transmitting learning signals and play an active role in information processing [e.g., 46–49]. From the viewpoint of machine learning, it has been shown that equilibrium RNNs are often more parameter-efficient than feedforward networks, while reaching state-of-the-art performance [3, 7, 9].

More concretely, we consider a generic neural network driven by a fixed input $x$ assumed to converge to an equilibrium state, obeying the following free dynamics (e.g. Figure 1.C):
$$\dot\phi = f(\phi, \theta) = -\phi + W\sigma(\phi) + Ux \tag{11}$$
with $\phi$ the neural activities, $W$ the internal synaptic weight matrix, $U$ the input weight matrix and $\sigma$ the activation function. Note that this notation is general enough to encompass feedforward and recurrent network architectures; while freely chosen $W$ and $U$ yield a vanilla RNN, structuring $W$ and $U$ appropriately (with block-diagonal structure) allows recovering standard feedforward fully-connected or convolutional deep neural networks. We point to Section S6 for full architectural details. The learning problem then consists of optimizing the network weights $\theta = \{W, U\}$ on a loss $L(\phi^*)$ at the equilibrium activity $\phi^*$. This loss is only measured on the output neurons, a subset of $\phi$.

We now add a control signal $\psi$ to the dynamics (11). Theorem 1 guarantees that if the controlled steady state $\psi_*$ is optimal, the following updates minimize the least-control objective $\mathcal{H}(\theta)$:
$$\Delta W = \psi_*\sigma(\phi_*)^\top, \quad \text{and} \quad \Delta U = \psi_* x^\top. \tag{12}$$
To showcase the flexibility of our principle, we use Theorem 2 to design various simple feedback circuits that compute these control signals, as we detail next. We visualize those circuits in Figure S2.

---

[3]Code for all experiments is available at `https://github.com/seijin-kobayashi/least-control`

### 3.1.1 A simple controller with direct linear feedback

To instantiate the least-control principle in its simplest form, we project the output controller $u$ onto the hidden neurons $\phi$ with direct linear feedback weights $Q$, which is a generalization of a recent feedforward network learning method [30] to equilibrium neural networks. Broadcasting output errors directly [50] may be seen as one of the simplest possible ways of avoiding the weight transport problem [11, 12, 16]. This results in the following dynamics:

$$\dot{\phi} = -\phi + W\sigma(\phi) + Ux + Qu, \quad \text{and} \quad \dot{u} = -\nabla_\phi L(\phi) - \alpha u \tag{13}$$

with $u$ a simple leaky integral output controller (c.f. Section 2.2), and $\nabla_\phi L$ the output error.

Theorem 2 guarantees that $Qu_*$ is an optimal control in the limit of $\alpha \to 0$, if the feedback weights $Q$ satisfy the column space condition $\text{col}[Q] = \text{row}\left[[0, \text{Id}](\text{Id} - W\sigma'(\phi_*))^{-1}\right]$ for all data samples. For a linear feedback mapping, this condition cannot be exactly satisfied for multiple samples as the row space is data-dependent, whereas $Q$ is not. Still, we learn the feedback weights $Q$ to approximately satisfy this column space condition, by using a local Hebbian rule that operates simultaneously to the learning of the other weights, inspired by recent work on feedforward networks [30, 51] (details in Section S6). The resulting update provably finds feedback weights $Q$ that satisfy the column space condition for one sample, and we empirically verify that the control signal it gives in the multiple samples regime is close to optimal (c.f. Section S6).

Despite the simplicity of this linear feedback controller, we show that the training procedure described above is still powerful enough to learn both a two-hidden-layer (each of 256 neurons) fully-connected feedforward network and an equilibrium RNN with a fully-connected recurrent layer of 256 neurons on the MNIST digit classification task [52] (c.f. Table 1, LCP-LF). Notably, it performs almost as well as backpropagation (BP) and recurrent backpropagation (RBP), the current methods of choice for equilibrium RNN training. We use fixed point iterations to find the steady states $\phi_*$ and $u_*$ in Eq. (13) for computational efficiency (c.f. Section S6 for more simulation details). We also observe that the empirical performance improves by changing the parameter updates to $\Delta W = \sigma'(\phi_*)Qu_*\sigma(\phi_*)^\top$ and $\Delta U = \sigma'(\phi_*)Qu_*x^\top$ when the column space condition is not perfectly satisfied, corroborating the findings of Meulemans et al. [30, 51] for feedforward neural networks.

| Method | FF | RNN |
|---|---|---|
| LCP-LF | $97.73^{\pm0.07}$ | $97.70^{\pm0.11}$ |
| LCP-DI | $98.11^{\pm0.07}$ | $97.58^{\pm0.12}$ |
| LCP-DI (KP) | $98.14^{\pm0.09}$ | $97.75^{\pm0.11}$ |
| LCP-EBD | $98.00^{\pm0.03}$ | $97.60^{\pm0.15}$ |
| BP/RBP | $98.29^{\pm0.14}$ | $97.87^{\pm0.19}$ |

Table 1: MNIST test set classification accuracy (%) for a feedforward network (2x256 neurons) and an RNN (256 neurons) trained by resp. backpropagation (BP) and recurrent backpropagation (RBP), and by the least-control principle (LCP) with linear feedback (LF), energy-based dynamics (EBD), dynamic inversion (DI) and with learned feedback weights (DI KP). Mean $\pm$ std computed over 3 seeds.

### 3.1.2 Learning with general-purpose optimal control dynamics

In the previous section we designed and tested a simple error broadcast circuit and showed that we can learn it such that the conditions of Theorem 2 are approximately met. This approximate feedback is likely insufficient for more complex tasks which require harnessing depth or strongly-recurrent dynamics [53]. This leads us to consider two feedback circuits with more detailed architectures, which have the capacity to steer the network to an (exact) optimally-controlled equilibrium.

The first circuit is based on our general-purpose dynamic inversion (8). Applied to RNNs it yields:

$$\dot{\phi} = -\phi + W\sigma(\phi) + Ux + \psi, \quad \text{and} \quad \dot{\psi} = -\psi + \sigma'(\phi)S\psi + u. \tag{14}$$

The second circuit exploits the dual energy minimization view of our principle presented in Section 2.4. Taking the energy-descending dynamics $\dot{\phi} = -\partial_\phi F(\phi, \theta, \beta)^\top$ of Eq. 10, we obtain

$$\dot{\phi} = -\psi + \sigma'(\phi)S\psi + u, \quad \text{and} \quad \psi = \phi - W\sigma(\phi) - Ux, \tag{15}$$

where we use the output controller $u$ of Eq. 6 as a generalization of the proportional control $-\beta\nabla_y L$ to make the two circuits more directly comparable. Above, we introduce decoupled feedback weights $S$ to avoid sharing weights between $\phi$ and $\psi$, and we learn $S$ to align with $W^\top$ by using the Kolen-Pollack local learning rule [42, 54], which simply transposes the weight update $\Delta W$ (12) and adds weight decay to both update rules: $\Delta S = \sigma(\phi_*)\psi_*^\top - \gamma S$.

Both circuits implement optimal control through neural dynamics, avoiding explicit matrix inverses and phases. In biological terms, they lead to different interpretations and implementation possibilities, highlighting the flexibility of our learning principle. The dynamic inversion circuit (14) fits naturally with dendritic error implementations, where $\psi$ can be linked to feedback dendritic compartments whose signals steer plasticity [14, 17, 20, 23, 30, 41, 51, 55]. In particular, by leveraging burst-multiplexing circuits [14, 23] or interneuron microcircuits [17], all information for the dynamics and weight updates can be made locally available (cf. Section S6). On the other hand, the energy-minimizing dynamics (15) naturally leads to an implementation based on prediction and error neuron populations, characteristic of predictive coding circuits [36, 56–58]. We provide a high-level discussion of the two circuit alternatives (14, 15) in Section S6.

First, we test our optimally-controlled neural networks by training a fully-connected equilibrium RNN on MNIST with circuit (14), while varying the recurrent layer width, cf. Figure 2. This analysis shows that our theoretical results of Section 2.3 translate to practice, as for strongly overparameterized networks, both $\mathcal{H}$ and $L$ are minimized to zero, whereas for underparameterized networks, minimizing the least-control objective $\mathcal{H}(\theta)$ leads to a significant decrease in $L$. Moreover, our feedforward (2x256 hidden neurons) and recurrent (256 hidden neurons) networks achieve competitive test performance compared to backpropagation (recurrent backpropagation, for RNNs), both when the feedback weights are trained (LCP‑DI KP) and when they are tied ($S = W^\top$; LCP‑DI and LCP-EBD), irrespective of whether we use the energy-based (15) or dynamic inversion (14) optimal control dynamics, cf. Table 1.

Figure 2: Training loss $L$ and the amount of control $\|\psi_\star\| = \sqrt{2\mathcal{H}}$ after training until convergence on MNIST with various recurrent layer widths.

Next, we test whether the circuits and rules obtained through our principle scale to more complex architectures and tasks and move to the CIFAR-10 image classification benchmark [60]. We study two distinct models: a standard feedforward convolutional neural network, and a deep equilibrium model (DEQ) consisting of a recurrent block of convolutions [9]. The latter can be interpreted as an efficient infinite-depth residual neural network with weight-sharing [61, 62]. Full architectural details are provided in Sections S6.4.3 and S6.5.2; we note that to showcase our least-control principle as a scalable learning method in the DEQ experiments, we make a number of design choices that include using automatic differ-

| Method | C10 (FF) | C10 (RNN) | INR |
|---|---|---|---|
| BP/RBP | $77.58^{\pm 0.14}$ | $80.14^{\pm 0.20}$ | $25.47^{\pm 4.16}$ |
| LCP-DI | $77.28^{\pm 0.10}$ | $80.26^{\pm 0.17}$ | $25.11^{\pm 2.90}$ |
| LCP-KP | $77.16^{\pm 0.10}$ | – | – |

Table 2: C10: CIFAR-10 test accuracy (%) of a convolutional feedforward (FF) and equilibrium RNN (RNN), learned by dynamic inversion (DI; Kollen-Pollack learning without weight sharing: KP) and by standard/recurrent backpropagation (BP/RBP). INR: the peak signal-to-noise ratio (in dB) of an implicit neural representation learned on a natural image dataset [59]. Mean $\pm$ std computed over 3 seeds.

entiation to compute certain derivatives. For our feedforward network experiments we make no such concessions, and we further investigate a weight-transport-free variant using Kollen-Pollack learning. We show on Table 2 that the networks trained by our principle are able to match those trained with backpropagation. Interestingly, we observe that the DEQ models considered here are not overparameterized enough to not need control, but that this does not impair final predictive performance (c.f. Table S5).

Lastly, we turn to implicit neural representations [63], yet another task for which recurrent connections are known to be particularly beneficial [10]. The network there has to represent an image in its weights so that it is able to map pixel coordinates to their color values. Briefly, we once again find that our principle performs competitively to RBP. We refer full details in Section S6 for more details.

## 3.2   The least-control principle for meta-learning

The generality of the least-control principle enables considering learning paradigms other than supervised learning. Here, we show that it can be applied to meta-learning, a framework that

aims at improving a learning algorithm by leveraging shared structure in encountered tasks [64]. This framework is gaining traction in neuroscience, having been featured in recent work aiming at understanding which systems [65–67] and rules [35] could support meta-learning in the brain.

**Meta-learning complex synapses.** Following [35, 68], we model the learning algorithm for training a neural network with synaptic weights $\phi$ on some task $\tau$ as the minimization of a data-driven loss $L_\tau^{\text{train}}(\phi)$, regularized by a quadratic term $\lambda||\phi - \omega||^2$. This regularizer can be interpreted as a model of a complex synapse; similar models have been introduced in studies of continual learning and long-term memory [69–73]. In the context of meta-learning, the fast weights $\phi$ quickly learn how to solve a task and $\lambda$ determines how strongly $\phi$ is pulled towards the slow weights $\omega$, which consolidate the knowledge that has been previously acquired. The capabilities of the learning algorithm are measured by evaluating $L_\tau^{\text{eval}}(\phi_\tau^*)$, the performance of the learned neural network on data from task $\tau$ that was held out during learning. Meta-learning finally corresponds to improving the learning performance on many tasks by adjusting the meta-parameters $\theta = \omega$, leading to the following problem:

$$\min_\theta \; \mathbb{E}_\tau \big[ L_\tau^{\text{eval}}(\phi_\tau^*) \big] \quad \text{s.t.} \;\; \phi_\tau^* = \arg\min_\phi L_\tau^{\text{learn}}(\phi) + \frac{\lambda}{2}\|\phi - \omega\|^2. \tag{16}$$

Above, the parameters $\phi$ no longer represent neural activities and $\theta$ synaptic weights as in the previous section; $\phi$ and $\theta = \omega$ now designate synaptic weights and meta-parameters, respectively.

**Meta-parameter updates.** Using the first-order conditions $-\nabla_\phi L_\tau^{\text{train}}(\phi_\tau^*) - \lambda(\phi_\tau^* - \theta) = 0$ associated to $\phi_\tau^*$ being a minimizer in the previous equation yields an optimization problem on which the least-control principle can be applied. Our principle prescribes running the controlled dynamics

$$\begin{aligned} \dot{\phi} &= -\nabla_\phi L_\tau^{\text{learn}}(\phi) - \lambda(\phi - \omega) + u, \\ \dot{u} &= -\alpha u - \nabla_\phi L_\tau^{\text{eval}}(\phi) \end{aligned} \tag{17}$$

or any other process that finds the same fixed points $(\phi_*^\tau, u_*^\tau)$. The theory we developed in Section 2.2 guarantees that $u_*^\tau$ is an optimal control as long as $\alpha \to 0$ and the Hessian of the evaluation loss $\partial_\phi^2 L_\tau^{\text{eval}}$ at the equilibrium $\phi_*^\tau$ is invertible. Hence, we do not need complex dynamics to compute the gradient, highlighting the flexibility and simplicity of learning according to the least-control principle. The resulting single-phase meta-learning rule is local to every synapse (c.f. Section S7):

| Method | Test accuracy |
|---|---|
| FOMAML [74] | $89.40^{\pm 0.50}$ |
| T1-T2 [75] | $89.72^{\pm 0.43}$ |
| Reptile [76] | $89.43^{\pm 0.14}$ |
| LCP (ours) | $91.00^{\pm 0.24}$ |
| iMAML [68] | $94.46^{\pm 0.42}$ |
| CML [35] | $94.16^{\pm 0.12}$ |

Table 3: Meta-learning neural networks with complex synapses on Omniglot 20-way 1-shot learning tasks. Mean $\pm$ std test set classification accuracy (%) computed over 3 seeds.

$$\Delta\omega = \lambda u_*^\tau. \tag{18}$$

**Experimental results.** Our meta-learning algorithm stands out from competing methods as being both principled, first-order and single-phase. On the one hand, existing first-order and single-phase methods either approximate meta-gradients (e.g., FOMAML [74] and T1-T2 [75]) or rely on heuristics (e.g. Reptile [76]), and therefore do not provably minimize a meta-objective function. On the other hand, full meta-gradient descent methods require two phases to backpropagate through learning [74], invert an intractable Hessian [68, 77], or learn a slightly modified task [35], and often require second derivatives [68, 74, 77]. Here, we compare our rule to competing first-order methods (FOMAML, T1-T2 and Reptile) and meta-gradient approximation algorithms that use the same synaptic model (implicit MAML and contrastive meta-learning; iMAML and CML) on the Omniglot few-shot visual classification benchmark [78]. Each Omniglot task consists of distinguishing 20 classes after seeing only 1 sample per class; we choose this 20-way 1-shot regime as the gap between first- and second-order methods is reputedly the largest compared to other variants of this benchmark. We find that our meta-learning rule significantly outperforms other single-phased first-order methods, while still lagging behind iMAML, a second-order and two-phased method, and CML, a two-phased first-order method that can approximate the meta-gradient arbitrarily well. These findings are particularly interesting since our synaptic model (16) is in the underparameterized regime: there is a separate set of weights $\phi_\tau$ per task $\tau$ but only one set of consolidated weights $\omega$ for the whole task distribution. The strong performance observed here is another demonstratation that least-control solutions can perform well even when overparameterization arguments (cf. Section 2.3) do not hold.

# 4  Discussion

**Control theories of neural network learning.** The relationship between control theory and neural network learning is an old and intricate one. It is by now well known that backpropagation, the method of choice for computing gradients in deep learning, can be seen as the discrete-time counterpart of the adjoint method developed in control theory for continuous-time systems [79, 80]. One of the early papers realizing this connection formulated neural network learning as an optimization problem under equilibrium constraints, as in the original problem statement (1), and rederived the recurrent backpropagation algorithm [81]. More recent work has explored using feedback neural control circuits which influence the neural dynamics of feedforward [30, 51] and recurrent [82–84] networks to deliver learning signals. Different from such approaches, our least-control principle is rooted in *optimal* control: instead of setting the control from an ansatz, we derived it from first principles. This allowed us to arrive at a simple local rule for gradient-based learning, and to identify general sufficient conditions for the feedback control signals.

**Predictive coding from the least-control perspective.** We have shown that least-control problems correspond to free-energy minimization problems. This dual view of least-control problems allows connecting to influential theories of neural information processing based on predictive coding [19, 36, 37, 56, 58]. Our results elucidate predictive coding theories in at least three ways. First, Theorem 1 justifies the parameter update taken in supervised predictive coding networks [36] as gradient-based optimization of our least-control objective. This connection offers a novel view of the predictive coding dynamics as an optimal control algorithm in charge of controlling hidden neurons, while 'clamping' output neurons to the respective targets can be seen as a form of output control. Previous arguments for gradient-based learning were only known for the weak nudging limit ($\beta \to 0$) where learning signals are vanishingly small [24], which is problematic in noisy systems such as the cortex [85]; our novel theoretical and experimental findings characterize the opposite perfect control ($\beta \to \infty$) end of the spectrum. Second, Theorem 2 broadens the class of neural network circuits which may be used to implement predictive coding, beyond the precisely-constructed microcircuits that have been proposed so far [17, 36]. Finally, our experiments on recurrent neural networks using free-energy-minimizing dynamics to assign credit are to the best of our knowledge the first of the kind, and serve to validate the effectiveness of predictive coding circuits beyond feedforward architectures.

**Limitations.** A limiting factor of our theory is that in its current form it is only applicable to equilibrium systems. While the study of neural dynamics at equilibrium is an old endeavor in neuroscience and neural network theory [86, 87], it remains unclear how strong an assumption it is when modeling the networks of the brain. Extending our principle to out-of-equilibrium dynamic is an exciting direction for future work. Another limitation concerning the breadth of problem (1) is that we did not allow the loss function $L$ to directly depend on the parameters $\theta$. We discuss extensions to this more general case in Section S2.2, however not all loss functions $L$ in this more general class satisfy the conditions needed to obtain a local parameter update. Lastly, it should be noted that although our principle requires only one relaxation phase, its computational cost often exceeds the combined cost of performing the two phases of recurrent backpropagation (cf. Section S6.6).

**Concluding remarks.** We have presented a new theory that enables learning equilibrium systems with local gradient-based rules. Our learning rules are driven by changes in activity generated by an optimal control, in charge of feeding back credit assignment information into the system as it evolves. Taken together, our results push the boundaries of what is possible with single-phase, activity-dependent local learning rules, and suggest an alternative to direct gradient-based learning which yields high performance in practice.

## Acknowledgments and Disclosure of Funding

This research was supported by an Ambizione grant (PZ00P3_186027) from the Swiss National Science Foundation and an ETH Research Grant (ETH-23 21-1) awarded to João Sacramento. Seijin Kobayashi was supported by the Swiss National Science Foundation (SNF) grant CRSII5_173721. Johannes von Oswald was funded by the Swiss Data Science Center (J.v.O. P18-03). We thank Angelika Steger, Benjamin Scellier, Walter Senn, Il Memming Park, Blake A. Richards, Rafal Bogacz, Yoshua Bengio, and members of the Senn, Bogacz and Richards labs for discussions.

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
