# Supplementary Materials

**Alexander Meulemans\*, Nicolas Zucchet\*, Seijin Kobayashi\*, Johannes von Oswald, João Sacramento**

# Table of Contents

# S1 Notation

In this section, we provide an extensive list of the notations we use. In particular, we detail what the different quantities precisely mean for learning recurrent neural networks and for meta-learning.

## S1.1 General notations

| Notation | Meaning |
|---|---|
| $\frac{\partial}{\partial x} = \partial_x$ | Partial derivative of a function. The derivative of a scalar function is a row vector of size $\lvert x \rvert$. |
| $\frac{\mathrm{d}}{\mathrm{d}x} = \mathrm{d}_x$ | Total derivative of a function. The derivative of a scalar function is a row vector of size $\lvert x \rvert$. |
| $\nabla_x$ | Gradient of a scalar function (column vector). We have $\nabla_x \cdot = (\mathrm{d}_x \cdot)^\top$. |
| $\dot{x}$ | Time derivative of $x$ (column vector). |
| $\lVert \cdot \rVert$ | Euclidean (L2) norm. |
| $\min_x$ | Minimum of a function with respect to $x$. |
| $\arg\min_x$ | Set of $x$ that (locally) minimize a function. |
| Id | Identity matrix (size defined by the context). |
| $\mathrm{col}[A]$ | Column space of $A$, that is $\mathrm{Im}[A]$. |
| $\mathrm{row}[A]$ | Row space of $A$, that is $\mathrm{Im}[A^\top]$. |

## S1.2 Notations for the least-control principle in general

| Notation | Meaning |
|---|---|
| $\phi$ | Dynamical parameters of the system. |
| $\theta$ | Parameters of the system. |
| $f(\phi, \theta)$ | Vector-field of the (free) dynamics on $\phi$. |
| $L$ | Learning loss that the system should minimize. |
| $y = h(\phi)$ | Output units (i.e. units on which the loss is evaluated) of the system. |
| $\mathcal{H}$ | Least-control objective. |
| $\psi$ | Controller whose goal is to help the system reaching a loss-minimizing state. |
| $u$ | Controller on the output units. |
| $Q(\phi, \theta)$ | General feedback mapping from output control $u$ to the entire system control $\psi$. We have $\psi = Q(\phi, \theta)u$ at equilibrium. |
| $\phi^*$ | Equilibrium of the free dynamics. |
| $\phi_\star, \psi_\star, u_\star$ | Optimal state, control and output control according for the least-control objective (c.f. Eq. 3). |
| $\phi_*, \psi_*, u_*$ | Equilibrium of the controlled dynamics. The least-control principle aims for $\psi_* = \psi_\star$, that is to find an optimal control through the steady state of controlled dynamics. |
| $\alpha$ | Leakage strength in a leaky integral controller ($\alpha = 0$ yields pure integral control). |
| $\beta$ | Strength of the proportional control, or of nudging in the energy-based view. |

## S1.3 Notations for recurrent neural networks

| Notation | Meaning |
|---|---|
| $\phi$ | Neural activities of all the neurons in the network (column vector). |
| $\theta = \{W, S\}$ | Learnable synaptic connection weights of the network. |
| $f(\phi, \theta)$ | Dynamics of the neural network. |
| $y = D\phi$ | The binary matrix $D$ selects output neurons $u$ from the rest of them $\phi$. |
| $\sigma$ | Non-linear activation function. |
| $x$ | Input of the neural network. |
| $y^{\mathrm{true}}$ | Desired output corresponding to an input $x$. |
| $L$ | Loss measuring the discrepancy between outputs of the network $y$ (that depends on $x$) and target output $y^{\mathrm{true}}$, averaged over all the pairs $(x, y^{\mathrm{true}})$ from a data set. |
| $\gamma$ | Weight decay for the Kollen-Pollack learning-rule. |

## S1.4 Notations for meta-learning

| Notation | Meaning |
|---|---|
| $\phi$ | Parameters (weights and biases) of the neural network that is learned. |
| $\theta = \{\omega, \lambda\}$ | Parameters of the learning algorithm, here the consolidated synaptic weights $\omega$, and eventually, the strength of attraction to those weights $\lambda$. |
| $f(\phi, \theta)$ | Dynamics of the learning algorithm. |
| $\tau$ | Index of the task considered. |
| $L_\tau^{\text{learn}}(\phi)$ | Data-driven loss that is combined with a prior-term to obtain the loss minimized by the learning algorithm. |
| $L_\tau^{\text{eval}}(\phi)$ | Loss measuring the performance of the output of the learning algorithm. |
| $\psi = u$ | Controller on all the units (they are all output units in this context). |

## S2 Proofs for the least-control principle

The purpose of this section is to prove all the theoretical results behind the least-control principle. Recall that our principle advocates to solve the following constrained optimization problem

$$\min_{\theta} \min_{\phi, \psi} \frac{1}{2}\|\psi\|^2 \quad \text{s.t.} \ f(\phi, \theta) + \psi = 0, \ \frac{\partial L}{\partial y}(h(\phi)) = 0, \tag{19}$$

that is minimizing the amount of control needed to reach a loss-minimizing state. Finding an optimal control $\psi_\star$ consists in solving the least-control problem:

$$(\phi_\star, \psi_\star) = \arg\min_{\phi, \psi} \frac{1}{2}\|\psi\|^2 \quad \text{s.t.} \ f(\phi, \theta) + \psi = 0, \ \frac{\partial L}{\partial y}(h(\phi)) = 0. \tag{20}$$

Most of the proofs that we present here rely on interpreting the problem above as a constrained minimization problem, and using the associated first-order stationarity condition to manipulate $\psi_\star$ and $\phi_\star$. Compared to the main text, we introduced the notation $y = h(\phi)$ in this formulation. It is here to select output units from the rest and clarifies mathematical manipulations.

### S2.1 Constrained optimization and Lagrange multipliers

Throughout the proofs for the least-control principle, we characterize the optimal control $\psi_\star$ and state $\phi_\star$ through the stationarity conditions that they verify, that are known as the KKT conditions. In particular, this allows considering $(\phi_\star, \psi_\star)$ as an implicit function of $\theta$, and computing derivatives through the implicit function theorem [89]. We provide a quick review on constrained optimization in Section S2.7 for the reader who is new to the topic.

We now state what are the KKT conditions for the constrained optimization problem (20). To do so, we introduce the corresponding Lagrangian, which we denote as the LCP-Lagrangian:

$$\mathcal{L}(\phi, \psi, \lambda, \mu, \theta) := \frac{1}{2}\|\psi\|^2 + \lambda^\top \left(f(\phi, \theta) + \psi\right) + \frac{\partial L}{\partial y}(h(\phi))\mu, \tag{21}$$

where $\lambda$ and $\mu$ are the Lagrange multipliers associated to the constraints $f(\phi, \theta) + \psi = 0$ and $\nabla_y L(h(\phi)) = 0$. Then, $(\phi_\star, \psi_\star)$ verifies the KKT conditions associated to Eq. 20 if there exists $(\lambda_\star, \mu_\star)$ such that

$$\begin{cases} \dfrac{\partial \mathcal{L}}{\partial \phi}(\phi_\star, \psi_\star, \lambda_\star, \mu_\star, \theta) = \lambda_\star^\top \dfrac{\partial f}{\partial \phi}(\phi_\star, \theta) + \mu_\star^\top \dfrac{\partial^2 L}{\partial y^2}(h(\phi_\star)) \dfrac{\partial h}{\partial \phi}(\phi_\star) = 0 \\[2mm] \dfrac{\partial \mathcal{L}}{\partial \psi}(\phi_\star, \psi_\star, \lambda_\star, \mu_\star, \theta) = \psi_\star^\top + \lambda_\star^\top = 0 \\[2mm] \dfrac{\partial \mathcal{L}}{\partial \lambda}(\phi_\star, \psi_\star, \lambda_\star, \mu_\star, \theta) = f(\phi_\star, \theta)^\top + \psi_\star^\top = 0 \\[2mm] \dfrac{\partial \mathcal{L}}{\partial \mu}(\phi_\star, \psi_\star, \lambda_\star, \mu_\star, \theta) = \dfrac{\partial L}{\partial y}(h(\phi_\star)) = 0. \end{cases} \tag{22}$$

In the proofs that follow, we use equivalently that $(\phi_\star, \psi_\star)$ is optimal for Eq. 20 and that it verifies the KKT conditions of Eq. 22. We need to invoke sufficient conditions, such as the positive definiteness of the Hessian of the Lagrangian, to make it rigorous and to show that a state satisfying the KKT conditions is a local minimizer; we discuss this matter in more details in Section S2.7. However, we omit these considerations in the proofs to keep the arguments as concise as possible.

### S2.2 Theorem 1: first-order gradient

We first state the formal version of Theorem 1 and then prove it, by differentiating through the KKT conditions (22). Note that we provide an alternate proof that leverages the energy-based formulation of the principle in Section S2.5.

**Theorem S1** (Formal). *Let $(\phi_\star, \psi_\star)$ an optimal control for the least-control problem* (20) *and $(\lambda_\star, \mu_\star)$ the Lagrange multipliers for which the KKT conditions are satisfied. Under the assumption*

that the Hessian of the LCP-Lagrangian $\partial^2_{\phi,\psi,\lambda,\mu}\mathcal{L}(\phi_\star, \psi_\star, \lambda_\star, \mu_\star, \theta)$ is invertible, the least-control principle yields the following gradient for $\theta$:

$$\left(\frac{\mathrm{d}}{\mathrm{d}\theta}\mathcal{H}(\theta)\right)^\top = -\frac{\partial f}{\partial \theta}(\phi_\star, \theta)^\top \psi_\star = \frac{\partial f}{\partial \theta}(\phi_\star, \theta)^\top f(\phi_\star, \theta). \tag{23}$$

*Proof.* The proof relies on two ingredients: rewriting $\mathcal{H}$ using the LCP-Lagrangian and then differentiating through this Lagrangian.

We start by rewriting the least-control objective $\mathcal{H}(\theta)$ using the LCP-Lagrangian. Recall that $\mathcal{H}$ is defined as

$$\mathcal{H}(\theta) = \frac{1}{2}\|\psi_\star\|^2. \tag{24}$$

As $(\phi_\star, \psi_\star)$ is an optimal control, there indeed exists Lagrange multipliers $(\lambda_\star, \mu_\star)$ such that $\partial_{\phi,\psi,\lambda,\mu}\mathcal{L}(\phi_\star, \psi_\star, \lambda_\star, \mu_\star, \theta) = 0$, as assumed in the statement of the theorem. We use this equality condition to implicitly define $(\phi_\star, \psi_\star, \lambda_\star, \mu_\star)$ as functions of $\theta$. The implicit function theorem along with the assumption that the Hessian is invertible ensures that these functions are well defined and differentiable. As all the constraints are satisfied for $(\phi_\star, \psi_\star)$, we have

$$\begin{aligned}
\mathcal{H}(\theta) &= \mathcal{L}(\phi_\star, \psi_\star, \lambda_\star, \mu_\star, \theta) \\
&= \frac{1}{2}\|\psi_\star\|^2 + \lambda_\star^\top (f(\phi_\star, \theta) + \psi_\star) + \frac{\partial L}{\partial y}(h(\phi_\star))\mu_\star \\
&= \frac{1}{2}\|\psi_\star\|^2
\end{aligned} \tag{25}$$

We can then calculate $\mathrm{d}_\theta \mathcal{H}(\theta)$:

$$\begin{aligned}
\frac{\mathrm{d}}{\mathrm{d}\theta}\mathcal{H}(\theta) &= \frac{\mathrm{d}}{\mathrm{d}\theta}\mathcal{L}(\phi_\star, \psi_\star, \lambda_\star, \mu_\star, \theta) \\
&= \frac{\partial \mathcal{L}}{\partial \theta} + \frac{\partial \mathcal{L}}{\partial \phi}\frac{\mathrm{d}\phi_\star}{\mathrm{d}\theta} + \frac{\partial \mathcal{L}}{\partial \psi}\frac{\mathrm{d}\psi_\star}{\mathrm{d}\theta} + \frac{\partial \mathcal{L}}{\partial \lambda}\frac{\mathrm{d}\lambda_\star}{\mathrm{d}\theta} + \frac{\partial \mathcal{L}}{\partial \mu}\frac{\mathrm{d}\mu_\star}{\mathrm{d}\theta} \\
&= \frac{\partial \mathcal{L}}{\partial \theta} + 0 + 0 + 0 + 0 \\
&= \lambda_\star^\top \frac{\partial f}{\partial \theta}(\phi_\star, \theta),
\end{aligned} \tag{26}$$

where we used the chain rule for the second equation, and the stationarity equations of the Lagrangian for the third equation. Through the stationarity equation $\partial_\psi \mathcal{L}(\phi_\star, \psi_\star, \lambda_\star, \mu_\star, \theta) = 0$, we have that $\lambda_\star = -\psi_\star$. The constraint on $\psi$ additionally implies that $\psi_\star = -f(\phi_\star, \theta)$, which finishes the proof. $\square$

**Extension to $\theta$-dependent losses.** Throughout this work, we focus on loss functions $L(\phi)$ that do not explicitly depend on $\theta$. The influence of $\theta$ on the loss only appears through the equilibrium condition $f(\phi, \theta) = 0$. Still, we can generalize Theorem S1 further by also considering loss functions that explicitly depend on $\theta$, on top of their implicit dependence on $\theta$ through $\phi$. A simple example is when we add a weight regularization term $\|\theta\|^2$ to the loss $L$ when training neural networks, or in meta-learning when $\theta$ does not only impact the learning algorithm of the system but also its prediction [90–92].

The proof of Theorem S1 can be slightly adapted to hold for the more general case, the only difference is that the equality $\partial_\theta \mathcal{L} = \lambda_*^\top \partial_\theta f(\phi_*, \theta)$ is no longer true. Instead, we have

$$\begin{aligned}
\frac{\mathrm{d}}{\mathrm{d}\theta}\mathcal{H}(\theta) &= \frac{\partial \mathcal{L}}{\partial \theta}(\phi_\star, \psi_\star, \lambda_\star, \mu_\star, \theta) \\
&= -\psi_\star^\top \frac{\partial f}{\partial \theta}(\phi_\star, \theta) + \mu_\star^\top \frac{\partial^2 L}{\partial \theta \partial y}(\phi_\star, \theta).
\end{aligned} \tag{27}$$

The second derivative $\partial_\theta \partial_y L(h(\phi_\star), \theta)$ captures mixed dependencies of the loss $L$ on $y$ and $\theta$. Note that when the loss $L$ contains no terms mixing $h(\phi)$ and $\theta$, as in the weight decay example we mentioned earlier, this second-order derivative is zero. As for this weight decay example the resulting

updates are the same as for the loss without any weight decay, this indicates that some $\theta$-dependent losses are not captured in the least-control principle we introduced in the main text.

We now propose an extended formulation of the least-control principle to fully incorporate $\theta$-dependent losses:

$$\min_{\theta} \min_{\phi, \psi} \frac{1}{2} \|\psi\|^2 + L(\phi, \theta) \quad \text{s.t.} \quad f(\phi, \theta) + \psi = 0, \quad \nabla_\phi L(\phi, \theta) = 0. \tag{28}$$

Intuitively, the least-control objective $\|\psi\|^2$ combined with the constraint that the loss is minimized w.r.t. $\phi$ at the controlled equilibrium captures all the ways $\theta$ influences the loss through $\phi$, and the new term $L(\phi, \theta)$ takes care of the ways in which $\theta$ influences the loss without influencing $\phi$, while keeping the same stationarity conditions for the LCP-Lagrangian (the extra term $\nabla_y L(h(\phi_\star), \theta)$ that now appears in $\partial_\phi \mathcal{L}$ vanishes). Using the same proof strategy as in Theorem 1, we obtain the following gradient:

$$
\begin{aligned}
\frac{\mathrm{d}}{\mathrm{d}\theta} \mathcal{H}(\theta) &= \frac{\partial \mathcal{L}}{\partial \theta} (\phi_\star, \psi_\star, \lambda_\star, \mu_\star, \theta) \\
&= -\psi_\star^\top \frac{\partial f}{\partial \theta} (\phi_\star, \theta) + \mu_\star^\top \frac{\partial^2 L}{\partial \theta \partial y} (\phi_\star, \theta) + \frac{\partial L}{\partial \theta} (\phi_\star, \theta).
\end{aligned}
\tag{29}
$$

This generalized least-control principle now allows to fully consider terms in the loss that depend on $\theta$, while keeping the nice properties of being single-phased, having no need to invert any Jacobian, and using no infinitesimal learning signals. However, now the learning rule is not always first-order anymore due the mixed second-order derivative.

One special case of interest where the resulting learning rule is first-order again is when the $\theta$-dependence of the loss is encapsulated in the decoder $h(\phi, \theta)$, i.e. when we have the loss $L(h(\phi, \theta))$. Applying Eq. 29 to this loss results in:

$$
\begin{aligned}
\frac{\mathrm{d}}{\mathrm{d}\theta} \mathcal{H}(\theta) &= -\psi_\star^\top \frac{\partial f}{\partial \theta} (\phi_\star, \theta) + \mu_\star^\top \frac{\partial^2 L}{\partial y^2} (h(\phi_\star, \theta)) \frac{\partial h}{\partial \theta} (\phi_\star, \theta) + \frac{\partial L}{\partial y} (h(\phi_\star, \theta)) \frac{\partial h}{\partial \theta} (\phi_\star, \theta). \\
&= -\psi_\star^\top \frac{\partial f}{\partial \theta} (\phi_\star, \theta) + \mu_\star^\top \frac{\partial^2 L}{\partial y^2} (h(\phi_\star, \theta)) \frac{\partial h}{\partial \theta} (\phi_\star, \theta) + 0
\end{aligned}
\tag{30}
$$

where the last term vanishes as $\partial_y L(h(\phi_\star, \theta)) = 0$ is enforced by the constraints of the least-control objective of Eq. 3. When we use the inversion dynamics (8) or the energy-based dynamics (10), we can easily access the Lagrange multiplier $\mu_\star$. For this, let us take a look at the equilibrium equation of the inversion dynamics (8), which is equivalent to the equilibrium condition of the energy-based dynamics (10) if $\partial_\phi f(\phi_\star, \theta)$ is invertible:

$$
0 = \left( \frac{\partial f(\phi_*, \theta)}{\partial \phi} \right)^\top \psi_* + \frac{\partial h(\phi_*, \theta)^\top}{\partial \phi} u_*
\tag{31}
$$

Comparing this equilibrium condition with the KKT conditions of Eq. 22, and using the relation $\lambda_\star = -\psi_\star$, we have that

$$
\frac{\partial^2 L(h(\phi_*, \theta))}{\partial y^2} \mu_\star = -u_*
\tag{32}
$$

when $\partial_\phi h(\phi_*, \theta)$ is full rank and in the limit of $\alpha \to 0$ such that $(\phi_*, \psi_*) = (\phi_\star, \psi_\star)$. Taking everything together, we arrive at the following simple first-order update rule for models with a $\theta$-dependent decoder $h$, in the limit of $\alpha \to 0$:

$$
\frac{\mathrm{d}}{\mathrm{d}\theta} \mathcal{H}(\theta) = -\psi_*^\top \frac{\partial f}{\partial \theta} (\phi_*, \theta) - u_*^\top \frac{\partial h(\phi_*, \theta)}{\partial \theta}
\tag{33}
$$

Note that this update only holds when the inversion dynamics (8) or energy-based dynamics (10) are used. If other dynamics are used for finding the optimal control $\psi_\star$, one needs to investigate case-by-case how to compute the Lagrange multiplier $\mu_\star$.

## S2.3 Theorem 2: general controller dynamics

Theorem 2 shows that we can find optimal control $\psi_\star$ and state $\phi_\star$ for Eq. 20 through dynamics exhibiting the following equilibrium equations

$$0 = f(\phi_*^\alpha, \theta) + Q(\phi_*^\alpha, \theta)u_*^\alpha \text{ and } 0 = -\nabla_y L(h(\phi_*^\alpha)) - \alpha u_*^\alpha \tag{34}$$

in the limit $\alpha \to 0$, when $Q(\phi_*^\alpha, \theta)$ satisfies the column space condition (35). We prove that by showing that the limit satisfies the KKT conditions, and hence is optimal. We restate Theorem 2 below with the full technical specifications.

**Theorem S2** (Formal). *Let $(\phi_*^\alpha, u_*^\alpha)$ be a steady state of the generalized dynamics satisfying* (34) *such that $\phi_*^\alpha$ admits a finite limit $\phi_\star$ when $\alpha$ goes to 0. Assume that $\partial_\phi f(\phi_\star)$ and $\partial_y^2 L(\phi_\star)$ are invertible, $\partial_\phi h(\phi_\star)$ is full rank, $Q$ is continuous, and that the following column space condition is satisfied:*

$$\text{col}\left[Q(\phi_\star, \theta)\right] = \text{row}\left[\frac{\partial h}{\partial \phi}(\phi_\star)\left(\frac{\partial f}{\partial \phi}(\phi_\star, \theta)\right)^{-1}\right] \tag{35}$$

*Then, $u_*^\alpha$ converges to a finite limit $u_\star$ and $(\phi_\star, Q(\phi_\star, \theta)u_\star)$ verifies the KKT conditions for the least-control problem* (20).

*Proof.* Note that there is a slight abuse of notation in the statement of the theorem as we use the superscript $_\star$ to denote the limits of the different quantities without yet knowing if they correspond to an optimal quantity for the least-control problem (20). Part of the proof is to actually show that those limits are optimal.

First, we show that $u_*^\alpha$ has a finite limit when $\alpha$ goes to 0. The function $\phi_*^\alpha$ admits a finite limit $\phi_\star$ when $\alpha$ goes to zero by hypothesis. Taking $\alpha \to 0$ in the equilibrium equations (34) results in

$$f(\phi_\star, \theta) - \lim_{\alpha \to 0} \frac{1}{\alpha} Q(\phi_*^\alpha, \theta)\nabla_y L(h(\phi_*^\alpha)) = 0, \tag{36}$$

as $f$ is continuous. The matrices $\partial_\phi h(\phi_\star)$ and $\partial_\phi f(\phi_\star, \theta)^{-1}$ are full rank, so the column space condition implies that $Q(\phi_\star, \theta)$ is also full rank. We can therefore extract $Q(\phi_*^\alpha, \theta)$ out of the limit in the previous equation:

$$\lim_{\alpha \to 0} \frac{1}{\alpha} Q(\phi_*^\alpha, \theta)\nabla_y L(h(\phi_*^\alpha)) = Q(\phi_\star, \theta) \lim_{\alpha \to 0} \frac{1}{\alpha} \nabla_y L(h(\phi_*^\alpha)), \tag{37}$$

so that $u_*^\alpha = -\alpha^{-1}\nabla_y L(h(\phi_*^\alpha))$ has a finite limit that we note $u_\star$, using once again the fact that $Q(\phi_\star, \theta)$ is full rank and (36). In particular, this implies that $\phi_\star$ is feasible as $\nabla_y L(h(\phi_*^\alpha)) = O(\alpha)$ so $\nabla_y L(h(\phi_\star)) = 0$.

We can now prove that the KKT conditions are satisfied at $(\phi_\star, \psi_\star)$ with $\psi_\star := Q(\phi_\star, \theta)u_\star$. Note that we have $f(\phi_\star, \theta) + \psi_\star = 0$ by taking the limit $\alpha \to 0$ in the equilibrium equation

$$f(\phi_*^\alpha, \theta) - \frac{1}{\alpha} Q(\phi_*^\alpha, \theta)\nabla_y L(h(\phi_*^\alpha)) = 0. \tag{38}$$

As $\text{col}\left[Q(\phi_\star, \theta)\right] = \text{row}\left[\partial_\phi h(\phi_\star)\partial_\phi f(\phi_\star, \theta)^{-1}\right]$, and $\partial_y^2 L(h(\phi_\star))$ is invertible, there exist a $\mu$ and $\mu_\star$, such that

$$\begin{aligned}
\psi_\star &= Q(\phi_\star, \theta)u_\star \\
&= \frac{\partial f}{\partial \phi}(\phi_\star, \theta)^{-\top}\frac{\partial h}{\partial \phi}(\phi_\star)^\top \mu \\
&= \frac{\partial f}{\partial \phi}(\phi_\star, \theta)^{-\top}\frac{\partial h}{\partial \phi}(\phi_\star)^\top \frac{\partial^2 L}{\partial y^2}(\phi_\star)\mu_\star.
\end{aligned} \tag{39}$$

Multiplying from the left with $\partial_\phi f(\phi_\star, \theta)^\top$ results in

$$-\frac{\partial f}{\partial \phi}(\phi_\star, \theta)^\top \psi_\star + \frac{\partial h}{\partial \phi}(\phi_\star)^\top \frac{\partial^2 L}{\partial y^2}(\phi_\star)\mu_\star = 0. \tag{40}$$

We now take $\lambda_\star := -\psi_\star$ and can easily check that $(\phi_\star, \psi_\star, \lambda_\star, \mu_\star)$ is a stationary point for the LCP-Lagrangian, i.e. it satisfies the KKT conditions (20). It follows that $\psi_\star$ is an optimal control and $\phi_\star$ an optimally-controlled state. $\square$

## S2.4 Propositions 3, 4, 5: the least-control principle solves the original learning problem

We here restate the results from Section 2.3 and prove them.

**Proposition S3.** *Assuming $L$ is convex on the system output $y$, we have that the optimal control $\psi_\star$ is equal to 0 iff. the free equilibrium $\phi^*$ minimizes $L$.*

*Proof.* First remark that, as $L$ is convex, $\phi$ being a minimizer of $L(h(\phi))$ is equivalent to $\nabla_y L(h(\phi)) = 0$. Then, the two conditions are equivalent to $\phi_\star$ and $\phi^*$ satisfying both $f(\phi, \theta) = 0$ and $\nabla_y L(h(\phi)) = 0$. □

**Proposition S4.** *Assuming $L$ is convex on the system output $y$, a local minimizer $\theta$ of the least-control objective $\mathcal{H}$ is a global minimizer of the original learning problem* (1), *under the sufficient condition that $\partial_\theta f(\phi_\star, \theta)$ has full row rank.*

*Proof.* As $\theta$ is a local minimizer of $\mathcal{H}(\theta)$, we have that

$$\frac{\mathrm{d}}{\mathrm{d}\theta} \mathcal{H}(\theta) = f(\phi_\star, \theta) \frac{\partial f}{\partial \theta}(\phi_\star, \theta) = 0. \tag{41}$$

The full row rank assumption of $\partial_\theta f(\phi_\star, \theta)$ ensures that $f(\phi_\star, \theta) = 0$. We hence have that $\phi_\star$ is a global minimizer of $L(h(\phi))$, as $\nabla_y L(h(\phi_\star)) = 0$, and $L$ is convex, and satisfies $f(\phi_\star, \theta) = 0$. It follows that $\theta$ is a global minimizer for the original learning problem. □

*Remark* S1. In the last two propositions, we assumed that the loss $L$ is convex on output units. This is for example the case in our supervised learning experiments, but not for meta-learning. If we relax this assumption, Propositions 3 and 4 still hold, with the difference that global minimizers become local ones.

Additionally, the result of those two propositions does not only hold in the strong control limit but also in the $\alpha > 0$ regime, under the strict feedback condition of Proposition S9. The proof is a direct combination of the proof of the two propositions above with the stationarity result of Proposition S9.

**Proposition S5.** *If $\frac{1}{2}\|f\|^2$ is $\mu$-strongly convex as a function of $\phi$, $L \circ h$ is $M$-Lipschitz continuous and the minimum of $L$ is 0, then*

$$L(h(\phi^*)) \leq \frac{\sqrt{\mu}}{\sqrt{2M}} \|\psi_\star\| \leq \frac{\sqrt{\mu}}{\sqrt{2M}} \|f(\phi_\star, \theta)\|. \tag{42}$$

*Proof.* As $h(\phi_*)$ minimizes $L$ and $L$ is Lipschitz-continuous,

$$L(h(\phi^*)) = L(h(\phi^*)) - L(h(\phi_\star)) \leq M\|\phi^* - \phi_\star\|. \tag{43}$$

The strong convexity of $\frac{1}{2}\|f\|^2$ yields

$$\frac{1}{2}\|f(\phi_\star, \theta)\| = \frac{1}{2}\|f(\phi_\star, \theta)\|^2 - \frac{1}{2}\|f(\phi^*, \theta)\|^2 \geq \frac{\mu}{2}\|\phi_\star - \phi^*\|^2, \tag{44}$$

as $\phi^*$ is a global minimizer of $\frac{1}{2}\|f\|^2$. Gathering the last two inequalities gives the desired result. □

We state and prove the remaining informal statements on the impact of approximate equilibria appearing in Section 2.3 in Section S2.6, as they require the energy-based view of the least control principle to be proved.

## S2.5 The least-control principle as constrained energy minimization

In Section 2.4, we show that substituting the constraint $f(\phi, \theta) + \psi = 0$ back into the least-control problem implies that $\phi_\star$ is a minimizer of

$$\min_\phi \frac{1}{2}\|f(\phi, \theta)\|^2 \quad \text{s.t.} \quad \nabla_y L(h(\phi)) = 0. \tag{45}$$

The same objective can be obtained from an energy-based perspective, by minimizing the augmented energy $F(\phi, \theta, \beta) = \frac{1}{2}\|f(\phi, \theta)\|^2 + \beta L(h(\phi))$ and taking the limit $\beta \to +\infty$. We formalize this result by first showing that the limit of stationary points of $F$ when $\beta \to \infty$ satisfies the

KKT conditions (Proposition S6), and that the global minimizers coincide, under more restrictive assumption (Proposition S7). Note that the KKT conditions associated to (45) are different from the ones in Eq. 22: the corresponding Lagrangian is

$$\mathcal{L}(\phi, \mu, \theta) := \frac{1}{2} \|f(\phi, \theta)\|^2 + \mu^\top \left( \nabla_y L(h(\phi)) \right) \tag{46}$$

and KKT conditions are

$$\begin{cases} \dfrac{\partial \mathcal{L}}{\partial \phi}(\phi_\star, \mu_\star, \theta) = \dfrac{\partial}{\partial \phi} \left[ \dfrac{1}{2} \|f(\phi_\star, \theta)\|^2 \right] + \mu_\star^\top \dfrac{\partial^2 L}{\partial y^2}(h(\phi_\star)) \dfrac{\partial h}{\partial \phi}(\phi_\star) = 0 \\ \dfrac{\partial \mathcal{L}}{\partial \mu}(\phi_\star, \mu_\star, \theta) = \nabla_y L(h(\phi_\star)) = 0 \end{cases}. \tag{47}$$

**Proposition S6.** *Let $\phi_*^\beta$ be a function of $\beta$ that admits a finite limit $\phi_\star$ when $\beta$ goes to infinity and verifies*

$$\frac{\partial F}{\partial \phi}(\phi_*^\beta, \beta, \theta) = 0 \tag{48}$$

*for every $\beta$. If we further assume that the loss Hessian $\partial_y^2 L(h(\phi_\star))$ is invertible and $\partial_\phi h(\phi_\star)$ is full rank, then $\phi_\star$ satisfies the KKT conditions associated to Eq. 45.*

*Proof.* We use the shorthand $E := \frac{1}{2}\|f\|^2$ for conciseness. For every $\beta$, $\phi_*^\beta$ verifies

$$\frac{\partial F}{\partial \phi}(\phi_*^\beta, \theta, \beta) = \frac{\partial E}{\partial \phi}(\phi_*^\beta, \theta) + \beta \frac{\partial L}{\partial y}(h(\phi_*^\beta)) \frac{\partial h}{\partial \phi}(\phi_*^\beta) = 0 \tag{49}$$

By taking $\beta$ to infinity in the equation above and using continuity of $\partial_\phi E$, we obtain

$$\frac{\partial E}{\partial \phi}(\phi_\star) = -\lim_{\beta \to \infty} \beta \frac{\partial L}{\partial y}(h(\phi_*^\beta)) \frac{\partial h}{\partial \phi}(\phi_*^\beta). \tag{50}$$

This first implies that

$$\frac{\partial L}{\partial y}(h(\phi_\star)) \frac{\partial h}{\partial \phi}(\phi_\star) = \lim_{\beta \to \infty} \frac{\partial L}{\partial y}(h(\phi_*^\beta)) \frac{\partial h}{\partial \phi}(\phi_*^\beta) = 0 \tag{51}$$

as $\partial_\phi h$, $\partial_y L$ and $h$ are continuous, and then that $\nabla_y L(h(\phi_\star)) = 0$ as $\partial_\phi h(\phi_\star)$ is full rank.

Eq. 50, along with $\partial_\phi h(\phi_\star)$ being full rank, implies that $\beta \partial_y L(h(\phi_*^\beta))$ admits a finite limit, note it $\mu$. As $\partial_y^2 L(h(\phi_\star))$ is invertible, there exists $\mu_\star$ such that $\partial_y^2 L(\phi_\star)\mu_\star = \mu$. Put back into (50) it yields

$$\frac{\partial E}{\partial \phi}(\phi_\star) + \mu_\star^\top \frac{\partial^2 L}{\partial y^2}(h(\phi_\star)) \frac{\partial h}{\partial \phi}(\phi_\star) = 0, \tag{52}$$

hence $\phi_\star$ satisfies the KKT conditions associated to (45). $\qquad\square$

**Proposition S7.** *Assume $L$ to positive and equal to 0 if and only if $\nabla_y L = 0$. Let $\phi_*^\beta$ be a global minimizer of the augmented energy $F$ for every $\beta$, such that $\phi_*^\beta$ admits a finite limit $\phi_\star$ when $\beta$ goes to infinity. Then the limit $\phi_\star$ is a global minimizer of the constrained optimization problem (45).*

*Proof.* First, remark that the condition on the minimizers of $L$ implies that solving

$$\min_\phi \frac{1}{2} \|f(\phi, \theta)\|^2 \quad \text{s.t. } \nabla_y L(h(\phi)) = 0 \tag{53}$$

is equivalent to solving

$$\min_\phi \frac{1}{2} \|f(\phi, \theta)\|^2 \quad \text{s.t. } L(h(\phi)) = 0. \tag{54}$$

We use the second characterization in the rest of the proof.

We introduce the short hand

$$f_\star := \min_\phi \frac{1}{2} \|f(\phi, \theta)\|^2 \quad \text{s.t. } L(h(\phi)) = 0. \tag{55}$$

For any $\phi$ we have that

$$F(\phi_*^\beta, \theta, \beta) \leq F(\phi, \theta, \beta) \tag{56}$$

by definition of $\phi_*^\beta$, in particular for all the $\phi$ verifying the constraint $L(h(\phi)) = 0$. It follows that

$$F(\phi_*^\beta, \theta, \beta) \leq \min_{L(h(\phi))=0} F(\phi, \theta, \beta) = \min_{L(h(\phi))=0} \frac{1}{2} \|f(\phi, \theta)\|^2 = f_\star. \tag{57}$$

The last inequality can be rewritten as

$$L(h(\phi_*^\beta)) \leq \frac{f_\star - \|f(\phi_*^\beta, \theta)\|^2/2}{\beta}.$$

The upper bound converges to 0 as the numerator converges to a finite value by continuity of $\|f\|^2$. Since $L$ is positive and continuous and $h$ is continuous we obtain $L(h(\phi_\star)) = 0$, i.e., $\phi_\star$ is feasible.

We now show that $\|f(\phi_\star, \theta)\|^2/2 = f_\star$. As $\phi_\star$ is feasible we have

$$f_\star = \min_{L(h(\phi))=0} \frac{1}{2} \|f(\phi, \theta)\|^2 \leq \frac{1}{2} \|f(\phi_\star, \theta)\|^2. \tag{58}$$

This implies that

$$\limsup_{\beta\to\infty} F(\phi_*^\beta, \theta, \beta) = \frac{1}{2} \|f(\phi_\star, \theta)\|^2 + \limsup_{\beta\to\infty} \beta L(h(\phi_*^\beta)) \geq f_\star \tag{59}$$

where we used the positivity of $L$ for the last inequality. Because of (57), we also have

$$\limsup_{\beta\to\infty} F(\phi_*^\beta, \beta) \leq f_\star \tag{60}$$

so that the combination of the last two equations yield

$$\limsup_{\beta\to\infty} \beta L(h(\phi_*^\beta)) = 0 \tag{61}$$

and

$$\frac{1}{2} \|f(\phi_\star, \theta)\|^2 = f_\star. \tag{62}$$

This finishes the proof. $\qquad\square$

*Remark* S2. Under the same assumptions, we can actually show that, for any $\beta, \beta' \in \mathbb{R}_+$, we have

$$0 \leq F(\phi_*^\beta, \theta, \beta) \leq F(\phi_*^{\beta'}, \theta, \beta') \leq \frac{1}{2} \|f(\phi_\star, \theta)\|^2. \tag{63}$$

This can be obtained by combining Proposition S7 and the fact that $\beta \mapsto F(\phi_*^\beta, \theta, \beta)$ is a non decreasing function as

$$\begin{aligned}
\frac{\mathrm{d}}{\mathrm{d}\beta} F(\phi_*^\beta, \theta, \beta) &= \frac{\partial F}{\partial \beta}(\phi_*^\beta, \theta, \beta) + \frac{\partial F}{\partial \phi}(\phi_*^\beta, \theta, \beta) \frac{\mathrm{d}\phi_*^\beta}{\mathrm{d}\beta} \\
&= L(\phi_*^\beta) + 0 \\
&\geq 0.
\end{aligned} \tag{64}$$

The function $\beta \mapsto F(\phi_*^\beta, \theta, \beta)$ therefore converges to the least-control objective $\mathcal{H}(\theta)$ from below.

In the energy-based view of the least-control principle, the least-control objective becomes

$$\mathcal{H}(\theta) = \frac{1}{2} \|f(\phi_\star, \theta)\|^2. \tag{65}$$

We use this formulation to provide an alternative proof to Theorem S1 that might be insightful for the reader.

*Alternative proof of Theorem S1.* Theorem S1 states that

$$\left(\frac{\mathrm{d}}{\mathrm{d}\theta}\mathcal{H}(\theta)\right)^{\top} = \frac{\partial f}{\partial \theta}(\phi_\star, \theta)^{\top} f(\phi_\star, \theta). \tag{66}$$

We use the shorthand notation $E := \frac{1}{2}\|f(\phi, \theta)\|^2$. The chain rule yields

$$\frac{\mathrm{d}}{\mathrm{d}\theta}E(\phi_\star, \theta) = \frac{\mathrm{d}}{\mathrm{d}\theta}E(\phi_\star, \theta) = \frac{\partial E}{\partial \theta}(\phi_\star, \theta) + \frac{\partial E}{\partial \phi}(\phi_\star, \theta)\frac{\mathrm{d}\phi_\star}{\mathrm{d}\theta}. \tag{67}$$

As $\nabla_\phi L$ is independent of $\theta$, we have that $\mathrm{d}_\theta h(\phi_\star) = 0$. This rewrites $\partial_\phi h(\phi_\star)\mathrm{d}_\theta \phi_\star = 0$. This can be combined to the KKT conditions to show that the indirect term is equal to 0: right multiplying the condition

$$0 = \frac{\partial E}{\partial \phi}(\phi_\star, \theta) + \mu_\star^{\top}\frac{\partial^2 L}{\partial y}(h(\phi_\star))\frac{\partial h}{\partial \phi}(\phi_\star) \tag{68}$$

by $\mathrm{d}_\theta \phi_\star$ gives

$$\begin{aligned}
0 &= \frac{\partial E}{\partial \phi}(\phi_\star, \theta)\frac{\mathrm{d}\phi_\star}{\mathrm{d}\theta} + \mu_\star^{\top}\frac{\partial^2 L}{\partial y}(h(\phi_\star))\frac{\partial h}{\partial \phi}(\phi_\star)\frac{\mathrm{d}\phi_\star}{\mathrm{d}\theta} \\
&= \frac{\partial E}{\partial \phi}(\phi_\star, \theta)\frac{\mathrm{d}\phi_\star}{\mathrm{d}\theta}.
\end{aligned} \tag{69}$$

This finishes the proof. $\qquad\square$

### S2.6 The parameter update is robust to approximate optimal control

The purpose of this section is to formally prove that the first-order update prescribed by Theorem 1 is robust to inaccurate approximations of the optimally-controlled state $\phi_\star$. More precisely, we show that the quality of the gradient can be linked to the distance between the approximate and the true optimally-controlled state (Proposition S8) and that the update is minimizing a closely related objective when the perfect control limit is not exactly reached (Proposition S9).

**Proposition S8.** *Let $\phi_*$ be an optimally-controlled state and $\hat{\phi}$ an approximation of it that is used to update the parameters $\theta$. Then, if the conditions of Theorem S1 apply and $\partial_\theta f(\phi, \theta)^{\top} f(\phi, \theta)$ is $M$-Lipschitz continuous we have that*

$$\left\|f(\hat{\phi}, \theta)^{\top}\frac{\partial f}{\partial \theta}(\hat{\phi}, \theta) - \frac{\mathrm{d}}{\mathrm{d}\theta}\mathcal{H}(\theta)\right\| \leq M\|\hat{\phi} - \phi_*\|. \tag{70}$$

*Proof.* We use Theorem S1 and then the Lipschitz continuity assumption:

$$\begin{aligned}
\left\|f(\hat{\phi}, \theta)^{\top}\frac{\partial f}{\partial \theta}(\hat{\phi}, \theta) - \frac{\mathrm{d}}{\mathrm{d}\theta}\mathcal{H}(\theta)\right\| &= \left\|f(\hat{\phi}, \theta)^{\top}\frac{\partial f}{\partial \theta}(\hat{\phi}, \theta) - f(\phi_\star, \theta)^{\top}\frac{\partial f}{\partial \theta}(\phi_\star, \theta)\right\| \\
&\leq M\|\hat{\phi} - \phi_\star\|.
\end{aligned} \tag{71}$$

$\square$

We empirically verify that the behavior predicted by Proposition S8 holds in practice. To do so, we compare the estimated gradient $f(\hat{\phi}, \theta)^{\top}\partial_\theta f(\hat{\phi}, \theta)$ for several approximations $\hat{\phi}$ of the optimally-controlled state $\phi_*$ to the true least-control gradient $\mathrm{d}_\theta \mathcal{H}(\theta)$ on a feedforward neural network learning problem. Figure S1 shows the gradient estimation error as a function of the steady state approximation $\|\hat{\phi} - \phi_*\|$. The evolution is almost linear, suggesting that the inequality in Eq. 70 is close to being an equality for some constant $M$.

**Proposition S9.** *Let $(\phi_*, u_*)$ verifying the steady-state equations*

$$f(\phi_*, \theta) + Q(\phi_*, \theta)u_* = 0, \quad \text{and} \quad -\alpha u_* - \nabla_y L(h(\phi_*)) = 0. \tag{72}$$

*If*

$$Q(\phi_*, \beta) = \partial_\phi f(\phi_*, \theta)^{-\top}\partial_\phi h(\phi_*)^{\top}, \tag{73}$$

*then $\phi_*$ is a stationary point of the augmented energy $F(\phi, \theta, \alpha) = \frac{1}{2}\|f(\phi, \theta)\|^2 + \alpha^{-1}L(h(\phi))$.*

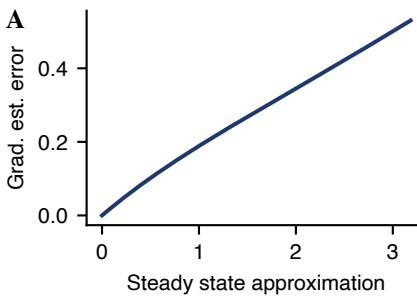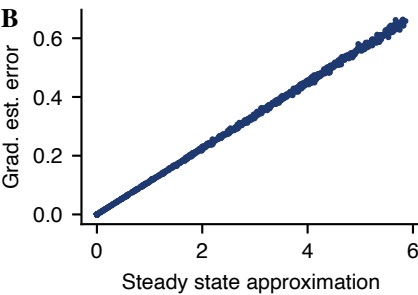

Figure S1: Empirical verification of Proposition S8 on a randomly initalized feedforward neural network with 3 hidden layers of size 300. The different gradients are estimated using 100 data samples randomly drawn from the MNIST dataset. (A) We run energy-based dynamics for $\alpha = 0$ with a time constant of 200 for 200 steps, compute the gradient estimation error at each step, and plot it as a function of the steady state approximation error $\|\hat{\phi} - \phi_*\|$. The optimally-controlled $\phi_*$ is computed by running the same dynamics until convergence. (B) After computing $\phi_*$, we perturb it with random noise drawn from $\mathcal{U}([0,1]) \times \mathcal{N}(0, 0.5)$, and plot the same relationship as in panel A.

*Proof.* We first use the formula for $Q$ and use the steady-state equation on $u_*$:

$$
\begin{aligned}
0 &= f(\phi_*, \theta) + Q(\phi_*, \theta)u_* \\
&= f(\phi_*, \theta) - \partial_\phi f(\phi_*, \theta)^{-\top} \partial_\phi h(\phi_*)^\top u_* \\
&= f(\phi_*, \theta) + \alpha^{-1} \partial_\phi f(\phi_*, \theta)^{-\top} \partial_\phi h(\phi_*)^\top \nabla_y L(h(\phi_*))
\end{aligned}
\tag{74}
$$

Left-multiplying by $\partial_\phi f(\phi_*, \theta)$ yields

$$
\partial_\phi f(\phi_*, \theta)^\top f(\phi_*, \theta) + \alpha^{-1} \partial_\phi h(\phi_*)^\top \nabla_y L(h(\phi_*)) = 0,
\tag{75}
$$

which is exactly $\nabla_\phi F(\phi, \theta, \alpha) = 0$. $\qquad\square$

Proposition S9 allows understanding what the update $\partial_\theta f(\phi_*, \theta)^\top f(\phi_*, \theta)$ is doing when the controlled dynamics reach an equilibrium, but $\alpha$ is non zero. In this case, we cannot guarantee that $Q(\phi_*, \theta)u_*$ is an optimal control. Still, when the strict condition on $Q$ of Eq. 73 is satisfied (it is strict compared to the more general condition (35) of Theorem S2), the update prescribed by the least-control principle is minimizing $F(\phi_*, \theta, \alpha)$ as

$$
\begin{aligned}
\frac{\mathrm{d}}{\mathrm{d}\theta} F(\phi_*, \theta, \alpha) &= \frac{\partial F}{\partial \theta}(\phi_*, \theta, \alpha) + \frac{\partial F}{\partial \phi}(\phi_*, \theta, \alpha) \frac{\mathrm{d}\phi_*}{\mathrm{d}\theta} \\
&= \frac{\partial F}{\partial \theta}(\phi_*, \theta, \alpha) + 0 \\
&= \frac{\partial f}{\partial \theta}(\phi_*, \theta)^\top f(\phi_*, \theta).
\end{aligned}
\tag{76}
$$

We showed in Section S2.5 that this objective is closely related to the least-control objective, and gets closer when $\alpha \to 0_+$. Note that the same conclusions apply if we use a proportional controller with strength $\beta$, as it verifies the steady-state equation (72) for $\alpha = \beta^{-1}$.

## S2.7 Quick review of constrained optimization

We heavily rely on the Karush–Kuhn–Tucker (KKT) conditions and Lagrange multiplier method for characterizing constrained optimum. We state here some important properties that are throughout the proofs, and we refer the interested reader to [93, 94] for a comprehensive overview of the relevant theory.

**KKT conditions.** Consider the following equality-constrained optimization problem

$$
\min_x f(x) \ \text{ s.t. } g(x) = 0,
\tag{77}
$$

with $f$ some scalar function to be optimized and $g$ a vector-valued function defining equality constraints on the set of feasible states. The corresponding Lagrangian is defined as

$$\mathcal{L}(x, \lambda) := f(x) + \lambda^\top g(x), \tag{78}$$

with $\lambda$ the so-called Lagrange multipliers. It can be shown, e.g. Nocedal and Wright [93, Theorem 12.1], that local solutions $x_\star$ of the constrained optimization problem defined above satisfy the KKT conditions: there exists a Lagrange multiplier $\lambda_\star$ such that

$$\begin{cases} 0 = \dfrac{\partial f}{\partial x}(x_\star) + \dfrac{\partial g}{\partial x}(x_\star)\lambda_\star \\ 0 = g(x) \end{cases} \tag{79}$$

or equivalently such that $(x_\star, \lambda_\star)$ is a stationary point of the Lagrangian, i.e. $\partial_{x_\star, \lambda_\star} \mathcal{L}(x_\star, \lambda_\star) = 0$.

The KKT conditions are first-order stationary conditions for constrained optimization, in the same way $\partial_x f = 0$ is a stationary condition for unconstrained minimization. On the other side, there exists sufficient conditions under which those stationary conditions give a local, constrained or unconstrained, minimum. For unconstrained optimization, it can be that the Hessian $\nabla_x^2 f$ is positive definite. For constrained optimization there are several variants, but those conditions can more or less be understood as the Hessian $\partial_x^2 \mathcal{L}(x_\star, \lambda_\star)$ of the Lagrangian w.r.t. $x$ is positive definite (see e.g. Nocedal and Wright [93, Theorem 12.5]).

**Differentiating through minima.** We now consider that both the objective $f$ and the constraints $g$ are dependent on some parameter $y$, and aim to calculate how $f(x_\star(y), y)$ will react to a change in $y$, that is we aim to compute the gradient associated to

$$\min_y f(x_\star(y), y) = \min_y \min_x f(x, y) \text{ s.t. } g(x, y) = 0. \tag{80}$$

For *unconstrained* bilevel optimization (i.e. without the equality constraints $g(x, y) = 0$), we differentiate through the inner minima $x_\star(y)$ using the necessary stationary condition $\partial_x f(x_\star(y), y) = 0$, combined with the implicit function theorem [89] to ensure that the implicit function $x_\star(y)$ is well defined. This gives

$$\begin{aligned} \frac{\mathrm{d}}{\mathrm{d}y} f(x_\star(y), y) &= \frac{\partial f}{\partial y}(x_\star(y), y) + \frac{\partial f}{\partial x}(x_\star(y), y)\frac{\mathrm{d}x_\star(y)}{\mathrm{d}y} \\ &= \frac{\partial f}{\partial y}(x_\star(y), y) + 0. \end{aligned} \tag{81}$$

Note that $\mathrm{d}_y x_\star(y)$ can be obtained via the implicit function theorem by using the first-order condition $\partial_x f(x_\star(y), y) = 0$ but we do not need to compute it.

A similar technique can be applied to differentiate through a *constrained* minimum. We first remark that $x_\star$ satisfies the KKT conditions so there exists a Lagrange multiplier $\lambda_\star(y)$ such that

$$f(x_\star(y), y) = \mathcal{L}(x_\star(y), \lambda_\star(y), y). \tag{82}$$

We then use the implicit function theorem on the KKT condition $\partial_{x,\lambda} \mathcal{L}(x_\star, \lambda_\star, y) = 0$ to show that the functions $x_\star(y)$ and $\lambda_\star(y)$ are well defined and differentiable, under the assumption that the Hessian of the Lagrangian w.r.t. $x$ and $\lambda$ is invertible. Finally, we have:

$$\begin{aligned} \frac{\mathrm{d}}{\mathrm{d}y} f(x_\star(y), y) &= \frac{\mathrm{d}}{\mathrm{d}y} \mathcal{L}(x_\star(y), \lambda_\star(y), y) \\ &= \frac{\partial \mathcal{L}}{\partial y} + \frac{\partial \mathcal{L}}{\partial x}\frac{\mathrm{d}x_\star}{\mathrm{d}y} + \frac{\partial \mathcal{L}}{\partial \lambda}\frac{\mathrm{d}\lambda_\star}{\mathrm{d}y} \\ &= \frac{\partial \mathcal{L}}{\partial y} + 0 + 0 \end{aligned} \tag{83}$$

where the last inequality holds as $x_\star(y)$ and $\lambda_\star(y)$ are stationary points of the Lagrangian $\mathcal{L}$.

## S3 Implicit gradient, recurrent backpropagation and the link to the least-control principle

The least-control principle aims at solving the learning problem

$$\min_{\theta} L(h(\phi^*)) \quad \text{s.t.} \ f(\phi^*, \theta) = 0 \tag{84}$$

indirectly, by minimizing the least-control objective. This leads to first-order updates, contrary to the direct minimization of the loss $L(\phi^*)$. In this section, we derive the implicit gradient associated to direct loss minimization, show how the recurrent backpropagation computes it, and finally highlight the similarities behind the least control principle and recurrent backpropagation.

**Implicit gradient.** The implicit gradient $\mathrm{d}_{\theta} L(\phi^*)$ can be calculated analytically using the implicit function theorem [89], we here calculate it. The quantity $\phi^*$ can be characterized as an implicit function of $\theta$, which verifies

$$f(\phi^*, \theta) = 0 \tag{85}$$

for all $\theta$. The implicit function theorem guarantees that $\phi^*$ is properly defined as an implicit function of $\theta$ if $\partial_{\phi} f(\phi^*, \theta)$ is invertible, and then

$$\frac{\mathrm{d}\phi^*}{\mathrm{d}\theta} = -\left(\frac{\partial f}{\partial \phi}(\phi^*, \theta)\right)^{-1} \frac{\partial f}{\partial \theta}(\phi^*, \theta).$$

We can then use the chain rule to obtain the implicit gradient:

$$\begin{aligned}
\frac{\mathrm{d}}{\mathrm{d}\theta} L(h(\phi^*)) &= \frac{\partial L}{\partial y}(h(\phi^*)) \frac{\partial h}{\partial \phi}(\phi^*) \frac{\mathrm{d}\phi^*}{\mathrm{d}\theta} \\
&= -\frac{\partial L}{\partial y}(\phi^*) \frac{\partial h}{\partial \phi}(h(\phi^*)) \left(\frac{\partial f}{\partial \phi}(\phi^*, \theta)\right)^{-1} \frac{\partial f}{\partial \theta}(\phi^*, \theta).
\end{aligned} \tag{86}$$

**Recurrent backpropagation.** The implicit gradient in computationally intractable for most of practical applications due to the matrix inversion. The key idea behind the recurrent backpropagation algorithm is that the column vector

$$\delta^* := \left(\frac{\partial f}{\partial \phi}(\phi^*, \theta)\right)^{-\top} \frac{\partial h}{\partial \phi}(\phi^*)^{\top} \frac{\partial L}{\partial y}(h(\phi^*))^{\top} \tag{87}$$

is the steady state of the dynamics

$$\dot{\delta} = \left(\frac{\partial f}{\partial \phi}(\phi^*, \theta)\right)^{\top} \delta - \frac{\partial h}{\partial \phi}(\phi^*)^{\top} \frac{\partial L}{\partial y}(h(\phi^*))^{\top} \tag{88}$$

or alternatively the fixed point of the iterative procedure

$$\delta_{\text{next}} = \delta + \left(\frac{\partial f}{\partial \phi}(\phi^*, \theta)\right)^{\top} \delta - \frac{\partial h}{\partial \phi}(\phi^*)^{\top} \frac{\partial L}{\partial y}(h(\phi^*))^{\top}. \tag{89}$$

Once $\delta^*$, or its estimation, is obtained through this dynamical procedure, it can then be used to estimate the implicit gradient with

$$\frac{\mathrm{d}}{\mathrm{d}\theta} L(\phi^*) = -\delta^{*\top} \frac{\partial f}{\partial \theta}(\phi^*, \theta) \tag{90}$$

**Backpropagation.** When the computational graph encoded in $f$ is acyclic, that is we can find a permutation of the state such that $f(\phi, \theta)_i$ only depends on the states $\phi_j$ for $j \leq i$, the computation of $\delta^*$ can be simplified. Indeed, instead of running the dynamics of Eq. 88, one can directly compute $\delta^*$ through the fixed-point equation

$$\left(\frac{\partial f}{\partial \phi}(\phi^*, \theta)\right)^{\top} \delta = \frac{\partial h}{\partial \phi}(\phi^*)^{\top} \frac{\partial L}{\partial y}(h(\phi^*))^{\top}. \tag{91}$$

as the Jacobian $\partial_{\phi} f(\phi^*, \theta)$ is a lower triangular matrix. This is for example how $\delta^*$ is calculated for feedforward neural networks. In such settings, we can therefore interpret the dynamics (88) to be the continuous time version of the backpropagation algorithm.

**Comparison with the least-control principle.** We now compare the implicit gradient, as computed with recurrent backpropagation, with the gradient of the least-control objective, which is equal to

$$\frac{\mathrm{d}\mathcal{H}}{\mathrm{d}\theta}(\theta) = -\psi_\star^\top \frac{\partial f}{\partial \theta}(\phi_\star, \theta) \tag{92}$$

as given by Theorem 1. In addition to the close resemblance between the gradients (90) and (92), the dynamical equations on $\delta$ and $\psi$ share some interesting similarities. Let us take the example of the inversion dynamics

$$\dot{\phi} = f(\phi, \theta) + \psi$$
$$\dot{\psi} = \left(\frac{\partial f}{\partial \phi}(\phi, \theta)\right)^\top \psi + \frac{\partial h}{\partial \phi}(\phi)^\top u \tag{93}$$

of Eq. 8. The $\psi$ dynamics are very similar to the one of $\delta$: they can be understood as multiplying the inverse of the Jacobian of $f$ with an error minimization signal, that is either $-\partial_y L$ or $u$, to yield $\delta^*$ or $\psi_\star$. The important conceptual difference is that the controlled dynamics of the least-control principle simultaneously compute both the steady state $\phi_*$ and optimal control $\psi_*$ in a single phase, whereas recurrent backpropagation uses two separate phases for computing $\phi^*$ and $\delta^*$. This unique feature of the least-control principle is made possible by letting the control $\psi$ influence the dynamics and hence the steady state of $\phi$.

## S4  On the relation between the least-control principle, free energy and predictive coding

Predictive coding [36, 37, 56, 58][95] emerged as an important framework to study perception and sensory processing in the brain. In this framework, the brain is assumed to maintain a probabilistic model of the environment, which is constantly used to perform inference on its sensory inputs. Here, we expand in more detail on the relation between our least-control principle and predictive coding. More specifically, we discuss how the constrained energy minimization of Eq. 9 relates to the free energy used in recent work on supervised predictive coding [36], and how the new insights of the least-control principle, such as more flexible dynamics, translate to the predictive coding framework.

### S4.1  Supervised predictive coding

The starting point of standard predictive coding theories is the specification of a probabilistic generative model from latent causes towards sensory inputs; this model is then used to infer the most likely causes underlying incoming sensory stimuli. Recent work [36] investigated supervised predictive coding, a form of predictive coding where the direction of causality is flipped, i.e., where one uses a discriminative model from sensory inputs towards latent causes, and trains this model by clamping its outputs to the correct teacher value and performs inference to propagate teaching signals throughout the network. Whittington and Bogacz [36] related the parameter updates resulting from supervised predictive coding network to error backpropagation [31, 32] when the output is weakly nudged towards the teacher value instead of being clamped. In Section S4.3 we show that the least-control principle relates the supervised predictive coding updates with standard teacher clamping to gradient updates on the least-control objective (3) thereby removing the need for the weak-nudging limit.

Supervised predictive coding uses a probabilistic latent variable model to compute the likelihood $p(y \mid x; \theta)$ of the output $y$ given the sensory input $x$. More specifically, it uses a hierarchical Gaussian model, a particular type of directed acyclic graphical model, with the following factorized distribution

$$
\begin{aligned}
p(z_1, \ldots, z_{L-1}, y \mid x; \theta) &= p(y \mid z_{L-1}) \prod_{l=1}^{L-1} p(\phi_l \mid z_{l-1}) \\
p(z_l \mid \phi_{l-1}) &= \mathcal{N}(z_l;\ W_l \sigma(z_{l-1}), \Sigma_l)) \\
p(y \mid z_{L-1}) &= \mathcal{N}(y;\ W_L \sigma(z_{L-1}, \Sigma_y))
\end{aligned}
\tag{94}
$$

with latent variables $z = \{z_l\}_{l=1}^{L-1}$, parameters $\theta = \{W_l\}_{l=1}^{L}$, $\mathcal{N}(v; \mu, \Sigma)$ a Gaussian distribution over a variable $v$ with mean $\mu$ and correlation matrix $\Sigma$, and where we take $\sigma(z_0) = x$. The goal of training this model is to change the model parameters $W_l$ to maximize the marginal likelihood for input-output pairs $(x, y)$:

$$
p(y \mid x; \theta) = \int p(z, y \mid x; \theta) \mathrm{d}z
\tag{95}
$$

As optimizing this marginal directly is intractable, predictive coding uses the variational expectation maximization method [95–98]. This method uses a variational distribution $q(z; \phi)$ with parameters $\phi$ to approximate the posterior $p(z \mid x, y)$ and then defines the free energy $\mathcal{F}$ as an upper bound on the negative log-likelihood:

$$
\begin{aligned}
\mathcal{F}(\phi, \theta) &= -\mathbb{E}_q[\log p(y, z \mid x; \theta)] + \mathbb{E}_q[\log q(z; \phi)] \\
&= -\log p(y \mid x; \theta) + D_{\mathrm{KL}}(q(z; \phi) \| p(z \mid x, y; \theta)) \geq -\log p(y \mid x; \theta),
\end{aligned}
\tag{96}
$$

with $D_{\mathrm{KL}}$ the KL divergence which is always positive. Now the variational expectation maximization method proceeds in two alternating phases. In the expectation phase, it minimizes the free energy $\mathcal{F}$ w.r.t. $\phi$ for the current parameter setting of $\theta$, i.e. it finds a good approximation $q(z; \phi)$ for the posterior. In the maximization phase, it minimizes the free energy w.r.t. $\theta$ for the updated parameter setting of $\phi$.

To be able to perform the two phases using simple neural dynamics and local update rules, predictive coding uses the Dirac delta distribution $q(z; \phi) = \delta(z - \phi)$ as the variational distribution [98]. Now, the entropy term $\mathbb{E}_q[\log p(y, z \mid x; \theta)]$ is independent of $\phi$ and hence we ignore it in the expectation

and maximization step.[4] Taken together, this result in the following free energy (without entropy term):

$$\mathcal{F}(\phi, \theta) = -\log p(y, \phi \mid x; \theta)$$
$$= \rho \frac{1}{2} \|y - W\sigma(\phi_{L-1})\|^2 + \frac{1}{2} \sum_{l=1}^{L-1} \|\phi_l - W_l\sigma(\phi_{l-1})\|^2 + C \quad (97)$$

with $C$ a constant independent of $\phi$ and $\theta$, $x = \sigma(\phi_0)$, and where we took for simplicity $\Sigma_l = \mathrm{Id}$ and $\Sigma_y = \rho^{-1}\mathrm{Id}$. For the expectation step, predictive coding interprets the parameters $\phi = \{\phi_l\}_{l=1}^{L-1}$ as a prediction neuron population and uses the gradient flow on $\mathcal{F}$ w.r.t. $\phi$ as neural dynamics to find a $\phi_*$ that minimizes $\mathcal{F}$. For the maximization step, predictive coding performs a gradient step on $\mathcal{F}(\phi_*, \theta)$ w.r.t. $\theta$, evaluated at the steady-state neural activity $\phi_*$. Thus,

$$\dot{\phi} = -\frac{\partial \mathcal{F}}{\partial \phi}(\phi, \theta)^\top,$$
$$\Delta\theta = -\frac{\partial \mathcal{F}}{\partial \theta}(\phi_*, \theta)^\top, \quad (98)$$

with the change $\Delta\theta$ applied after the neural dynamics has converged, i.e., when $\dot{\phi} = 0$.

### S4.2 Equivalence between the free and augmented energies

To link supervised predictive coding to the least-control principle, we further manipulate the free energy of Eq. 97 to relate it to the augmented energy $F = \frac{1}{2}\|f(\phi, \theta)\|^2 + \beta L$ introduced in Section 2.4. For this, we introduce a 'ghost layer' $z_L$ in between $z_{L-1}$ and $y$, that exists solely for the purpose of analyzing the free energy. The joint probability distribution is now given by

$$p(z, y \mid x; \theta) = p(y \mid z_L) \prod_{l=1}^{L} p(z_l \mid z_{l-1})$$
$$p(z_l \mid z_{l-1}) = \mathcal{N}(z_l; \ W_l\sigma(z_{l-1}), \mathrm{Id}) \quad (99)$$
$$p(y \mid z_L) = \mathcal{N}(y; \ z_L, \beta^{-1}\mathrm{Id})$$

We recover the same marginal likelihood $p(y \mid x; \theta)$ as before if we have that $\rho^{-1} = \beta^{-1} + 1$, assuming that $\rho^{-1} > 1$. Using again the dirac delta distribution as variational distribution, we get the following augmented free energy

$$\mathcal{F}(\phi, \theta, \beta) = \log p(\phi, y \mid x; \theta) = \frac{1}{2} \sum_{l=1}^{L} \|\phi_l - W_l\sigma(\phi_{l-1})\|^2 + \beta \frac{1}{2}\|y - \phi_L\|^2$$
$$= \frac{1}{2}\|\phi - W\sigma(\phi) - Ux\|^2 + \beta L(D\phi) \quad (100)$$
$$W = \begin{bmatrix} 0 & 0 & 0 & 0 \\ W_2 & 0 & 0 & 0 \\ 0 & \ddots & 0 & 0 \\ 0 & 0 & W_L & 0 \end{bmatrix}, \quad U = \begin{bmatrix} W_1 \\ 0 \end{bmatrix}, \quad D = [0 \quad \mathrm{Id}],$$

with $\phi$ the concatenation of $\{\phi_l\}_{l=1}^{L}$, $L$ the squared error loss and where we ignored the constant term $C$. Hence, we see that the free energy used in predictive coding is equivalent to the augmented energy $F = \frac{1}{2}\|f(\phi, \theta)\|^2 + \beta L$ introduced in Section 2.4. Consequently, we can use the least-control principle to characterize predictive coding in the limit of $\beta \to \infty$.

---

[4]Note however that this entropy term is infinite for the delta distribution, hence for making the derivation rigorous, certain limits need to be taken. For example, Friston et al. [95] uses a Laplace approximation instead of the delta distribution.

### S4.3 Learning with teacher clamping results in gradients on the least-control objective

Whittington and Bogacz [36] connected supervised predictive coding to gradient-based optimization of the loss $L$, when the variance of the output layer $y$ goes to infinity. In this limit, there is only an infinitesimal effect of the teaching signal on the rest of the network, arising by clamping $y$ towards $y^{\text{true}}$, as the model 'does not trust' the output layer $y$ by construction. This *weak nudging* limit is captured in the augmented energy of Eq. (100) as the limit of $\beta \to 0$, and can be linked to gradient-based optimization of the loss $L$ for more general energy functions beyond predictive coding [24].

Equipped with the least-control principle, we can now relate predictive coding to gradient-based optimization on the least-control objective of Eq. 3, without the need for infinite variance in the output layer. In Sections 2.4 and S2.5, we show that when the neural dynamics minimize the augmented energy $F(\phi, \theta, \beta)$ for $\beta \to \infty$, the parameter update $\Delta\theta$ of Eq. 98 follows the gradient on the least-control objective (3). This perfect control limit of $\beta \to \infty$ corresponds to the conventional teacher clamping setting, where the output $y$ has a finite variance (without loss of generality, $\rho = 1$ in this case in our derivation, as $\rho^{-1} = \beta^{-1} + 1$).

### S4.4 The optimal control interpretation of predictive coding leads to flexible dynamics.

Now that we connected predictive coding to the least-control principle, we can further investigate predictive coding through the lens of optimal control. In predictive coding, the neural dynamics follow the gradient flow on the augmented free energy $\mathcal{F}$ of Eq. (100):

$$
\begin{aligned}
\dot{\phi} &= -\frac{\partial\mathcal{F}}{\partial\phi}(\phi, \theta, \beta)^\top \\
&= -\phi + W\sigma(\phi) + Ux + \sigma'(\phi)W^\top\big(\phi - W\sigma(\phi) - Ux\big) - \beta D^\top\nabla_y L(D\phi)
\end{aligned}
\tag{101}
$$

with $\sigma'(\phi) = \partial_\phi\sigma(\phi)$. By comparing the above equation to $\dot{\phi} = f(\phi, \theta) + \psi$, we have the optimal control $\psi = \sigma'(\phi)W^\top\big(\phi - W\sigma(\phi) - Ux\big) - \beta D^\top\nabla_y L(D\phi)$ For $\beta \to \infty$, the last term $\beta D^\top\nabla_y L(D\phi)$ corresponds to an infinitely fast proportional control that clamps the output $y = \phi_L$ to the teacher $y^{\text{true}}$. The term $\sigma'(\phi)W^\top\big(\phi - W\sigma(\phi) - Ux\big)$ then optimally propagates the control at the output level towards the rest of the network. In predictive coding, the term $\phi - W\sigma(\phi) - Ux$ is usually interpreted as an error neuron population $\epsilon$. Hence, in the optimal control view, the error neuron population $\epsilon$ optimally controls the prediction neuron population $\phi$ via $\psi = \sigma'(\phi)W^\top\epsilon - \beta D^\top\nabla_y L(D\phi)$, to reach a controlled network state that exactly matches the output target $y^{\text{true}}$, while having the smallest possible error signals $\|\epsilon\|^2 = \|\phi - W\sigma(\phi) - Ux\|^2$. Note that at equilibrium, we have that $\psi = -f(\phi, \theta) = \epsilon$. This optimal control view is further corroborated by isolating the error neurons $\epsilon$ at the equilibrium of the neural dynamics (101), leading to $\epsilon_* = \psi_* = -(I - W\sigma'(\phi_*))^{-\top}D^\top\beta\nabla_y L(D\phi_*)$, which satisfies the column space condition of Theorem 2 with $u_* = -\beta\nabla_y L(D\phi_*)$, confirming that the error neurons steady state $\psi_*$ is an optimal control.

Importantly, using the least-control principle, we can go beyond the gradient-flow dynamics on $F$ and generalize the predictive coding framework to more flexible neural dynamics. First, instead of clamping the output of the network towards the teacher value with an infinitely fast proportional controller, one can use more general output controllers that satisfy the equilibrium condition $\alpha u_* + \nabla_y L(D\phi) = 0$ for $\alpha \to 0$, such as an integral controller. Next, as we identified the error neurons $\epsilon = \phi - W\sigma(\phi) - Ux$ as being an optimal control $\epsilon_* = \psi_\star$ at equilibrium, we can use any dynamics satisfying Theorem 2 to compute these optimal error neurons and resulting neural dynamics $\dot{\phi}$. For example, we can use the inversion dynamcis of Eq. 8 and 14 to dynamically compute the error neuron activity. Strikingly, at the steady state, the error neurons computed with these inversion dynamics are indistinguishable from the error neurons computed with the energy-based dynamics of Eq. 101, even though the underlying dynamics and hence circuit implementation are completely different. This 'circuit invariance' under the optimality condition of Theorem 2 opens new routes towards finding cortical circuits implementing predictive coding, which is a topic of high relevance in neuroscience [58]. Furthermore, the least-control principle allows unifying predictive coding with other existing theories for learning in the brain [17, 30] by uncovering possible equivalences of the underlying credit assignment techniques.

## S4.5   Beyond feedforward neural networks.

The underlying theory of predictive coding, which starts from an acyclic graphical model, is tailored towards feedforward neural networks. However, starting from the augmented energy $F$ of Eq. (100), we can generalize predictive coding to equilibrium recurrent neural networks with arbitrary synaptic connectivity matrices $W$ and $U$. Although the link with probabilistic latent variable models is not straightforwardly extendable to equilibrium RNNs, the predictive coding interpretation based on error and prediction neurons remains valid. To the best of our knowledge, our experiments testing this new setting of predictive coding for equilibrium RNNs are the first of their kind.

## S5 Remarks on the least-control principle when using multiple data points

Without loss of generality, we consider a single data point in the formulation of the least-control principle in Section 2, for clarity of presentation. Here, we discuss in more detail how multiple data points can be incorporated in the least-control principle.

### S5.1 Loss defined over finite sample of data points

In many learning problems, as in supervised learning, the loss $L$ is defined over a finite set of data samples:

$$L(\phi) = \sum_{b=1}^{B} L^b(\phi). \tag{102}$$

As the losses associated to different samples are different, the corresponding teaching signal will impact the state differently. We therefore consider $\phi$ to be a concatenation of all the sample-specific states $\{\phi^b\}_{b=1}^{B}$. The loss is then equal to

$$L(\phi) = \sum_{b=1}^{B} L^b(\phi^b). \tag{103}$$

Similarly, we define $f(\phi, \theta)$ as a concatenation of all $f(\phi^b, \theta)$, and $\psi$ as a concatenation of all the controls $\psi^b$. We can then apply the least-control principle on this concatenated quantities.

As the least-control objective of Eq. 3 can be rewritten as the sum $\sum_b \|\psi_\star^b\|^2$, and has constraints that do not interact in between different datapoints $b$, its gradient can also be rewritten as a sum over the datapoints:

$$\frac{\mathrm{d}\mathcal{H}}{\mathrm{d}\theta}(\theta) = \sum_{b=1}^{B} \frac{\mathrm{d}\mathcal{H}^b}{\mathrm{d}\theta}(\theta) \tag{104}$$

with $\mathcal{H}^b(\theta)$ the least-control objective for a single data point $b$. It follows that one can use standard stochastic or mini-batch optimization methods to minimize the least-control objective $\mathcal{H}(\theta)$.

### S5.2 Loss defined by expectation over an infinite inputspace

When there exists an infinite number of data samples (e.g. a continuous input space), the loss can be defined as an expectation over this infinity of data samples:

$$L(\phi) = \mathbb{E}_b[L_b(\phi)] \tag{105}$$

One approach to incorporate this case in the least-control principle is to first sample a finite amount of data samples, and then apply the arguments of Section S5.1.

Another approach is to define the least-control objective as an expectation over the infinity of samples:

$$\begin{aligned} \mathcal{H}(\theta) &= \mathbb{E}_b[\mathcal{H}^b(\theta)] \\ \mathcal{H}^b(\theta) &= \min_{\phi^b, \psi^b} \|\phi^b\|^2 \quad \text{s.t.} \ f(\phi^b, \theta) + \psi^b = 0, \ \nabla_y L(h(\phi^b)) = 0 \end{aligned} \tag{106}$$

Then, under standard regularity conditions for exchanging the gradient and expectation operator, we can sample gradients of this least-control objective:

$$\frac{\mathrm{d}\mathcal{H}}{\mathrm{d}\theta}(\theta) = \mathbb{E}\left[\frac{\mathrm{d}\mathcal{H}^b}{\mathrm{d}\theta}(\theta)\right] \tag{107}$$

with $\mathrm{d}_\theta \mathcal{H}^b(\theta)$ given by Theorem 1.

# S6    The least-control principle for equilibrium RNNs

Here, we provide additional details on the equilibrium recurrent neural network models, the derivation of the local learning rules and the various controller designs.

## S6.1    Equilibrium recurrent neural network model specifications

We use an equilibrium RNN with the following free dynamics:
$$\dot{\phi} = f(\phi, \theta) = -\phi + W\sigma(\phi) + Ux, \tag{108}$$
with $\phi$ the neural activities of the recurrent layer, W the recurrent synaptic weight matrix, U the input weight matrix and $\sigma$ the activation function. We evaluate the performance of this RNN at equilibrium, by selecting a set of output neurons and measuring its loss $L(D\phi^*)$, with $D = [0 \text{ Id}]$ a fixed decoder matrix.

**Learning a decoder matrix.** In practice, we can achieve better performance if we learn a decoder matrix to map the recurrent activity $\phi$ to the output $y$, instead of taking a fixed decoder $D = [0 \text{ Id}]$. To prevent the loss $L(D\phi)$ from depending on the learned parameters $\theta$, we augment the recurrent layer $\phi$ with an extra set of output neurons, and include a learned decoder matrix $\tilde{D}$ inside the augmented recurrent weight matrix $\bar{W}$, leading to the following free dynamics:
$$\dot{\bar{\phi}} = f(\bar{\phi}, \theta) = -\bar{\phi} + \bar{W}\sigma(\bar{\phi}) + \bar{U}x,$$
$$\bar{W} = \begin{bmatrix} W & 0 \\ \tilde{D} & 0 \end{bmatrix}, \quad \bar{U} = \begin{bmatrix} U \\ 0 \end{bmatrix}, \tag{109}$$
with $\bar{\phi}$ the concatenation of $\phi$ with a set of output neurons and $\theta = \{W, U, \tilde{D}\}$ the set of learned parameters. To map the augmented recurrent layer $\bar{\phi}$ to the output $y$, we again use a fixed decoder $D = [0 \text{ Id}]$ that selects the 'output neurons' in $\bar{\phi}$. At equilibrium, we now have $y = D\bar{\phi} = \tilde{D}\sigma(\phi)$. Hence, we see that at equilibrium, the augmented system of Eq. 109 with a fixed decoder $D$ and hence a loss $L(D\bar{\phi})$ independent of $\theta$ is equivalent to the original system of Eq. 108 with a learned decoder $D$ and loss $L(D\sigma(\phi))$. We can now use the least-control principle on the augmented system of Eq. 109 to learn the weight matrices $\theta = \{W, U, \tilde{D}\}$, which we show in the next section.

**Including biases.** In practice, we equip the neurons with a bias parameter $b$, leading to the following free dynamics
$$\dot{\phi} = f(\phi, \theta) = -\phi + W\sigma(\phi) + Ux + b, \tag{110}$$
with the set of learned parameters $\theta = \{W, U, b\}$. This can be extended to the augmented system of Eq. 109 by adding the augmented bias parameters $\bar{b}$, which is a concatenation of the original bias $b$ and the decoder bias $\tilde{b}$. At equilibrium, we now have $y = D\bar{\phi} = \tilde{D}\sigma(\phi) + \tilde{b}$. For ease of notation, we omit the bias terms in most of the derivations in this paper.

**Link with feedforward neural networks.** When the recurrent weight matrix has a lower block-diagonal structure, the equilibrium of the RNN corresponds to a conventional deep feedforward neural network. More specifically, taking
$$W = \begin{bmatrix} 0 & 0 & 0 & 0 \\ W_2 & 0 & 0 & 0 \\ 0 & \ddots & 0 & 0 \\ 0 & 0 & W_L & 0 \end{bmatrix}, \quad U = \begin{bmatrix} W_1 \\ 0 \end{bmatrix}, \quad D = [0 \quad \text{Id}], \tag{111}$$
gives the following feedforward mappings at equilibrium
$$\phi_l = W_l\sigma(\phi_{l-1}) + b_l, \; 1 \le l \le L \tag{112}$$
where we structure $\phi$ into $L$ layers $\phi_l$, and take $\sigma(\phi_0) = x$ and $y = D\phi = \phi_L$. Hence, we see that the equilibrium RNN model also includes the conventional deep feedforward neural network. Similarly, by further structuring $W$ with blocks of Toeplitz matrices, the equilibrium point corresponds with a convolutional feedforward neural network.

**Fixed point iterations.** For computational efficiency, we use fixed point iterations for finding the equilibrium of the free dynamics (108), which we need for the recurrent backpropagation baseline, and for evaluating the test performance of the trained equilibrium RNN.
$$\phi_{\text{next}} = W\sigma(\phi) + Ux \tag{113}$$

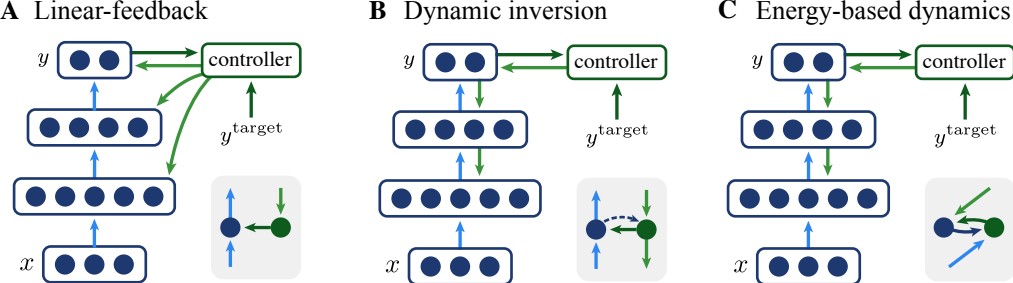

**A** Linear-feedback  **B** Dynamic inversion  **C** Energy-based dynamics

Figure S2: Comparison of the different types of controllers we use on a 2-hidden-layer feedforward neural network in a supervised learning setting. The light blue arrows represent the weights of the network that are used for inference, the light green ones the feedback weights (the $Q$ mapping of Eq.6), the dark green arrows identity mappings and the dark blue ones transformations through the activation function. Dashed arrows indicates the use of the activation function derivative. The grey rectangle shows how a state neuron $\phi$ and its corresponding control neuron $\psi$ are connected, along with how they are connected to the layers above and below. For example, a light blue bottom-up arrow on the left indicates that the state neurons receive bottom-up input from the state neurons of the layer below through the inference weights. Alternatively, a light green top-down arrow from the right to the left shows that the control signal from the layer above influence the state neurons through the feedback weights. (A) Linear feedback (LF). The output control signal $u$ is directly propagated to the hidden layers through the linear mapping $\psi = Qu$. (B) Dynamic inversion (DI). The control signal is propagated through the hidden layers by leveraging layer-wise feedback mechanisms $\dot{\psi} = -\psi + \sigma'(\phi)S\psi + D^\top u$ and influences the state variable jointly with bottom up input: $\dot{\phi} = -\phi + W\sigma(\phi) + Ux + \psi$. (C) Energy-based dynamics (EBD). The control signal is here equal to the difference between the current state and the bottom-up input ($\psi = \phi - W\sigma(\phi) - Ux$) and is indirectly propagated to the remote layers through the state variables: $\dot{\phi} = -\psi + \sigma'(\phi)S\psi + D^\top u$.

### S6.2  Deriving a local learning rule for equilibrium RNNs

The least-control principle prescribes to augment the RNN dynamics of Eq. 108 with an optimal controller $\psi$ that leads the neural dynamics to a controlled equilibrium $\phi_*$, where the output neurons $y_* = D\phi_*$ minimize the loss $L$, while being of minimum norm $\|\psi_\star\|$. This optimal control contains a useful learning signal, which we leverage in Theorem 1 to derive local first-order parameter updates $\Delta\theta = -\mathrm{d}_\theta\mathcal{H}(\theta) = \psi_\star^\top \partial_\theta f(\phi_\star, \theta)$, with $(\psi_\star, \phi_\star)$ the minimizers of the least-control problem of Eq. 3, i.e. the optimal control and optimally controlled state, respectively. In the next sections, we discuss in detail how to compute $(\psi_\star, \phi_\star)$ for the equilibrium RNN using neural dynamics. Here, we apply the first-order update $\Delta\theta$ to the equilibrium RNN to arrive at the local synaptic updates of Eq. 12.

**Derivation of the local update rules.** To avoid using tensors in $\partial_\theta f(\phi_\star, \theta)$, we vectorize the weight matrices $W$ and $U$ and use Kronecker products to rewrite the free dynamics of Eq. 108:[5]

$$\dot{\phi} = -\phi + (\sigma(\phi)^\top \otimes \mathrm{Id})\vec{W} + (x^\top \otimes \mathrm{Id})\vec{U} + b \tag{114}$$

with $\otimes$ the Kronecker product, Id the identity matrix and $\vec{W}$ and $\vec{U}$ the vectorized form (concatenated columns) of $W$ and $U$, respectively. Taking $\theta$ the concatenation of $\vec{W}, \vec{U}$ and $b$, we get

$$\Delta\theta = \psi_\star^\top \frac{\partial f}{\partial \theta}(\phi_\star, \theta) = \psi_\star^\top \left[(\sigma(\phi)^\top \otimes \mathrm{Id}) \quad (x^\top \otimes \mathrm{Id}) \quad \mathrm{Id}\right] \tag{115}$$

We can rewrite the resulting update in matrix form again, leading to the following parameter updates for the RNN:

$$\Delta W = \psi_\star \sigma(\phi_\star)^\top, \quad \Delta U = \psi_\star x^\top, \quad \Delta b = \psi_\star. \tag{116}$$

Now assuming we have a controller $\psi$ that leads the dynamics to the optimally controlled state $(\psi_*, \phi_*) = (\psi_\star, \phi_\star)$, we arrive at the updates of Eq. 12. Note that we can use these updates in any

---

[5]For this, we use the following property: $\mathrm{vec}(ABC) = (C^\top \otimes A)\mathrm{vec}(B)$, with vec the vectorization operator that concatenates the columns of a matrix and $\otimes$ the Kronecker product.

gradient-based optimizer. Applying a similar derivation to the augmented RNN dynamics of Eq. 109, we arrive at the following updates for $\theta = \{W, U, b, \tilde{D}, \tilde{b}\}$:

$$\Delta W = \psi_\star \sigma(\phi_\star)^\top, \quad \Delta U = \psi_\star x^\top \quad, \Delta \tilde{D} = \tilde{\psi}_\star \sigma(\phi_\star)^\top, \quad \Delta b = \psi_\star, \quad \Delta \tilde{b} = \tilde{\psi}_\star \quad (117)$$

with $\tilde{\psi}$ the control applied on the set of output neurons within $\bar{\phi}$, and $\psi$ the control applied to the set of recurrent neurons $\phi$ (i.e. $\bar{\phi}$ without the output neurons).

**Controller dynamics.** Theorem 2 shows that we can compute an optimal control $\psi_\star$ by using any controller $\psi$ that leads the system to a controlled equilibrium

$$0 = f(\phi_*, \theta) + Q(\phi_*, \theta)u_*, \quad 0 = -\alpha u_* - \frac{\partial L}{\partial y}(h(\phi_*))^\top \quad (118)$$

in the limit of $\alpha \to 0$ and with $Q(\phi_*, \theta)$ satisfying the column space condition

$$\text{col}\left[Q(\phi_*, \theta)\right] = \text{row}\left[\frac{\partial h}{\partial \phi}(\phi_*)\left(\frac{\partial f}{\partial \phi}(\phi_*, \theta)\right)^{-1}\right]. \quad (119)$$

Under these conditions, we have that $Q(\phi_*, \theta)u_*$ is an optimal control $\psi_\star$. Applied to the equilibrium RNN setting, this result in the following column space condition.

$$\text{col}\left[Q(\phi_*, \theta)\right] = \text{row}\left[D\big(\text{Id} - W\sigma'(\phi_*)\big)^{-1}\right], \quad (120)$$

with $\sigma'(\phi) := \partial_\phi \sigma(\phi)$. In the next sections, we propose various control dynamics $\dot{\psi}$ that satisfy these conditions at equilibrium and hence lead to an optimal control $\psi_\star$.

## S6.3   A controller with direct linear feedback

We start with the simplest controller, that broadcasts the output controller $u$ to the recurrent neurons $\phi$ with a direct linear mapping $Q$, resulting in the following dynamics:

$$\tau\dot{\phi} = -\phi + W\sigma(\phi) + Ux + Qu, \quad \tau_u \dot{u} = -\frac{\partial L}{\partial y}(D\phi)^\top - \alpha u, \quad (121)$$

with $\tau$ and $\tau_u$ the time constants of the neural dynamics and output controller dynamics, respectively. We visualize an example of such a direct linear feedback controller on a feedforward neural network in Figure S2.A. We assume without further discussion that the output controller $u$ which integrates the output error $\partial_y L$ is implemented by some neural integrator circuit, a topic which has been extensively studied in ref. [99].

### S6.3.1   A single-phase learning scheme for updating the feedback weights

Theorem 2 guarantees that this dynamics converges to the optimal control $\psi_\star$ if the columnspace condition of Eq. 120 is satisfied. As $Q$ is now a linear mapping, this condition cannot be exactly satisfied for all samples in the dataset, as the right-hand side of Eq. 120 is data-dependent, whereas $Q$ is not. Still, we learn the feedback weights $Q$ simultaneously with the other weights $W$ and $U$, to approximately fulfill this column space condition, without requiring separate phases. For this, we add noise to the dynamics and we adapt a recently proposed feedback learning rule for feedforward neural networks [30, 51] towards our RNN setup, leading to the following system dynamcis and simple anti-Hebbian plasticity dynamics:

$$\tau\dot{\phi} = -\phi + W\sigma(\phi) + Ux + Qu + s\epsilon \quad (122)$$

$$\tau_Q \dot{Q} = -s\epsilon u_{\text{hp}}^\top - \gamma_Q Q \quad (123)$$

Here, $\gamma_Q$ is a small weight-decay term, $\epsilon$ is noise from an Ornstein-Uhlenbeck process, i.e. exponentially filtered white noise with standard deviation $s$ and timeconstant $\tau_\epsilon$. As post-synaptic plasticity signal, we use a high-pass filtered version $u_{\text{hp}}$ of the output control $u$, to extract the noise fluctuations. We model the plasticity with continuous dynamics instead of a single update at equilibrium, as the noise $\epsilon$ changes continuously. Building upon the work of Meulemans et al. [30, 51], we show in Proposition S10 that the feedback learning dynamics of Eq. 123 drives the feedback weights to a setting that satisfies the column space condition of Eq. 120, when trained on a single data point.

**Proposition S10.** *Assuming (i) a separation of timescales $\tau \ll \tau_\epsilon \ll \tau_u \ll \tau_Q$, (ii) a small noise variance $s^2 \ll \|\phi\|^2$, (iii) a high-pass filtered control signal $u_{\mathrm{hp}} = u - u_{\mathrm{ss}}$ with $u_{\mathrm{ss}}$ the steady state of the first moment of the output controller trajectory $u$, and (iv) stable dynamics, then, for a single fixed input $x$, the feedback learning dynamics of Eq. 123 let the first moment $\mathbb{E}[Q]$ converge towards a setting satisfying the column space condition of Theorem 2:*

$$\mathbb{E}\left[Q_{\mathrm{ss}}^\top\right] \propto \frac{\partial^2 L}{\partial y^2}(D\phi_{\mathrm{ss}})D(\mathrm{Id} - W\sigma'(\phi_{\mathrm{ss}}))^{-1} \tag{124}$$

*with $\phi_{\mathrm{ss}}$ the steady state of the first moment of the state trajectory $\phi$, and $\sigma'(\phi) = \partial_\phi \sigma(\phi)$.*

*Proof.* The general idea behind the feedback learning rule of Eq. 123 is to correlate the noise fluctuations of the output controller $u$ with the noise $\epsilon$ inside the neurons that caused these fluctuations. As the injected noise propagates through the whole network, and the neural dynamics $\dot{\phi}$ evolve on a faster timescale compared to the noise $\epsilon$, the noise fluctuations in the output contain useful information on the network transformation $D(\mathrm{Id} - W\sigma')^{-1}$ which can be extracted by correlating these fluctuations with the noise $\epsilon$. We refer the interested reader to Appendix C of Meulemans et al. [30] for a detailed discussion of this type of feedback learning rule for feedforward neural networks.

First, we define the neural fluctuations $\tilde{\phi} := \phi - \phi_{\mathrm{ss}}$ and output controller fluctuations $\tilde{u} = u - u_{\mathrm{ss}}$, with $\phi_{\mathrm{ss}}$ and $u$ the steady state of the first moment of $\phi$ and $u$, respectively. Using the assumption that the noise variance $s^2$ is much smaller than the neural activity $\|\phi\|^2$ and that we have stable (hence contracting) dynamics, we perform a first-order Taylor approximation around $\phi_{\mathrm{ss}}$ and $u$, which for simplicity we assume to be exact, leading to the following dynamics:

$$\tau\dot{\tilde{\phi}} = -\tilde{\phi} + W\sigma'(\phi_{\mathrm{ss}})\tilde{\phi} + Q\tilde{u} + \epsilon, \tag{125}$$

$$\tau_u \dot{\tilde{u}} = \frac{\partial^2 L}{\partial y^2}(D\phi_{\mathrm{ss}})D\tilde{\phi} - \alpha\tilde{u} \tag{126}$$

where we used that $-\phi_{\mathrm{ss}} + W\sigma(\phi_{\mathrm{ss}}) + Ux + Qu_{\mathrm{ss}} = 0$ and $-\partial_y L(D\phi_{\mathrm{ss}}) - \alpha u_{\mathrm{ss}} = 0$ by definition. To compress notation, we introduce the variable $\nu$ as a concatenation of $\tilde{\phi}$ and $\tilde{u}$.

$$\dot{\nu} = -A\nu + B\epsilon$$

$$A := \begin{bmatrix} \frac{1}{\tau}(\mathrm{Id} - W\sigma'(\phi_{\mathrm{ss}})) & -\frac{1}{\tau}Q \\ \frac{1}{\tau_u}\partial_y^2 L(D\phi_{\mathrm{ss}})D & \frac{\alpha}{\tau_u}\mathrm{Id} \end{bmatrix} \quad B := \begin{bmatrix} \frac{s}{\tau}\mathrm{Id} \\ 0 \end{bmatrix} \tag{127}$$

We solve this linear time-invariant stochastic differential equation using the method of variation of constants [100], while assuming that $\nu = 0$ at time $t_0$, i.e. that the dynamics already have converged to the steady state at $t_0$.

$$\nu(t) = \int_{t_0}^t \exp\left(-A(t - t')\right)B\epsilon(t')\mathrm{d}t'. \tag{128}$$

Now we turn to the first moment of the feedback weight learning (123), using $\tilde{u} = C\nu$ with $C = [0 \; \mathrm{Id}]$:

$$\mathbb{E}\left[\tilde{u}(t)\epsilon(t)^\top\right] = C\mathbb{E}\left[\nu(t)\epsilon(t)^\top\right] = \frac{1}{2\tau_\epsilon}C\int_{t_0}^t \exp\left(-\left(A + \frac{1}{\tau_\epsilon}\mathrm{Id}\right)(t - t')\right)B\mathrm{d}t' \tag{129}$$

for which we used that $\mathbb{E}\left[\epsilon(t)\epsilon(t')^\top\right] = \frac{1}{2\tau_\epsilon}\exp(\frac{1}{\tau_\epsilon}|t - t'|)$ for an Ornstein-Uhlenbeck process with time constant $\tau_\epsilon$. Now assuming that $t \gg t_0$ and that we have stable dynamics (i.e. $A$ has strictly positive eigenvalues), we can solve the integral of the matrix exponential:

$$\mathbb{E}\left[\tilde{u}(t)\epsilon(t)^\top\right] = \frac{1}{2\tau_\epsilon}C\left(A + \frac{1}{\tau_\epsilon}\mathrm{Id}\right)^{-1}B \tag{130}$$

We solve the inverse of this $2 \times 2$ block matrix analytically (following e.g. Lu and Shiou [101]), leading to

$$\mathbb{E}\left[\tilde{u}(t)\epsilon(t)^\top\right] = \frac{1}{2\tau_\epsilon}\frac{s}{\tau}\left[-\Delta^{-1}\frac{1}{\tau_u}\frac{\partial^2 L}{\partial y^2}(D\phi_{\mathrm{ss}})D\tau(\mathrm{Id} - W\sigma'(\phi_{\mathrm{ss}}))^{-1}\right]$$

$$\Delta := \left(\frac{\alpha}{\tau_u} + \frac{1}{\tau_\epsilon}\right)\mathrm{Id} + \frac{1}{\tau_u}\frac{\partial^2 L}{\partial y^2}(D\phi_{\mathrm{ss}})D\left(\frac{1}{\tau}(\mathrm{Id} - W\sigma'(\phi_{\mathrm{ss}}))\right)^{-1}\frac{1}{\tau}Q \tag{131}$$

Using the separation of timescales $\tau \ll \tau_\epsilon \ll \tau_u$, this simplifies to

$$\mathbb{E}\left[\tilde{u}(t)\epsilon(t)^\top\right] \approx -\frac{s}{2\tau_u}\frac{\partial^2 L}{\partial y^2}(D\phi_{\mathrm{ss}})D(\mathrm{Id} - W\sigma'(\phi_{\mathrm{ss}}))^{-1} \tag{132}$$

Remembering that $\tilde{u} = u_{\mathrm{hp}}$, we get the following plasticity dynamics using Eq. 123:

$$\tau_Q\frac{\mathrm{d}}{\mathrm{d}t}\mathbb{E}\left[Q(t)^\top\right] := \tau_Q\frac{\mathrm{d}}{\mathrm{d}t}\bar{Q}(t) = \frac{s^2}{2\tau_u}\frac{\partial^2 L}{\partial y^2}(D\phi_{\mathrm{ss}})D(\mathrm{Id} - W\sigma'(\phi_{\mathrm{ss}}))^{-1} - \gamma_Q\bar{Q}(t) \tag{133}$$

with $\bar{Q}(t)$ the first moment of $Q(t)$. For $\gamma > 0$, these dynamics are stable and lead to

$$\mathbb{E}\left[Q_{\mathrm{ss}}^\top\right] = \frac{s^2}{2\tau_u\gamma_Q}\frac{\partial^2 L}{\partial y^2}(D\phi_{\mathrm{ss}})D(\mathrm{Id} - W\sigma'(\phi_{\mathrm{ss}}))^{-1} \tag{134}$$

thereby concluding the proof. $\qquad\square$

**Isolating the postsynaptic noise $\epsilon$ with a multicompartment neuron model.** The feedback learning rule of Eq. 123 and the corresponding Proposition S10 represent the feedback learning idea in its simplest form. However, for this learning rule, the feedback synapses need some mechanism to isolate the postsynaptic noise $\epsilon$ from the (noisy) neural activity. Meulemans et al. [30, 51] solve this by considering a multicompartment model of the neuron, inspired by recent dendritic compartment models of the cortical pyramidal neuron [14, 17, 41]. Here, a feedback (apical) compartment integrates the feedback input $Qu$, a basal compartment integrates the recurrent and feedforward input $W\sigma(\phi) + Ux$ and a central (somatic) compartment combines the two other compartments together and transmits the output firing rate $\sigma(\phi)$ to the other neurons. In this model, we can assume that a part of the noise $\epsilon$ enters through the feedback compartment, and change the feedback learning rule to

$$\tau_Q\dot{Q} = -\left(Qu + s\epsilon^{\mathrm{fb}}\right)u_{\mathrm{hp}}^\top - \gamma_Q Q, \tag{135}$$

where $\left(Qu + s\epsilon^{\mathrm{fb}}\right)$ is the feedback compartment activity, and hence locally available for the synaptic updates. As $\mathbb{E}[Quu_{\mathrm{hp}}^\top] = Q\mathbb{E}[u_{\mathrm{hp}}u_{\mathrm{hp}}^\top] = Q\Sigma_u$ with $\Sigma_u$ the positive definite auto-correlation matrix of $u_{\mathrm{hp}}$, the result of Proposition S10 can be adapted to $\mathbb{E}\left[Q_{\mathrm{ss}}^\top\right] \propto M\partial_y^2 L(D\phi)_{\mathrm{ss}})D(\mathrm{Id} - W\sigma'(\phi_{\mathrm{ss}}))^{-1}$, with $M = (\Sigma_u + \gamma\mathrm{Id})^{-1}$ a positive definite matrix. This new fixed point of $Q$ also satisfies the column space condition of Eq. 120.

**Improving the parameter updates.** When the column space condition is not perfectly satisfied, as is the case with this direct linear feedback controller when multiple data points are used, the empirical performance can be improved by changing the parameter updates towards

$$\Delta W = \sigma'(\phi_*)Qu_*\sigma(\phi_*)^\top, \quad \Delta U = \sigma'(\phi_*)Qu_*x^\top \tag{136}$$

This learning rule uses the local derivative of the nonlinearity $\sigma$ as a heuristic to prevent saturated neurons from updating their weights, which is known to improve performance in feedforward networks [30, 51].

**Alignment results.** We empirically tested the controller with direct linear feedback and trained feedback weights $Q$ on the MNIST digit recognition task. Figure S3 shows that the updates of Eq. 136 are approximately aligned to the gradients $-\nabla_\theta\mathcal{H}(\theta)$ on the least-control objective, indicating that the feedback weight learning for $Q$ is successful and the controller with direct linear feedback approximates the optimal control.

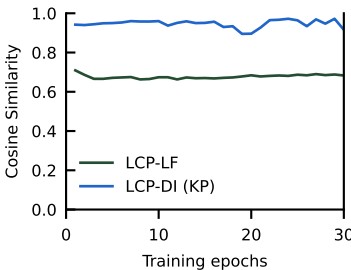

Figure S3: The linear feedback controller and inversion dynamics lead to approximate optimal control. Cosine similarity between the true gradient $-\nabla_W\mathcal{H}(\theta)$ and approximate updates $\Delta W$ of the least-control principle for training an equilibrium RNN on MNIST: a controller with trained linear feedback connections (LCP-LF) and a controller using dynamic inversion with trained feedback weights via Kolen-Pollack learning (LCP-DI KP).

### S6.3.2 Simulation details

For computational efficiency, we run fixed-point iterations for finding the equilibrium of the controlled neural dynamics shown in Eq. 121, which in this case is equivalent to simulating the dynamics with the forward Euler method with a simulation stepsize equal to the timeconstant:

$$\phi_{\text{next}} = W\sigma(\phi) + Ux + Qu$$
$$u_{\text{next}} = (1 - \alpha)u - \frac{\partial L}{\partial y}(D\phi) \tag{137}$$

We use the fixed points $\phi_*$ and $u_*$ to update the parameters $\theta = \{W, U, b, \tilde{D}, \tilde{b}\}$ according to Eq. 117 with $\psi_* = Qu_*$. For training the feedback weights, we simulate the stochastic differential equations of Eq. 122 with the Euler-Maruyama method [100], starting from the fixed points computed in Eq. 137:

$$\epsilon_{\text{next}} = \epsilon + \frac{\Delta t}{\tau_\epsilon}\left(-\epsilon + \frac{1}{\sqrt{\Delta t}}\Delta\xi\right)$$
$$\phi_{\text{next}} = \phi + \frac{\Delta t}{\tau}\left(-\phi + W\sigma(\phi) + Ux + Qu + s\epsilon\right) \tag{138}$$
$$u_{\text{next}} = u + \frac{\Delta t}{\tau_u}\left(-\frac{\partial L}{\partial y}(D\phi) - \alpha u\right)$$

with $\Delta t$ the simulation stepsize and $\Delta\xi \sim \mathcal{N}(0, \text{Id})$ white Gaussian noise that is drawn independently for each simulation step. The factor $\frac{1}{\sqrt{\Delta t}}$ in front of $\Delta\xi$ is due specific properties of Brownian motion in the simulation of stochastic differential equations. For computational efficiency, we accumulate the weight updates for $Q$ over the simulation interval, and apply them at the end:

$$\Delta Q = -\frac{1}{\tau_Q}\left[\frac{1}{s^2}\sum_m (Qu[m] + \epsilon[m])(u[m] - u_*)^\top - \gamma Q\right] \tag{139}$$

where $m$ is the simulation step and we use $u_{\text{hp}} = u - u_*$, with $u_*$ the fixed point of Eq. (137), and where we scale by the noise variance $s^2$ to remove the dependence of the update magnitude on this hyperparameter. For all parameter updates, we use the Adam optimizer.

### S6.4 Using general-purpose dynamics for computing an optimal control

In the main text (Section 3.1.2) we discussed the application of our two general-purpose optimal control methods to an equilibrium RNN. Here, we provide an expanded discussion on the neural network architectures each method yields. While we leave the development of more detailed biological models for future work, we comment on how these controlled RNNs may be implemented in cortical circuits, sketching a number of alternative circuit designs. We end this section with providing additional simulation and experimental details.

### S6.4.1 Dendritic error compartments for dynamic inversion

In Section 3.1.2 we applied our general-purpose dynamic inversion method (8) to steer equilibrium RNNs towards optimally-controlled states. We restate here the resulting optimal control circuit (14) and discuss afterwards two possible network implementations, where feedback error signals are represented in dendritic compartments. Assuming without loss of generality that the output controller is of the leaky-integral kind, the resulting network dynamics (14) is:

$$\dot{\phi} = -\phi + W\sigma(\phi) + Ux + \psi,$$
$$\dot{\psi} = -\psi + \sigma'(\phi)S\psi + D^\top u \tag{140}$$
$$\dot{u} = -\alpha u - \nabla_y L(y).$$

where $y = D\phi$ is the output neuron state, a subset of the full neuron state $\phi$ determined by a fixed selector matrix $D$. Figure S2.B shows an instantiation of this circuit on a feedforward neural network.

In Eq. 140, the control variable $\psi$ follows its own dynamics $\dot{\psi}$, which serves the purpose of inverting $\partial_\phi f(\phi, \theta)^\top$. This can be seen by inspecting the controlled steady state:

$$\psi_* = (\text{Id} - \sigma'(\phi_*)S)^{-1}D^\top u_*. \tag{141}$$

If the inverted factor appearing in the equation above would correspond to $\partial_\phi f(\phi_*, \theta)^{-\top}$, the network would be at an optimally-controlled state, where the weight updates following from our theory assume the Hebbian form

$$\Delta W = \psi_* \sigma(\phi_*)^\top$$
$$\Delta U = \psi_* x^\top. \tag{142}$$

For this to be the case, we need to satisfy the symmetry requirement $S = W^\top$. This corresponds to a form of weight transport, assuming that $S$ and $W$ are two distinct sets of synaptic connections.

We present now two alternative circuit implementations which allow maintaining $S \approx W^\top$ using local learning rules, so that the equations governing the change $\Delta W$, $\Delta U$ and $\Delta S$ of all synaptic weights only depend on quantities presumed to be available pre- and post-synaptically and at the same time. In both cases, we build upon previous theories of error-driven cortical learning and assume that our control signal $\psi$ corresponds to the apical dendritic activity of a cortical pyramidal cell, and $\phi$ to its somatic activity. According to this view, apical dendrites encode neuron-specific credit assignment signals $\psi_*$ which are responsible for determining which neurons should potentiate or depress their connections, and by how much. We therefore extend previous models of apically-controlled synaptic plasticity [14, 18, 20, 23, 41], for which there is some experimental evidence [102], to the case of recurrently-connected networks.

**Multiplexed codes.** Interpreting our apical dendrite dynamics $\dot{\psi}$ in literal terms appears to require dendrodendritic synapses which bypass the soma, due to the apical-to-apical term $\psi \leftarrow S\psi$ occurring in (140). While such connections are known to exist in the brain [103], they are rare. We thus highlight an alternative implementation, which builds on the interesting proposal that pyramidal cells might be capable of transmitting credit assignment signals alongside their intended outputs, by means of a multiplexed code [14, 23]. The basic premise of this theory is that certain synaptic connections are sensitive to high-frequency bursts of spikes only, which can be triggered by increasing apical activity, whereas other connections are tuned to filter out such events. Each connection type would then differentially reflect either apical or somatic activity, respectively; the required specific tuning to burst events or otherwise can be achieved for example by resorting to different profiles of short-term plasticity for each connection type. Here, we leverage on this idea and propose to replace the dendrodendritic (direct apical-to-apical) connections appearing in (140) by common (axodendritic) apical-targetting synapses mediated through the soma (i.e., soma-to-apical $\psi \leftarrow \phi$ synapses), tuned to be sensitive to burst events. This design enables keeping the circuit approximately symmetric ($S \approx W^\top$) through local burst-dependent plasticity rules, $\Delta S = \sigma(\phi_*)\psi_*^\top - \gamma S$ and $\Delta W = \psi_* \sigma(\phi_*)^\top - \gamma W$, a form of Kolen-Pollack learning [42, 54].

Interestingly, the resulting network model differs in a conceptually important – and experimentally testable – way from the multiplexing model originally proposed by Payeur et al. [23]: in our model, apical control signals should steer (thus, influence) the network state $\phi$ towards a controlled equilibrium, and therefore should *not* be filtered out from the somatic voltage dynamics $\dot{\phi}$, while the model of Payeur et al. [23] fully multiplexes credit assignment information and the actual computations carried out by the network. Critically, our major functional requirement is that apical-targetting connections be capable of selectively transmitting burst events (e.g., through short-term facilitating plasticity), and we make no further claims about the short-term plasticity of the remaining connections, leaving open the possibility of using this mechanism for other computational purposes unrelated to multiplexing.

**Dendritic errors via lateral inhibition.** Our network dynamics (140) is also compatible with an alternative proposal for how apical dendrites may come to encode credit assignment information, then used to steer synaptic plasticity within a neuron [17, 39]. In that model, dendritic errors emerge thanks to an auxiliary population of neurons, modeled after a particular class of inhibitory interneurons which preferentially target apical dendrites. These interneurons are set up such that they learn to reproduce the activity of pyramidal neurons. Both pyramidal neurons and interneurons feedback to apical dendrites, but with opposite sign. Sacramento et al. [17] showed that when the interneurons perfectly track pyramidal neurons (sometimes referred to as a 'tight' inhibitory-excitatory balance [104]) and an output error is generated, the remaining steady-state apical activity corresponds to a backpropagated error.

We briefly sketch here a proof-of-concept mapping of our dynamics (140) onto this model, while leaving more realistic implementations to future work. We first replicate our recurrently-connected

neurons with a population of interneurons, whose state we denote by $\delta$, and let both types of neurons feedback onto apical dendrites. Now, we follow the supplementary analysis of ref. [17] and assume that both populations have exactly the same connectivity and weights, a condition to ensure successful tracking and error propagation, which can be reached with the local weight updates for the interneurons introduced in ref. [17]. Moreover, for simplicity, here we assume our neurons are linear and ignore factors involving the derivative of the transfer function $\partial_\phi \sigma$, although it is possible to take such factors into account with a more detailed construction. Taking $\dot{\psi} = -\psi + S\phi - S\delta + D^\top u$ as our apical dendrite dynamics and inspecting its steady-state yields $\psi_* = (\mathrm{Id} - S)^{-1} D^\top u_*$, as the interneuron activity perfectly tracks the pyramidal neuron activity $W\phi + Ux$, but does not include the apical activity $\psi$. This is the optimally-controlled equilibrium we sought. As with the multiplexed code described above, approximate symmetry $S \approx W^\top$ can then be maintained with local rules via Kolen-Pollack learning.

### S6.4.2   Error neurons for energy-based dynamics

In the main text and in Sections S2.5 and S4, we have seen that the optimal control problem arising under our principle can be generically approached using a (constrained) energy-minimizing dynamics. We restate here the simple gradient-flow dynamics (15) presented in the main text:

$$\begin{aligned}
\dot{\phi} &= -\psi + \sigma'(\phi)S\psi + D^\top u, \\
\psi &= \phi - W\sigma(\phi) - Ux,
\end{aligned} \tag{143}$$

where $u$ can be any output controller which enforces the constraint that the loss $L$ is minimized with respect to the output neural activity $y$. The circuit corresponding to those dynamics is shown on Figure S2.C. We now briefly discuss how this system can be implemented using a predictive coding architecture featuring two distinct subpopulations of neurons, generalizing the model of Whittington and Bogacz [36] to the RNN setting. In this architecture, $\psi$ is interpreted as the activity of a population of prediction error neurons, while $\phi$ corresponds to the activity of prediction neurons. Error neurons $\psi$ act as comparators, which measure deviations of the actual state of prediction neurons $\phi$ from the expected state $\mu := W\sigma(\phi) - Ux$; in this model, such deviations dynamically adjust the predictions $\phi$ towards an optimally-controlled steady-state in which the output loss is minimal (c.f. Section S4). The critical architectural difference to the models of Whittington and Bogacz [36] and Rao and Ballard [56] is that in our case the expected state $\mu$ does not necessarily stem from different network layers. A hierarchically-structured connectivity matrix $W$ is a special case of our model; we allow for top-down, lateral and bottom-up contributions to the expected states $\mu$.

### S6.4.3   Training deep equilibrium models with the least-control principle.

We use deep equilibrium models that consist out of three main modules: (i) an encoder $g(x, \theta)$ that maps the input $x$ to the implicit layer $\phi$, (ii) an implicit layer defined by $f(\phi, \theta) = 0$ where $g(x, \theta)$ is incorporated inside $f(\phi, \theta)$, and (iii) a decoder $h(\phi, \theta)$ that maps the implicit layer towards the output $y$.

When $h(\phi, \theta)$ is independent of $\theta$, we can use the inversion dynamics of Eq. 8 or the energy-based dynamics of Eq. 10 to compute an optimal control $\psi_\star$ and use the first-order parameter updates of Theorem 1. To harvest the full power of deep equilibrium models, it is beneficial to train the decoder $h(\phi, \theta)$ as well. However, this poses the problem that the loss $L(h(\phi, \theta))$ is now dependent on $\theta$ and hence Theorem 1 cannot be applied out-of-the-box. There are two general approaches to solve this challenge: (i) similar to the vanilla equilibrium RNNs, we can extend the state $\phi$ with an extra set of output neurons to absorb the $\theta$-dependent decoder $h(\phi, \theta)$ inside of the implicit layer; and (ii) we can use the extended version of the least-control principle derived in Section S2.2 that can handle $\theta$-dependent losses, to derive the parameter updates.

**Augmenting the implicit layer.** We can use an augmented state $\bar{\phi}$ that concatenates the system states $\phi$ with an extra set of output neurons $\tilde{\phi}$. Next we augment the implicit layer $f(\phi, \theta)$ towards

$$\bar{f}(\bar{\phi}, \theta) = \begin{bmatrix} f(\phi, \theta) \\ h(\phi, \theta) - \tilde{\phi} \end{bmatrix}. \tag{144}$$

Now, we can use a fixed linear decoder $D = [0 \; \mathrm{Id}]$ to select the output states $\tilde{\phi}$ for the loss $L(D\bar{\phi})$, which now is independent of $\theta$. This augmented system is then controlled by $\bar{\psi}$ which has the

following dynamics when using the dynamic inversion algorithm of Eq. 8:

$$\dot{\tilde{\psi}} = \frac{\partial \bar{f}}{\partial \bar{\phi}}(\bar{\phi}, \theta)^\top \bar{\psi} + D^\top u \tag{145}$$

Combining this with the system dynamics of Eq. 144 leads to the following set of equilibrium conditions

$$
\begin{aligned}
0 &= f(\phi_*, \theta) + \psi_* \\
0 &= -\tilde{\phi}_* + h(\phi_*, \theta) + \tilde{\psi}_* \\
0 &= \frac{\partial f}{\partial \phi}(\phi_*, \theta)^\top \psi_* + \frac{\partial h}{\partial \phi}(\phi_*, \theta)^\top \tilde{\psi}_* \\
0 &= -\tilde{\psi}_* + u_* \\
0 &= \alpha u_* + \frac{\partial L}{\partial y}(\phi_*)
\end{aligned} \tag{146}
$$

with $\tilde{\psi}$ the control applied to the augmented output neurons $\tilde{\phi}$. Grouping the parameters $\theta$ into the encoder parameters $\theta_g$, the implicit module parameters $\theta_f$, and the decoder parameters $\theta_h$, we get the following parameter gradients:

$$\frac{\mathrm{d}}{\mathrm{d}\theta_f}\mathcal{H}(\theta) = \psi_*^\top \frac{\partial f}{\partial \theta_f}(\phi_*, \theta), \quad \frac{\mathrm{d}}{\mathrm{d}\theta_g}\mathcal{H}(\theta) = \psi_*^\top \frac{\partial f}{\partial \theta_g}(\phi_*, \theta), \quad \frac{\mathrm{d}}{\mathrm{d}\theta_h}\mathcal{H}(\theta) = \tilde{\psi}_*^\top \frac{\partial h}{\partial \theta_h}(\phi_*, \theta) \tag{147}$$

Note that due to the general treatment, if the implicit module, encoder or decoder consist of multiple computational layers, the above partial derivatives do not result in a local update of presynaptic activity times postsynaptic control signal, but require instead to backpropagate the credit assignment signal (the control) through the computational layers within the module. The updates however remain first-order updates which is important for computational efficiency, and they remain local in time in contrast to recurrent backpropagation (c.f. Section S3). When a design requirement is to have local updates in space, e.g. for biological plausibility, we can augment the system state $\phi$ with states representing each computational layer in the modules, such that each layer receives a dedicated control, making the resulting updates local in time and space.

**Using the least-control principle for $\theta$-dependent losses.** In Section S2.2, we extend the least-control principle to $\theta$-dependent losses, and among others derive a first-order update when the decoder $h(\phi, \theta)$ is $\theta$-dependent, which we restate below.

$$\frac{\mathrm{d}}{\mathrm{d}\theta}\mathcal{H}(\theta) = -\psi_*^\top \frac{\partial f}{\partial \theta}(\phi_*, \theta) - u_*^\top \frac{\partial h}{\partial \theta}(\phi_*, \theta) \tag{148}$$

This first-order learning rule equals the gradient when the inversion dynamics (8) or energy-based dynamics (10) are used for computing the optimal control $\psi_\star$, in the limit of $\alpha \to 0$.

As the strategy of augmenting the implicit layer to have a loss independent of $\theta$ only uses the theory explained in the main manuscript, we use this strategy for our experiments.

### S6.4.4 Simulation details

**Equilibrium RNNs.** We use the following fixed point iterations for finding the equilibrium of the inversion dynamics of Eq. 14:

$$
\begin{aligned}
\phi_{\text{next}} &= W\sigma(\phi) + Ux + \psi \\
\psi_{\text{next}} &= \sigma'(\phi)S\phi + D^\top u \\
u_{\text{next}} &= (1 - \alpha)u - \frac{\partial L}{\partial y}(D\phi)^\top
\end{aligned} \tag{149}
$$

The fixpoint iterations terminates for a given input when either the relative change in the norm of the dynamic states $\Phi = \{\phi\}$ or $\Phi = \{\phi, \psi, u\}$ is less than some threshold $\epsilon$, i.e. $\frac{\|\Phi_t - \Phi_{t+1}\|^2}{\|\Phi_t\|\|\Phi_{t+1}\|} \leq \epsilon$, or when some iteration budget `max_steps` is exhausted.

When we learn the feedback weights $S$ with Kolen-Pollack learning (LCP-DI KP), we use the following weight updates for $S$ and $W$:

$$\Delta W = \psi_* \sigma(\phi)^\top - \gamma W, \quad \Delta S = \sigma(\phi)\psi_*^\top - \gamma S \tag{150}$$

with $\gamma$ the weight decay parameter. When we do not train the feedback weights $S$ (LCP-DI), we fix them towards the transpose of the recurrent weights $S = W^\top$ and we use standard weight update for $W$ without weight decay (12).

**Deep equilibrium models.** We generalize the above fixed point iterations to more complex recurrent architectures, by using the abstract inversion dynamics of Eq. 8 and the equilibrium equations of Eq. 146. Note that we directly substitute $\tilde{\psi}_* = u_*$ in the fixed point equations, to eliminate an equation.

$$
\begin{aligned}
\phi_{\text{next}} &= \phi + f(\phi, \theta) + \psi \\
\tilde{\phi}_{\text{next}} &= h(\phi, \theta) + u \\
\psi_{\text{next}} &= \psi + \frac{\partial f}{\partial \phi}(\phi, \theta)^\top \psi + \frac{\partial h}{\partial \phi}(\phi, \theta)^\top u \\
u_{\text{next}} &= (1 - \alpha)u - \frac{\partial L}{\partial y}(\tilde{\phi})^\top
\end{aligned}
\tag{151}
$$

We use the first-order updates discussed in Eq. 147 for training the DEQs, with $\tilde{\psi}_* = u_*$.

### S6.5 Additional experimental details

Here, we provide additional details on the experiments introduced in Section 3.1, including the architecture of the networks, the used hyperparameters, possible data preprocessing and the used optimization algorithms.

### S6.5.1 Training feedforward networks and equilibrium RNNs on MNIST

We evaluate the least-control principle combined with the various feedback circuits introduced in Section 3.1 on the MNIST digit classification task [52], and compare their performance to recurrent backpropagation (RBP).

**Data preprocessing and augmentation.** No data preprocessing or augmentation was employed in our experiment.

**Feedforward network architecture.** We use two hidden layers of 256 neurons to have the same amount of parameters as the equilibrium RNN architecture used for MNIST. In the hidden layers, we use $\tanh$ nonlinearities and on the output layer a softmax nonlinearity.

**RNN architecture.** We use 256 recurrent units in the RNN used for the results of table 1. We include a trained decoder $\tilde{D}$ inside the recurrent layer, as detailed in Section S6.1 and Eq. 109. For the results shown in Fig. 2, we run the equilibrium RNN on MNIST with $[23, 32, 45, 64, 90, 128, 181, 256]$ recurrent units. We use the $\tanh$ nonlinearity for all recurrent units, and include bias terms according to Eq. 110.

**Optimization.** For the classification benchmark, we train the models for 30 of the MNIST train set, with a minibatch size of 64 and with the Adam optimizer [88]. When we learn the feedback weights $S$ with Kolen-Pollack learning, we apply the weight decay after every gradient update, i.e. we apply it outside of the optimizer.

For training the feedback weights $Q$ of the linear feedback controller, we run the stochastic dynamics in Eq. 138 for `max_steps` timesteps, and we update $Q$ following Eq. 139 by averaging $\Delta Q$ over the minibatch. For the simulation of the stochastic dynamics, we use $\Delta t = \tau'^2$, $\tau = \tau'^2$, $\tau_\epsilon = \tau'$ and $\tau_u = 1$ for some hyperparameter $\tau'$. Table S2 contains details on the used and scanned hyperparameters.

**Energy-based dynamics** LCP-EBD models were trained by optimizing the energy for `max_steps` steps using the Adam optimizer [88] with learning rate `ebd_inner_lr`. Due to computational efficiency reasons, we initialized $\phi$ at the free-dynamic equilibrium before the energy optimization for the RNN experiments. For the feedforward networks, we initialized $\phi$ at zero. As initializing the network to its free equilibrium results effectively in a two-phase algorithm, we investigated

whether the same fixed points are reached when initializing at the free equilibrium or at a zero. We observed that both initializations converge to the same fixed point, but that the zero initialization needs many more iterations to reach the fixed point, compared to the combined free-phase iterations and controlled dynamics iterations starting from the free equilibrium.

For the overparametrization experiments, we use the same hyperparameters except for a learning rate of $0.001$ for both RBP and LCP-DI, with a cosine annealing scheduler [106] annealing the learning rate down to $0.0001$ over a total training time of 50 epochs.

For all RNN experiments, we use gradient clipping whereby the norm of the gradient, computed over all gradients concatenated into a single vector, is normalized to some value (chosen to be 10) whenever above it. The gradient clipping was particularly crucial for a stable training of the RBP model.

**Additional results.** In table S3 we show additional experimental results for the MNIST experiments. Note that the hyperparameter was chosen to maximise the test set accuracy.

**Robustness to $\alpha$.** While our theory characterizes the strong control limit $\alpha \to 0$, some interesting properties still hold outside this regime. In Proposition S9, we showed that under a strict alignment condition on the feedback mapping $Q$ (Eq. 76), the controlled steady state is a local minimizer of the augmented energy $F(\phi, \theta, \alpha) = ||f(\phi, \theta)||^2 + \alpha^{-1} L(\phi)$. This strict alignment condition holds for energy-based (Eq. 10) and inversion (Eq. 8) dynamics. This has two consequences. First, the update prescribed by Theorem 1 follows the gradient of some surrogate objective function (cf. the end of Section S2.6) which is closely related to the least-control objective (cf. Remark 1 in Section S2.5). Additionally, this surrogate objective (for $\alpha \neq 0$) is still a meaningful one for the original learning problem: similar results to Propositions 3 and 4 hold for this objective (c.f. Section S2). However, the feedback mapping flexibility of Theorem 2 does not hold anymore outside the strong control limit $\alpha \to 0$, as the column space condition of Eq. 7 is now replaced by the strict alignment condition of Eq. 76.

To corroborate these theoretical insights and to verify whether the least-control principle exhibits robust performance for a wide range of values of $\alpha$, we repeat the MNIST experiments for LCP-DI with a feedforward neural network for varying values of the controller leakage $\alpha$. Note that for $\alpha = 0$, a softmax output layer combined with one-hot targets would lead to an infinite control, as a softmax needs an infinite input to produce a one-hot output. This can be remedied by either increasing $\alpha$ or by using *soft targets*, i.e. taking $1 - \epsilon$ as target for the correct class, and $\epsilon/(N-1)$ for the other $N-1$ classes. As we want to include $\alpha = 0$ in this line of experiments, we resort here to soft targets with $\epsilon = 0.01$, whereas for the experiments in the main text we increase the controller leakage to $\alpha = 0.1$ (c.f. Table S2). Table S1 shows that the performance of LCP-DI is robust to a wide range of values for $\alpha$.

|  | Test accuracy |
| --- | --- |
| $\alpha = 0$ | 98.09 % |
| $\alpha = 0.01$ | 98.04 % |
| $\alpha = 0.1$ | 98.10 % |
| $\alpha = 1$ | 97.99 % |

Table S1: Test set accuracy on MNIST for a feedforward network (2x256 neurons) trained by LCP-DI with different values for the controller leakage $\alpha$. We used the hyperparameters mentioned in Table S2, combined with forward Euler integration of the dynamical equations with timestep $\texttt{dt} = 0.01$.

| Hyperparameter | RBP | LCP-DI | LCP-DI (KP) | LCP-LF | LCP-EBD |
|---|---|---|---|---|---|
| num_epochs | 30 | 30 | 30 | 30 | 30 |
| batch_size | 64 | 64 | 64 | 64 | 64 |
| $\alpha$ | - | 0.1 | 0.1 | 0.1 | 0.1 |
| lr | $\{3.10^{-4}, \mathbf{10^{-3}}, 3.10^{-3}\}$ | $\{3.10^{-4}, \mathbf{10^{-3}}, 3.10^{-3}\}$ | $\{3.10^{-4}, \mathbf{10^{-3}}, 3.10^{-3}\}$ | $\{3.10^{-4}, \mathbf{10^{-3}}, 3.10^{-3}\}$ | $\{3.10^{-4}, \mathbf{10^{-3}}, 3.10^{-3}\}$ |
| optimizer | ADAM | ADAM | ADAM | ADAM | ADAM |
| scheduler | $\{none, \mathbf{cos}\}$ | $\{none, \mathbf{cos}\}$ | $\{none, \mathbf{cos}\}$ | $\{none, \mathbf{cos}\}$ | $\{none, \mathbf{cos}\}$ |
| max_steps | $\{50, \mathbf{200}, 800\}$ | $\{50, 200, \mathbf{800}\}$ | $\{200, \mathbf{800}\}$ | $\{200, \mathbf{800}\}$ | 800 (FF) / 200 (RNN) |
| $\epsilon$ | $\{10^{-5}, \mathbf{10^{-4}}, 10^{-3}\}$ | $\{\mathbf{10^{-6}}, 10^{-5}, 10^{-4}\}$ | $\{\mathbf{10^{-6}}, 10^{-5}, 10^{-4}\}$ $\{\mathbf{10^{-6}}, 10^{-5}, 10^{-4}, 10^{-3}\}$ | $\{\mathbf{10^{-6}}, 10^{-5}, 10^{-4}\}$ | - |
| $\gamma$ | - | - | - | - | - |
| $s$ | - | - | - | $\{\mathbf{0.01}, 0.1\}$ | - |
| $\tau'$ | - | - | - | $\{0.1, \mathbf{0.2}\}$ | - |
| $\tau_Q$ | - | - | - | $\{100, 1000, 10000, \mathbf{100000}\}$ | - |
| $\gamma_Q$ | - | - | - | $\{10^{-4}, 10^{-3}, \mathbf{10^{-2}}, 0.1\}$ | - |
| ebd_inner_optimizer | - | - | - | - | ADAM |
| ebd_inner_lr | - | - | - | - | 0.01 (FF) / 0.001 (RNN) |

Table S2: Hyperparameter search space for the MNIST Feed forward and RNN experiment. A grid search with one seed for each configuration was done to find the best parameters, which are marked in bold. The same set of hyperparameters for both RNN and feedforward models yielded the best performance, except for the LCP-EBD model where the RNN model needed shorter inner optimization steps due the free-phase initialization.

|           | Train set | | Test set | |
|-----------|-----------|----------|----------|----------|
|           | NLL | Accuracy | NLL | Accuracy |
| FF-LCP-LF | 0.00087 ± 0.00084 | 99.98 ± 0.03 | 0.1910 ± 0.0055 | 97.73 ± 0.07 |
| FF-LCP | 0.00004 ± 0.00002 | 100.00 ± 0.00 | 0.1869 ± 0.0154 | 98.11 ± 0.07 |
| FF-LCP-KP | 0.00003 ± 0.00002 | 100.00 ± 0.00 | 0.2041 ± 0.0173 | 98.14 ± 0.09 |
| FF-LCP-EBD | 0.02645 ± 0.00183 | 99.90 ± 0.00 | 0.0848 ± 0.0020 | 98.00 ± 0.03 |
| FF-BP | 0.00002 ± 0.00001 | 100.00 ± 0.00 | 0.0879 ± 0.0027 | 98.29 ± 0.14 |
| RNN-LCP-LF | 0.0021 ± 0.0019 | 99.96 ± 0.04 | 0.1917 ± 0.0068 | 97.70 ± 0.11 |
| RNN-LCP | 0.0121 ± 0.0013 | 99.80 ± 0.02 | 0.1630 ± 0.0007 | 97.58 ± 0.12 |
| RNN-LCP-KP | 0.0105 ± 0.0011 | 99.84 ± 0.04 | 0.1739 ± 0.0012 | 97.75 ± 0.11 |
| RNN-LCP-EBD | 0.0103 ± 0.0026 | 99.74 ± 0.05 | 0.1324 ± 0.0079 | 97.60 ± 0.15 |
| RNN-RBP | 0.0036 ± 0.0047 | 99.94 ± 0.09 | 0.0961 ± 0.0022 | 97.87 ± 0.19 |

Table S3: Classification accuracy (%) and negative log likelihood (NLL) on the MNIST train and test set, of a Feed forward (FF) and RNN model trained by backpropagation (BP/RBP) and least control (LCP). The reported mean±std is computed over 3 seeds.

### S6.5.2 Training convolutional networks and deep equilibrium models on CIFAR-10

Here, we train a convolutional network and Deep Equilibrium model and evaluate their performances on the CIFAR-10 image classification benchmarks [60]. We compare the performance of the models trained via either RBP or LCP.

**Data preprocessing and augmentation.** We normalize all images channel-wise by subtracting the mean and dividing by the standard deviation, which are computed on the training dataset. We employed no data augmentation in our experiments.

**Convolutional network architecture.** We use a convolutional network with 3 convolutional layers of kernel 5-by-5 and channel sizes 96-128-256 followed by two hidden fully connected layers of 2048 neurons. All layers are followed by ReLU non linearity. We do not use batch normalization (for simplicity, we dispense with normalization layers altogether). To keep the architecture as simple as possible we do not use max-pooling units except after the first convolutional layer, and resort for all other convolutional layers to simple strided convolutions to implement downsampling.

**Deep equilibrium model architecture.** The model consists of 3 sub-modules that are invoked sequentially: an encoder, an implicit module, and a decoder. The encoder consists in a single convolutional layer with bias, followed by a Batchnorm layer [107]. The implicit module finds the fixed point of the following equation:

$$f(\phi, e, \theta) = -\phi + \text{norm}(\text{ReLU}(\phi + \text{norm}(e + c_2(\text{norm}(\text{ReLU}(c_1(\phi))))))) \qquad (152)$$

where both $c_1, c_2$ are convolutional layers, norm are group normalization layers [108], and $e$ is the output of the encoder module $g(x, \theta)$. Finally, the decoder module contains a batchnorm layer, followed by an average-pooling layer of kernel size 8-by-8, and a final linear classification layer. All convolutional layers use 48 channels, a filter size of 3-by-3, padding of 1, and are without biases unless specified otherwise.

**Optimization.** We train the model by finding the equilibrium of the controlled dynamics using the fixed point iterations of Eq. 151 and we update the parameters with the first order updates of Eq. 147. Note that the implicit module contains multiple convolutional layers, hence the partial derivatives in Eq. 147 backpropagate the credit assignment signal $\psi$ towards the parameters of the different computational layers. We use automatic differentiation for this. If local parameter updates in space are a design requirement, one can augment the system state $\phi$ with additional states for the output of each convolutional layer, resulting in dedicated control signals for each computational layer, similarly to Section S6.4.3 where we augment the state $\phi$ with output neurons to incorporate the decoder. Both RBP and LCP-DI are trained on 75 epochs for the Deep Equilibrium model, while the Convolutional network is trained on 30 epochs. For both models, we use the stochastic gradient descent optimizer with a minibatch size of 64, combined with an annealing learning rate using the cosine scheduler [106] which divides the learning rate by 10 over the training epochs. We used a controller leakage of $\alpha = 1$ to enable a faster convergence to the controlled equilibrium and hence reduce the computational requirements. Table S4 contains further details on the hyper parameters.

| Hyperparameter | CONV-BP | CONV-LCP-DI | DEQ-RBP | DEQ-LCP-DI |
|---|---|---|---|---|
| num_epochs | 30 | 30 | 75 | 75 |
| batch_size | 64 | 64 | 64 | 64 |
| $\alpha$ | - | 0.1 | - | 3 |
| lr | 0.1 | 0.1 | 0.03 | 0.05 |
| optimizer | SGD | SGD | SGD | SGD |
| scheduler | cos | cos | cos | cos |
| max_steps | - | 800 | 200 | 800 |
| $\epsilon$ | - | $10^{-6}$ | $10^{-4}$ | $10^{-4}$ |
| Gradient clipping | No | No | Yes | Yes |

Table S4: Hyperparameters used for the CIFAR-10 experiment, for the convolutional (CONV) and deep equilibrium model (DEQ) for backpropagation (-BP), recurrent backpropagation (-RBP) and LCP-DI training algorithms .

| | Train set | | Test set | |
|---|---|---|---|---|
| | NLL | Accuracy | NLL | Accuracy |
| DEQ-RBP | 0.2403±0.0027 | 92.50±0.14 | 0.6172±0.0097 | 80.14±0.20 |
| DEQ-LCP | 0.3218±0.0062 | 89.55±0.17 | 0.5874±0.0077 | 80.26±0.17 |

Table S5: Classification accuracy (%) and negative log likelihood (NLL) on the CIFAR-10 train and test set, of a DEQ model trained by recurrent backpropagation (RBP) and least control (LCP). The reported mean±std is computed over 3 seeds.

**Additional results.** In table S5 we show additional experimental results for the CIFAR-10 experiments. Note that the hyperparameter was chosen to maximise the test set accuracy. As can be seen, the training loss is not close to 0, indicating that the model considered here is not overparametrized enough for perfectly fitting the training set. Yet, LCP still manages to optimize for the training loss, and crucially, in such a way that generalizes competitively.

### S6.5.3 Neural implicit representations

Inspired by the recent success of using implicit layers for implicit neural representations [10], we investigate using the least-control principle in this setup for learning. Implicit neural representations (INRs) are a quickly emerging field in computer vision and beyond with the aim to represent any signal through a continuous and differentiable neural network. Here we strictly follow the experimental setup of Huang et al. [10] and leverage the continuity of implicit neural representations to generalize on trained images. In particular, given an Image $I$, we train the INR to represent 25% of the pixels (we simply jump over every second pixel and every second row) while testing the networks prediction on 25% of unknown pixels and measuring the peak signal-to-noise ratio (PSNR). This can be regarded as simple image inpainting.

**Architecture.** Compared to Huang et al. [10], we slightly modify the implicit layer resulting in the following free dynamics

$$f_{\text{INR}}(\phi; x) = -\phi + \text{ReLU}(W\phi) + Ux. \tag{153}$$

We include a trained decoder matrix $\tilde{D}$ inside the implicit layer, as discussed in Section S6.1. We use fixed point iterations on the inversion dynamics (151) for finding an optimal control, and use the weight updates for the equilibrium RNNs (117). Note that we did not include spectral normalization or a learning rate scheduler to train our models as in Huang et al. [10]. Despite these simplifications we obtain comparable results across methods and models.

**Hyperparameters.** Note the small discrepancy between the INR models trained with RBP and LCP-DI in Table S7. Interestingly, we had an easier time scanning hyperparameters to obtain competitive performance and stable forward dynamics when using LCP-DI compared to RBP, for which we had to include gradient clipping and weight decay. See Table S7 for the hyperparameters found when repeating the the grid search identical to the one seen in Table S2. Following Huang et al. [10], we

|  | Natural | | Text | |
|---|---|---|---|---|
|  | 1L-256D | 1L-512D | 1L-256D | 1L-512D |
| SIREN (input inj.) | 22.88 ± 3.0 | 24.52 ± 3.28 | 24.54 ± 2.19 | 25.69 ± 2.18 |
| iSIREN | 24.28 ± 3.37 | 24.92 ± 3.58 | 26.06 ± 2.18 | 26.81 ± 2.09 |
| INR-RBP | 24.72 ± 3.90 | 25.47 ± 4.16 | 25.34 ± 1.67 | 26.53 ± 2.40 |
| INR-LCP | 24.25 ± 2.72 | 25.11 ± 2.90 | 26.71 ± 2.40 | 27.53 ± 2.00 |

Table S6: Peak signal-to-noise ratio (PSNR; in dB) for all models on the natural image and text generalization task [10]. The reported mean ± std is taken over the individual PSNR of the 16 images. SIREN and iSIREN results are taken from Huang et al. [10]. Note that while our INR model is trained without spectral normalization, we reach competitive results. Interestingly, LCP does perform on par with RBP while not relying on weight-decay, learning-rate scheduling or gradient clipping.

| Hyperparameter | RBF-Natural | LCP-Natural | RBF-Text | LCP-Text |
|---|---|---|---|---|
| iteration per image | 500 | 500 | 500 | 500 |
| $\alpha$ | - | 0 | - | 0 |
| lr | 0.0001 | 0.0001 | 0.0001 | 0.0001 |
| optimizer | Adam | Adam | Adam | Adam |
| max_steps | 50 | 50 | 50 | 50 |
| $\epsilon$ | $10^{-5}$ | $10^{-5}$ | $10^{-5}$ | $10^{-5}$ |
| weight decay | 0.1 | 0.1 | 0.1 | 0.1 |
| Omega | 15 | 35 | 35 | 50 |
| Gradient clipping | Yes | No | Yes | No |

Table S7: Hyperparameter used for the INR experiments.

include an extra hyperparameter (`Omega`), which is a scalar multiplier applied to all neural activity, and we scale the initialization of the weights accordingly with $1/$`Omega`.

**Additional results.** Table S6 shows the full experimental results for learning neural implicit representations on the natural images and text dataset introduced by Huang et al. [10]. As we changed the network architecture slightly, we include the SIREN and iSIREN results of Huang et al. [10] for comparison. We see that the least-control principle achieves competitive performance on this task, compared to RBP and the results of Huang et al. [10].

### S6.6    Computational cost

Here, we include a computational comparison between LCP-DI and RBP on the MNIST experiments for equilibrium RNNs. The table below shows the amount of iterations (phase length), the computation time for one single iteration (silicon time; obtained by using the NVIDIA/PyProf profiler offline), and the total training time for both algorithms. Note that RBP has two phases, whereas LCP only one. We see that LCP requires more iterations compared to the combined two phases of RBP. Furthermore, RBP requires significantly fewer computations per iteration, as LCP needs to compute both the system states and control states every iteration, resulting in more matrix-vector products. Consequently, LCP requires significantly more compute compared to RBP on this line of experiments, when simulating on standard digital hardware. Hence, despite its desirable single-phase property, the added computational cost per iteration results in a significantly higher computational cost of LCP. We observed that increasing the controller leakage $\alpha$ decreases the number of iterations required for reaching the controlled equilibrium significantly, and hence also decreases the computational cost.

### S6.7    Related work

Meulemans et al. [30] introduced an approach for training feedforward neural networks with local update rules by minimizing control. The mechanistic implementation of this method can be considered as a specific instance of the least-control principle applied to feedforward neural networks, where we use direct linear feedback connectivity from an output controller to the network to approximate an optimal control (c.f. Section S6.3). Importantly however, the least-control principle is the first to

|                                                       | LCP-DI  | RBP   |
| ----------------------------------------------------- | ------- | ----- |
| Average 1st phase length                              | 483     | 77    |
| Average silicon time per iteration (1nd phase) (ns)   | 221634  | 28735 |
| Average 2nd phase length                              | -       | 72    |
| Average silicon time per iteration (2nd phase) (ns)   | -       | 32479 |
| Total training time (s)                               | 35280   | 1683  |

Table S8: Computational cost

put the idea of learning by minimizing control on a strong and general theoretical foundation, as the theory of Meulemans et al. [30] contains a problematic assumption.

When computing the gradient of the surrogate loss $\mathcal{H}$ w.r.t. $\theta$ with the implicit function theorem, Meulemans et al. 2022 consider the direct feedback weights $Q$ to be independent of $\theta$. However, after taking an update, the feedback weights need to change to satisfy the column space condition (Meulemans et al. 2022, Condition 1) again for the new parameter setting $\theta$. Hence, the loss $\mathcal{H} = \|Qu\|^2$ is affected by this change in Q, which was not incorporated in the gradient calculation, making Strong-DFC only a heuristically-derived method. We were not able to incorporate a $\theta$-and $\phi$ dependent feedback mapping $Q$ into the theoretical framework following the methods of Meulemans et al. 2022, using the implicit function theorem alone. We could only resolve these fundamental issues by formulating least-control as an optimal control problem.

Furthermore, the least-control principle provides a way to learn general dynamical systems that reach an equilibrium. These systems perform computations both in space and time and cannot in general be reduced to a feedforward computational graph. Consequently, the least-control principle provides credit assignment both in space and time, significantly widening the range of systems that can be learned compared to Meulemans et al. [30].

Finally, we introduce various different controller designs compatible with the least-control principle, that move beyond the direct linear feedback controller introduced by Meulemans et al. [30]. Importantly, the inversion dynamics of Eq. 8 and the energy-based dynamics of Eq. 10 satisfy the column space condition of Theorem 2 exactly and thereby lead to gradient-following updates, in contrast to the direct linear feedback that cannot satisfy the column space condition perfectly and hence results in only approximate gradient updates.

## S7 The least-control principle for meta-learning

In Section 3.2, we consider a meta-learning problem in which the objective is to improve the performance of a learning algorithm as new tasks are encountered. To do so, we equip a neural network with fast weights $\phi$ that are learned on each task, and slow parameters $\theta = \{\omega, \lambda\}$ that parametrize the learning algorithm. The fast weights $\phi$ quickly learn how to solve a task $\tau$ and $\lambda$ determines how strongly $\phi$ is pulled towards the slow weights $\omega$, which consolidate the knowledge that has been previously acquired. The capabilities of the learning algorithm are then measured by evaluating $L_\tau^{\text{eval}}(\phi_\tau^*)$, the performance of the learned neural network on data from task $\tau$ that was held out during learning. Meta-learning finally corresponds to improving the learning performance on many tasks by adjusting the meta-parameters $\theta = \{\omega, \lambda\}$. This can be formalized through the following bilevel optimization problem [35, 68]:

$$\min_\theta \mathbb{E}_\tau \left[ L_\tau^{\text{eval}}(\phi_\tau^*) \right] \quad \text{s.t.} \quad \phi_\tau^* = \arg\min_\phi L_\tau^{\text{learn}}(\phi) + \sum_i \frac{\lambda_i}{2} (\phi_i - \omega_i)^2. \tag{154}$$

Note that compared to Eq. 16 in the main text, we now consider the more general setting in which each synapse $i$ has its own attraction strength $\lambda_i$ and meta-learn them.

We now derive the update rules prescribed by the least-control principle, review the meta-learning algorithms we are comparing against and detail our experimental setup.

### S7.1 Derivation of the meta-parameter updates

The first step of the derivation is to replace the $\arg\min$ in Eq. 154 by the associated stationarity condition:

$$\nabla_\phi L_\tau^{\text{learn}}(\phi_\tau^*) + \lambda(\phi_\tau^* - \omega) = 0. \tag{155}$$

This equation can be obtained from a dynamical perspective: let us assume that our learning algorithm is gradient descent, so that

$$\dot\phi = f(\phi, \theta) = -\nabla_\phi L_\tau^{\text{learn}}(\phi) - \lambda(\phi - \omega). \tag{156}$$

The stationarity conditions therefore correspond to $f(\phi_\tau^*, \theta) = 0$ and we can apply the least control principle.

To do so, we run the controlled dynamics

$$\dot\phi = -\nabla_\phi L_\tau^{\text{learn}}(\phi) - \lambda(\phi - \omega) + u, \quad \text{and} \quad \dot u = -\alpha u - \nabla_\phi L_\tau^{\text{eval}}(\phi), \tag{157}$$

and note $\phi_*^\tau$ and $u_*^\tau$ the equilibrium. Note that this corresponds to taking $Q = \text{Id}$ and $h(\phi) = \phi$ in the general feedback dynamics of Eq. 6. Theorem 2 therefore guarantees that $\phi_*^\tau$ is an optimally controlled state and $u_*^\tau$ an optimal control in the limit $\alpha \to 0$, as long as $\partial_\phi f(\phi_*^\tau, \theta)$ and $\partial_y^2 L(h(\phi_*^\tau))$ are invertible, and the column space condition

$$\text{col}\,[Q] = \text{row} \left[ \frac{\partial h}{\partial \phi}(\phi_*^\tau) \frac{\partial f}{\partial \phi}(\phi_*^\tau, \theta)^{-1} \right] \tag{158}$$

is satisfied. The column space conditions here rewrites

$$\text{col}[\text{Id}] = \text{row} \left[ \left( \frac{\partial^2 L_\tau^{\text{learn}}}{\partial \phi^2}(\phi_*^\tau) + \lambda\text{Id} \right)^{-1} \right], \tag{159}$$

and is automatically satisfied through the assumption that the Jacobian of $f$ is invertible. To summarize, the controlled dynamics are optimal in the limit $\alpha \to 0$, if $\partial_y^2 L_\tau^{\text{learn}}(\phi_*^\tau) + \lambda\text{Id}$ and $\partial_y^2 L_\tau^{\text{eval}}(\phi_*^\tau)$ are invertible.

We now put ourselves in a regime in which the conditions detailed above are satisfied, so that $\phi_*^\tau = \phi_\star^\tau$ and $u_*^\tau = u_\star^\tau$. Theorem 1 then gives

$$\Delta\theta = -\left( \frac{\mathrm{d}}{\mathrm{d}\theta} \mathcal{H}(\theta) \right)^\top = \frac{\partial f}{\partial \theta}(\phi_*^\tau, \theta)^\top u_*^\tau = -\frac{\partial f}{\partial \theta}(\phi_*^\tau, \theta)^\top f(\phi_*^\tau, \theta), \tag{160}$$

as $\psi = Qu$ is here equal to $u$. A simple calculation gives

$$\frac{\partial f}{\partial \omega}(\phi_*^\tau, \theta) = \mathrm{diag}(\lambda)$$
$$\frac{\partial f}{\partial \lambda}(\phi_*^\tau, \theta) = -\mathrm{diag}(\phi_*^\tau - \omega), \tag{161}$$

where $\mathrm{diag}(x)$ is a diagonal matrix whose diagonal values are the elements of the vector $x$. It further implies that the updates of the parameters are local, as

$$\Delta\omega = \lambda u_*^\tau = \lambda \left( \nabla_\phi L_\tau^{\mathrm{learn}}(\phi_*^\tau) + \lambda(\phi_*^\tau - \omega) \right)$$
$$\Delta\lambda = -(\phi_*^\tau - \omega)u_*^\tau = -(\phi_*^\tau - \omega) \left( \nabla_\phi L_\tau^{\mathrm{learn}}(\phi_*^\tau) + \lambda(\phi_*^\tau - \omega) \right), \tag{162}$$

the multiplications being performed element wise.

## S7.2   Comparison to existing meta-learning algorithms

We here review existing meta-learning rules that we compare against in Section 3.2. We separate those methods in two categories, the one relying on implicit differentiation and the Reptile algorithm [76].

**Implicit differentiation methods.** The first set of methods we are reviewing aims at directly minimizing the bilevel optimization problem (154) using the implicit gradient

$$\frac{\mathrm{d}}{\mathrm{d}\theta} L_\tau^{\mathrm{eval}}(\phi_\tau^*) = \frac{\partial L_\tau^{\mathrm{eval}}}{\partial \phi}(\phi_\tau^*) \left( \frac{\partial^2 L_\tau^{\mathrm{train}}}{\partial \phi^2}(\phi_\tau^*) + \mathrm{diag}(\lambda) \right)^{-1} \frac{\partial^2}{\partial\theta\partial\phi} \left[ \sum_i \frac{\lambda_i}{2}(\phi_{\tau,i}^* - \omega_i)^2 \right]. \tag{163}$$

In practice, this quantity is intractable because of the inverse of the Hessian and has to be estimated. This lead to different algorithms. T1-T2 [75] uses a first-order approximation of the implicit gradient by simply ignoring the inverse Hessian term:

$$\Delta\omega_i^{\mathrm{T1-T2}} = \lambda_i \frac{\partial L_\tau^{\mathrm{eval}}}{\partial \phi_i}(\phi_\tau^*), \quad \text{and} \quad \Delta\lambda_i^{\mathrm{T1-T2}} = -(\phi_{\tau,i}^* - \omega_i)\frac{\partial L_\tau^{\mathrm{eval}}}{\partial \phi_i}(\phi_\tau^*). \tag{164}$$

Alternatively, a better estimate of the gradient can be obtained by iteratively minimizing the quadratic form

$$\delta \mapsto \delta \frac{1}{2} \left( \frac{\partial^2 L_\tau^{\mathrm{train}}}{\partial \phi^2}(\phi_\tau^*) + \mathrm{diag}(\lambda) \right) \delta^\top + \delta \frac{\partial L_\tau^{\mathrm{eval}}}{\partial \phi}(\phi_\tau^*) \tag{165}$$

and updating the meta-parameters with

$$\Delta\omega_i^{\mathrm{Impl}} = -\lambda_i \delta_{\tau,i}^*, \quad \text{and} \quad \Delta\lambda_i^{\mathrm{Impl}} = (\phi_{\tau,i}^* - \omega_i)\delta_{\tau,i}^*, \tag{166}$$

where $\delta_\tau^*$ is the result of the minimization procedure. Using gradient descent to minimize the quadratic form (165) yields the recurrent backpropagation algorithm [33, 34, RBP][109], also known as Neumann series approximation in this context [77], we discussed in more details in Section S3. Gradient descent is a generic algorithm and more specialized optimization algorithms exist for quadratic forms, such as the conjugate gradients one used by Rajeswaran et al. [68, iMAML]. Note that the updates (162), (164) and (166) share some similar structure, but do not incorporate the teaching signal in the same one. The update of T1-T2 does not correspond to any gradient, but the other two do. The estimation of the implicit gradient through Eq. 166 requires computing second derivatives, whereas the other two updates don't need to.

The implicit gradient can be approximated in a totally different way, using the equilibrium propagation theorem [24], as done by the contrastive meta-learning rule [35]. The idea is to first minimize the so-called augmented loss

$$\mathcal{L}_\tau(\phi, \theta, \beta) := L_\tau^{\mathrm{learn}}(\phi) + \sum_i \frac{\lambda_i}{2}(\phi_i - \omega_i)^2 + \beta L_\tau^{\mathrm{eval}}(\phi) \tag{167}$$

for two different values of $\beta$ ($\beta = 0$ and some small positive value, note $\phi_{\tau,0}^*$ and $\phi_{\tau,\beta}^*$ the solutions) and then update the meta-parameters with

$$\Delta\omega_i^{\mathrm{CML}} = \frac{\phi_{\tau,\beta,i}^* - \phi_{\tau,0,i}^*}{\beta}, \quad \text{and} \quad \Delta\lambda_i^{\mathrm{CML}} = -\frac{(\phi_{\tau,\beta,i}^* - \omega_i)^2 - (\phi_{\tau,0,i}^* - \omega_i)^2}{2\beta}. \tag{168}$$

The equilibrium propagation theorem guarantees that those updates will converge to the true gradient when $\beta \to 0$. Both the contrastive meta-learning rule and the rule prescribed by our principle are first-order, as that they don't require computing second derivative. However, the former approximates the implicit gradient when the latter approximates the gradient of the least-control objective $\mathcal{H}$. Interestingly, as we mentioned in Section 2.4, the augmented solution $\phi_\beta^*$ and the controlled equilibrium $\phi_*^\tau$ are both stationary points of the augmented loss $\mathcal{L}_\tau$ (taking $\beta = \alpha^{-1}$ for $\phi_*^\tau$). However, the contrastive meta-learning update (168) is justified in the weak nudging limit ($\beta \to 0$), whereas the update (162) is justified in the perfect control limit ($\alpha = \beta^{-1} \to 0$).

We refer to Zucchet and Sacramento [110] for a more detailed review of those methods.

**Reptile.** The Reptile algorithm [76] meta-learns the initialization of a neural network such that it can adapt to a new task $\tau$ in very few gradient descent steps. Let us denote $\omega$ the parameters of the network at initialization. The Reptile algorithm first minimizes the learning loss $L_\tau^{\text{learn}}(\phi)$ and then update $\omega$ through

$$\Delta\omega^{\text{Reptile}} = \phi_\tau^* - \omega, \tag{169}$$

with $\phi_\tau^*$ the result of the learning loss minimization. Contrary to the methods presented above, this algorithm is only heuristically motivated and does not rely on any theoretical foundations. Still, it performs surprisingly well on many few-shot learning tasks.

## S7.3 Experimental details

We compare the LCP to existing meta-learning and few-shot learning algorithms on the Omniglot few-shot image classification benchmark [78]. We follow the standard training and evaluation used in prior works [68, 74]. We focus on the 20-way 1-shot setting, and only meta-learn $\theta = \{\omega\}$ while choosing a global $\lambda$ as a hyperparameter. The iMAML result in Table 3.2 is taken from Rajeswaran et al. [68] while those of Reptile and FOMAML are taken from Nichol et al. [76].

**Data preprocessing and augmentation.** We resize the Omniglot images to $28 \times 28$ and augment the images by random rotation by multiples of 90 degrees during meta-training.

**Architecture.** For our classifier, we use the same architecture as in previous works [68, 74], which consists in 4 convolutional modules with a $3 \times 3$ convolutions and 64 filters, each followed by a batch normalization layer, a ReLU nonlinearity, and a $2 \times 2$ max-pooling layer. The output of the module is then flattened and fed into a final classification layer followed by a softmax activation.

**Optimization.** We train all models on 100000 tasks, split into meta batches of size 16. Each task consists in a 20-way classification problem with a single example per class provided for the adaptation phase, and 15 examples per class for evaluating the adaptation. For T1-T2, the adaptation consists in 100 gradient steps with learning rate $\eta = 0.03$. For LCP, we use 100 steps of the discretized version of the dynamics (157):

$$
\begin{aligned}
\phi_{\text{next}} &= \phi + \eta \left( -\nabla_\phi L_\tau^{\text{learn}}(\phi) - \lambda(\phi - \omega) + u \right) \\
u_{\text{next}} &= u + \eta \left( -\alpha u - \nabla_\phi L_\tau^{\text{eval}}(\phi) \right),
\end{aligned}
\tag{170}
$$

where $\eta = 0.03$. The Adam optimizer takes the $\Delta\omega$ as input and updates the meta-parameters, with learning rate 0.001. Finally, the models are evaluated on 100 test tasks. See Table S9 for an overview of the hyperparameters.

| Hyperparameter | T1-T2 | LCP |
|---|---|---|
| num_tasks | 100000 | 100000 |
| meta_batch_size | 16 | 16 |
| optimizer_outer | ADAM | ADAM |
| lr_outer | 0.001 | 0.001 |
| steps_inner | 100 | 100 |
| $\eta$ | 0.03 | 0.03 |
| $\alpha$ | - | 0.1 |
| $\lambda$ | 0.001 | 0.001 |

Table S9: Hyperparameter used for the 20-way 1-shot Omniglot experiment.

## Supplementary References

[89] Asen L. Dontchev and R. Tyrrell Rockafellar. *Implicit functions and solution mappings*. Springer, 2009.

[90] Luisa Zintgraf, Kyriacos Shiarlis, Vitaly Kurin, Katja Hofmann, and Shimon Whiteson. Fast context adaptation via meta-learning. In *International Conference on Machine Learning*, 2019.

[91] Kwonjoon Lee, Subhransu Maji, Avinash Ravichandran, and Stefano Soatto. Meta-learning with differentiable convex optimization. In *Proceedings of the IEEE/CVF Conference on Computer Vision and Pattern Recognition*, 2019.

[92] Dominic Zhao, Seijin Kobayashi, João Sacramento, and Johannes von Oswald. Meta-learning via hypernetworks. In *Workshop on Meta-Learning at NeurIPS*, 2020.

[93] Jorge Nocedal and Stephen J. Wright. *Numerical optimization*. Springer, 2006.

[94] Dimitri P. Bertsekas. *Constrained optimization and Lagrange multiplier methods*. Academic Press, 2014.

[95] Karl Friston, James Kilner, and Lee Harrison. A free energy principle for the brain. *Journal of Physiology-Paris*, 100(1-3):70–87, 2006.

[96] Radford M. Neal and Geoffrey E. Hinton. A view of the EM algorithm that justifies incremental, sparse, and other variants. In *Learning in Graphical Models*, pages 355–368. Springer Netherlands, 1998.

[97] Dimitris G. Tzikas, Aristidis C. Likas, and Nikolaos P. Galatsanos. The variational approximation for Bayesian inference. *IEEE Signal Processing Magazine*, 25(6):131–146, 2008.

[98] Rafal Bogacz. A tutorial on the free-energy framework for modelling perception and learning. *Journal of Mathematical Psychology*, 76:198–211, 2017.

[99] Mark S. Goldman, Albert Compte, and Xiao-Jing Wang. Neural integrator models. *Encyclopedia of Neuroscience*, pages 165–178, 2010.

[100] Simo Särkkä and Arno Solin. *Applied stochastic differential equations*. Cambridge University Press, 2019.

[101] Tzon-Tzer Lu and Sheng-Hua Shiou. Inverses of 2 × 2 block matrices. *Computers & Mathematics with Applications*, 43(1):119–129, 2002.

[102] Jeffrey C. Magee and Christine Grienberger. Synaptic plasticity forms and functions. *Annual Review of Neuroscience*, 43:95–117, 2020.

[103] Gordon M. Shepherd. Dendrodendritic synapses: past, present, and future. *Annals of the New York Academy of Sciences*, 1170(1):215–223, 2009.

[104] Guillaume Hennequin, Everton J. Agnes, and Tim P. Vogels. Inhibitory plasticity: balance, control, and codependence. *Annual Review of Neuroscience*, 40:557–579, 2017.

[88] Diederik P. Kingma and Jimmy Ba. Adam: A method for stochastic optimization. In *International Conference on Learning Representations*, 2015.

[106] Ilya Loshchilov and Frank Hutter. SGDR: Stochastic gradient descent with restarts. In *International Conference on Learning Representations*, 2017.

[107] Sergey Ioffe and Christian Szegedy. Batch normalization: accelerating deep network training by reducing internal covariate shift. In *International Conference on Machine Learning*, 2015.

[108] Yuxin Wu and Kaiming He. Group normalization. In *Proceedings of the European Conference on Computer Vision*, 2018.

[109] Renjie Liao, Yuwen Xiong, Ethan Fetaya, Lisa Zhang, KiJung Yoon, Xaq Pitkow, Raquel Urtasun, and Richard Zemel. Reviving and improving recurrent back-propagation. In *International Conference on Machine Learning*, 2018.

[110] Nicolas Zucchet and João Sacramento. Beyond backpropagation: bilevel optimization through implicit differentiation and equilibrium propagation. *Neural Computation*, 34(12), 2022.