# OpenReview forum: "The least-control principle for local learning at equilibrium"
_NeurIPS.cc/2022/Conference — NeurIPS 2022 Accept_

### Official Review · Reviewer_GGtZ · 2022-07-04

**Rating:** 8
**Confidence:** 5
**Soundness:** 3 good
**Presentation:** 3 good
**Contribution:** 3 good

**Summary:**

This paper presents a principled generalization of “Minimizing Control for Credit Assignment with Strong Feedback” (Meulemans et al, 2022) rooted in control theory and applying to any implicit model whose forward pass is defined as an equilibrium state. Following the same route as the aforementioned paper, the proposed approach builds upon the idea that credit assignment can be achieved by minimizing the amount of control needed to reach an equilibrium which minimizes the loss of interest (Eq. 1). This is referred to as the “least control principle” (LCP).

- Subsection 2.1 is dedicated to how to train the recurrent feedforward weights. More precisely, the optimization problem at hand is defined (Eq. 2), as well as the subsequent notion of “optimal control” (Eq. 3). Theorem 1 shows that, assuming such an optimal control, the gradient of the amount of control reads as a relatively simple local learning rule.

- Subsection 2.2 describes how to train the feedback weights. Namely, the control is specified as the linear mapping of a low-pass filtered loss gradient (with respect to the output units) onto each neuron (Eq. 6). Theorem 2 states that the resulting steady state control is optimal if it satisfies a “column space” (CS) condition (which can be remotely regarded as a kind of “weight symmetry” requirement) and in the limit of infinitely slow leaky integral controller ($\alpha \to 0$). The authors consequently prescribe general dynamics for the controller such that, by construction, the steady state control satisfies the CS condition (Eq.8). These dynamics are referred to as “dynamic inversion” (DI).

- In Subsection 2.3, it is shown under which hypothesis least control solutions lead to solutions of the original learning problem (Eq. 1) in Prop. 4. Extra theoretical guarantees as to when the hypothesis of the theory do not hold exactly are provided in the Appendix.
\item In Subsection 2.4, the least control principle is re-casted into a constrained energy minimization problem, and the connection between this reformulation and predictive coding when the implicit function F is implemented by a feedforward network is discussed in the Appendix. This leads to the definition of energy-based control dynamics (Eq. 10).

- In Subsection 3.1, the equilibrium RNN model is introduced (Eq. 11) along with the LCP feedforward weight updates. Different feedback circuitries to implement the control are proposed.

- Subsubsection 3.1.1 proposes a control implemented as a direct linear feedback (LCP-LF). The rule used to train the feedback weights to achieve the CS condition is introduced in Meulemans et al (2022). Subsubsection 3.1.2. proposes one circuitry implementing dynamic inversion (LCP-DI, Eq. 8), another one implementing energy minimization, with only the former being experimentally tested. A more bioplausible implementation of DI with distinct feedback weights learned by Kollen-Pollack (LCP-DI, KP) learning rule is also considered. The resulting three algorithms (LCP-LF, LCP-DI, LCP-DI-KP) are tested on MNIST and achieve comparable accuracy with recurrent backpropagation (RBP). Next, LCP-DI is applied to a more complex deep equilibrium model (described in the Appendix) on CIFAR-10 and implicit neural representation (INR), with again an accuracy comparable to RBP.

- In subsection 3.2, it is shown that LCP can be used as a single phase first-order metalearning technique. The resulting inner-loop parameter updates, as well as the outer-loop metaparameter updates read in a simple fashion and achieve superior performance to other first-order single phase metalearning techniques.

- A thick appendix ($\approx 40$ pages) is provided to support the main mathematical claims, as well as many extra mathematical results.

**Questions:**

Below, I shall refer to your approach as LCP.
- L. 43: I think it’s a bit misleading to say that SDFC deals with “feedforward” architectures while LCP might extend the idea to “equilibrium architectures”: the architectures are also recurrent and converge to equilibrium in this past work! Wouldn’t it be a better way to phrase the difference in this regard (rather than feedforward models VS equilibrium models) that LCP can deal with any recurrent connectivity pattern?

- L.93-99 (main): what is the difference between LCP applied to a implicit function with a feedforward connectivity pattern on the one hand, and SDFC on the other hand? My take is that LCP allows you to have Q to depend on $\theta$ and $\phi$ (as you point out in the appendix), and therefore use the general-purpose algorithm of Eq. 8, while you couldn’t in the SDFC setting since Q is assumed to be constant, but I’m not sure. It would help to have an extra section in the appendix comparing precisely SDFC and LCP in the case of a feedforward connectivity pattern.

- The following point is particularly important. My understanding of the proposed approach is that the control applies to every single neuron of the dynamical system. If this is correct, in some sense, each neuron of the dynamical system receives a direct feedback from the controller – in this regard, it would be no surprise that the learning rule is local in space. However, I fail to see clearly if your learning theory extends to the case of partial control, e.g. only a subset of the neurons of the system are controlled by $\psi$. For instance, imagine an implicit model defined as the steady state of a layered Hopfield network – this falls in the general category of implicit functions you are considering in this paper. Now imagine that only the last layer of the model is controlled. While the remaining neurons are not explicitly acted upon by the controller, they might well be influenced, through recurrent feedback connections, by the output neurons being controlled. The resulting displacement of the neurons might contain some error information (at the expense of time locality though), e.g.
$\hat{\nabla}_{\Phi} \ell \approx \Phi_{controlled} - \Phi_{not controlled}$.
However, your learning rule prescribes a zero weight update for such neurons. What is happening here? I could have missed something here. But if not, I think it should be clearly stated that your theory does not cover this case and that the theory can only handle *direct* control.

- The constrained energy minimization view is very interesting, as well as the discussion about the two proposed models in Eqs. (14) and (15). However you say “we do not expect major differences in practice between the two, as they have the same equilibrium”. Still, it would have been interesting to see the results you obtain, for instance on MNIST, with the energy-based approach to motivate it beyond the theoretical connection. It seems that you have gone only halfway through the energy-based approach.

- I am a bit worried about the CIFAR-10 experiments. L. 1281, you say that the model is made up of an encoder, an implicit module and a decoder. How do you train the encoder $g(x,\theta)$? The encoder is made up of conv layer and batchnorm layer. Unless you: 1/ explicitely augment the state \phi of your model dynamics with the output of the conv layer and that of the BN (the same strategy that you apply to train the decoder) 2/ apply control $\psi$ to these two, I cannot see how you proceed. Also, to run the controller dynamics (Eq. 147), you need to compute the Jacobians $\partial f/\partial \phi$ and $\partial h/\partial\phi$. However f reads as the complex composition of several convolutional and group normalization operations, and g as the composition of batchnorm, average pooling and linear operations. Don’t you need to backprop through f and g and therefore all these complex operations (batch norm, group normalization) to run the controller dynamics? If it is so, it should be clearly stated in the limitations of the paper. Beyond this, the batchnorm operation is not very bioplausible and other more simple (non parametric) normalization schemes have been proposed (like in this paper: https://arxiv.org/abs/2003.01513). If my understanding of the CIFAR-10 experiment is not correct, please provide more details in the Appendix.


**Ethics Review Area:**

["I don’t know"]

**Limitations:**

My overall understanding of this paper is that credit assignment through control minimization (as introduced in Strong Deep Feedback Control) can be casted into an principled control theory-based approach, applying to any implicit model (as opposed to SDFC which only applies to converging RNNs with a feedforward connectivity pattern), which allows for $\phi$ and $\theta$ dependent feedback control weights $Q(\theta, \phi)$. Crucially, this allows to use a novel algorithm to satisfy the column space condition beyond the linear constant feedback matrix $Q$, possibly for more complex models on more complex tasks, as motivated in the main. However, as per my understanding, this new algorithm comes at the expense of having to backprop through the implicit model f and the decoder h, as it seems to be the case in the CIFAR-10 experiments. It's a little bit frustrating because the precise motivation of this new algorithm to train the feedback controller is to train more complex models for which the linear feedback controller might not sufficient for, and while you take care of not using BP through Kollen Pollack kind of mechanisms for the MNIST experiments, it seems to be overlooked in the CIFAR-10 experiments. Moreover, unless I misunderstood something again, it seems to me that the scope of the theory is limited to the case of a *direct* feedback upon *all* the neurons of the dynamical system. If all these remarks are sound -- which again may not be the case if I misunderstood something -- if should be very clearly stated in the limitations section.

**Strengths And Weaknesses:**

Strengths:
- The paper is very well written.
- The theory is solid and all the mathematical content of the paper is accurate. I spent hours checking every single proof of the Appendix.
- An experimental effort has been made to demonstrate that LCP applies to a large body of models and problems (MNIST, CIFAR-10, INR, Omniglot).
- The proposed theory achieves credit assignment in a single phase – the forward pass and the gradient computation occur simultaneously. Such theories are rare in this literature (Payeur et al 2020, Sacramento et al 2018, Meulemans et al 2022).
- The Appendix is extremely detailed and rigorous.

Weaknesses:
- 10 lines, deep in the Appendix of around 40 pages, to compare your approach with the work *closest to yours* is not enough to my eyes. More precisely, the present paper does not build enough explicit comparison with Strong Deep Feedback Control (SDFC) – “Minimizing Control for Credit Assignment with Strong Feedback”, Meulemans et al, 2022 – which is the closest to theirs. The only explicit comparison appears in S6.3.3, L.1094-1103 of the Appendix. In the light of SDFC, the least control principle idea itself no longer appears as much of an original idea – see Eq. 5 of that paper. Also this approach also already achieves locality in space and time. While the authors re-build from scratch all the maths from first principles of constrained optimization in their appendix, I am very surprised they chose not to include as well the detailed theoretical foundations of this past work.
- The scope of the theory is not clear to me. Most importantly, I am not sure whether the theory extends or not to the case where only a subset of the neurons of the dynamical system are controlled. I detail my concern below. However, I might be wrong and I might well change my opinion about this if the authors address this concern accordingly.

- The details of the CIFAR-10 experiments are hidden in the Appendix. I am a bit skeptical about the model choices – I’m afraid there could be some hidden backprop here. I am also disappointed about the 80% performance – for instance, equilibrium propagation along with the Kollen Pollack algorithm can achieve almost 87% test accuracy when applied to a Hopfield network without weight symmetry (Laborieux et al, 2021). I’ll come back to this in details in the questions part.

- Finally, as per the NeurIPS 2020 reviewing guideline (https://nips.cc/Conferences/2020/CallForPapers): "*While theoretically grounded arguments are encouraged, it is counterproductive to add “decorative math” whose primary purpose is to make the submission look more substantial or even intimidating, without adding significant insight*". While I do appreciate a lot the degree of details and rigor of the appendix, I think the authors could have better selected the most important information to put in the Appendix. Also, I'm wondering whether it can be expected that reviewers go through 40 pages of appendix when reviewing multiple papers over a short timeframe.

---

> ### Author Response · Authors · 2022-08-02
> **Answer to reviewer GGtZ (part I)**
>
> We thank the reviewer for the detailed and constructive comments, and for thoroughly checking all theoretical and empirical results of the paper. Below, we address the main questions and concerns raised by the reviewer.
>
> **Comparison with Strong-DFC.** We thank the reviewer for the sharp comments concerning the relation between Strong-DFC (Meulemans et al. 2022) and the least-control principle. The idea of learning feedforward computations by minimizing control indeed originates from Meulemans et al. 2022. However, we would like to stress that our present contribution not only generalizes the idea to arbitrary systems at equilibrium, that are not limited to feedforward neural computations and not even to neural computations, but also puts the idea of 'minimizing control' in a strong theoretical foundation, in contrast to the derivation of Meulemans et al. 2022 which contains a problematic approximation. We elaborate in more detail on these two points below:
> - **Credit assignment for general systems at equilibrium.** The reviewer is indeed correct in pointing out that the model of Meulemans et al. 2022 becomes recurrent when learning, when the output controller is turned on. However, during inference the model is effectively feedforward, and its predictions can be obtained by doing a single feedfoward pass through the network. It is unclear how to extend Strong-DFC beyond this case. It is thus important to differentiate recurrence in the functional computation of the network from recurrence introduced by the controller for credit assignment. By contrast, the least-control principle provides a way to learn general dynamical systems that reach an equilibrium. These systems perform computations both in space and time and cannot in general be reduced to a feedforward computational graph. Consequently, the least-control principle provides credit assignment both in space and time, significantly widening the range of systems that can be learned. We believe this is a fundamental advance over Strong-DFC. When $f(\phi, \theta)$ defines a feedforward network at equilibrium, the mechanistic implementation of Strong-DFC can be considered as a specific instance of the least-control principle, where we use direct linear feedback connectivity from an output controller to the network to approximate an optimal control, although the underlying theory remains different.
> - **A firm theoretical foundation for the 'minimizing control' idea.** When computing the gradient of the surrogate loss $\mathcal{H}$ w.r.t. $\theta$ with the implicit function theorem, Meulemans et al. 2022 consider the feedback weights Q to be independent of $\theta$. However, after taking an update, the feedback weights need to change to satisfy the column space condition (Meulemans et al. 2022, Condition 1) again for the new parameter setting $\theta$. Hence, the loss $\mathcal{H}=||Qu||^2$ is affected by this change in Q, which was not incorporated in the gradient calculation, making Strong-DFC only a heuristically-derived method. We were not able to incorporate a $\theta$ -and $\phi$ dependent feedback mapping $Q$ into the theoretical framework following the methods of Meulemans et al. 2022, using the implicit function theorem alone. We could only resolve these fundamental issues by formulating least-control as an optimal control problem.
>
> We will modify the introduction as well as Section 2 to make these key points distinguishing LCP from the theory behind Strong-DFC very clear. We will also include feedforward network results on MNIST and CIFAR-10 using the dynamic inversion and energy-minimizing dynamics (as described in a separate post below) optimal control methods enabled by the present theory.

---

> > ### Author Response · Authors · 2022-08-02
> > **Answer to reviewer GGtZ (part II)**
> >
> > **Partial control vs complete control.** We thank the reviewer for the insightful and interesting comment on differentiating complete control from partial control. When considering the main formulation of the least-control principle (Section 2.1-2.3), our theory achieves a first-order local update through a complete control $\psi$ that controls each state variable $\phi$ of the system. When making no additional assumptions on the free dynamics $f(\phi, \theta)$, having partial control influencing only a subset of the system state variables $\phi$, will in general not be an optimal control as defined in Eq. 3. Consequently, the partial control cannot be used for computing the gradient of the least-control objective of Eq. 3 in a local manner.
> >
> > Interestingly, however, our work allows seeing the controlled energy-based dynamics (Eq. 10) as a form of partial control: only the output units are directly controlled; the dynamics of the network are then in charge of optimally propagating this control to the rest of (not directly controlled) units. Importantly, we consider the free dynamics now as the gradient flow on the energy $E=||f(\phi, \theta)||^2$, which contains symmetries that make it possible to let credit assignment information flow into the network while only explicitly controlling the output neurons. The credit assignment signals resulting from the optimal activity changes are now measured by $f(\phi_*, \theta)$ instead of a complete controller $\psi$. For example, when considering a feedforward dynamics $f(\phi, \theta)$, the energy-based dynamics result in the predictive coding network of Eq. 100, which uses the weight transpose $W^\top$ inside its dynamics, which can be used to translate credit assignment on the output neurons towards credit assignment signals for the hidden neurons.
> >
> > Applying the insights from the least-control principle to energies beyond the one studied in our paper ($E=||f(\phi, \theta)||^2$) and to contrastive updates is an exciting direction for future work that we are currently exploring. The energy $E=||f(\phi, \theta)||^2$ has the special property that the free dynamics minimizes it to zero, as $f(\phi^*, \theta)=0$ at the free equilibrium. Hence, we have that $\partial_\theta E(\phi^*, \theta) = f(\phi^*, \theta)^\top \partial_\theta f(\phi^*, \theta) = 0$ at the free equilibrium and consequently, the contrastive update $\Delta \theta = -\partial_\theta E(\phi_*, \theta) + \partial_\theta E(\phi^*, \theta)$ is equal to the least-control update $-f(\phi_*, \theta)^\top \partial_\theta f(\phi_*, \theta)$ which is local in time. Not all energies have this special property, hence, in general, we need full contrastive updates to update the parameters of the energy-based model, similar to equilibrium propagation (EP; Scellier & Bengio 2017). The theoretical tools from the least-control principle might now enable obtaining new insights in the strong control limit of $\beta \to \infty$, where the output of the network exactly matches the target, instead of the weak-nudging limit considered in EP.
> >
> > We will include a condensed version in the discussion of the paper, for which we provide a preliminary version below.
> >
> > > Our principle yields a first-order local (in space and time) parameter update by controlling each and every state variable $\phi$ through a dedicated control variable $\psi$, which puts constraints on neural circuit design. This auxiliary variable is analogous to the neuron-specific error signals which drive learning in a well-known class of models broadly falling under the umbrella of predictive coding (Rao & Ballard, 1999; Whittington & Bogacz, 2017; Sacramento et al., 2018), as formally identified in Section 2.4. One possibility to avoid the introduction of such dedicated control circuits is to consider Hopfield-like energies, which enjoy gradient-based learning through local in space (though not in time) contrastive parameter updates in the weak nudging limit ($\beta \to 0$) (Scellier & Bengio, 2017). An important direction for future work is to extend the present theory to such models, aiming at characterizing their behavior in the opposite perfect control limit ($\beta \to \infty$).

---

> > > ### Author Response · Authors · 2022-08-02
> > > **Answer to reviewer GGtZ (part III)**
> > >
> > > **Implementation details CIFAR10 experiments.** We thank the reviewer for the insightful comments on the implementation details of the CIFAR10 experiments. The motivation for the CIFAR10 experiments with deep equilibrium networks is to test our new learning principle in a more challenging task with a more complex network architecture, and to show that our general-purpose dynamic inversion algorithm of Eq. 8 is a scalable single-phase alternative to recurrent backpropagation. The experimental results show indeed that our new learning principle combined with the dynamic inversion algorithm leads to competitive results. In this line of experiments, biological plausibility is not our main focus, which we instead explore in the MNIST experiments and the new CIFAR10 experiments with a feedforward convolutional network (described in a separate point below). The reviewer is correct in that in the current implementation of the CIFAR10 experiments, within the implicit module we backpropagate the credit assignment signal from the control $\psi$ towards the parameters of the module, as now $\partial_\theta f(\phi, \theta)$ is not a local outer product with the presynaptic input. Furthermore, as shown in Eq. 8, the controller dynamics need access to $\partial_\phi f(\phi, \theta)^\top \psi$, which we compute using a Jacobian-vector product using automatic differentiation. Note that despite the backpropagation in space, all the updates remain local in time. We will clarify the motivation for this series of experiments in the main text, and make it more explicit that we backpropagate within each module. Moreover, we will update the supplementary sections S6.4.3 and S6.5.2 with all details on the dynamics and resulting updates, and explicitly mention where there is a need for backpropagating through the modules. As pointed out by the reviewer, we can avoid the spatial backpropagation within the implicit module $f(\phi, \theta)$ by augmenting the network state $\phi$ with extra states representing each layer of computation within the module neurons, which we leave to future work.
> > >
> > > **New experiments on feedforward neural networks for CIFAR10 and MNIST.** Encouraged by the reviewer’s comments, we investigate the least-control principle on CIFAR10 in a more biologically plausible setting, now with a feedforward convolutional neural network where each layer is explicitly represented in the network state $\phi$. We use a convolutional network with 3 convolutional layers of channel sizes 96-128-256 followed by two hidden fully connected layers of 2048 neurons. Following the suggestion of the reviewer, we do not use batch normalization (for simplicity, we dispense with normalization layers altogether). To keep the architecture as simple as possible we do not use max-pooling units either, and resort to simple strided convolutions to implement downsampling. We trained this convolutional network with the inversion dynamics of Eq. 8 (LCP-DI), and for biologically plausibility, we avoid weight transport trough Kolen-Pollack learning of the feedback weights (LCP-DI (KP)). The table below shows that both versions perform competitively to error backpropagation. Note that we use a simple convolutional architecture, as a proof-of-principle that the least-control principle can train more complex architectures compared to fully connected networks in a biologically plausible manner, while remaining competitive to backpropagation. We leave it to future work to scale the models to improve the results further, and to further increase the biological plausibility of the network model, e.g. by incorporating local connectivity instead of convolutions (Pogodin et al. 2021).
> > > |             | Mean   | Std.  |
> > > |-------------|--------|-------|
> > > | LCP-DI      | 77.45% | 0.57% |
> > > | LCP-DI (KP) | 77.08% | 0.46% |
> > > | BP          | 77.58% | 0.14% |
> > >
> > > We also investigate the feedforward architectures on MNIST, for which we use a fully-connected network with two hidden layers of 256 parameters, to have the same amount of learnable parameters as in the equilibrium RNN experiments on MNIST. The table below shows that all variants of the least-control principle reach competitive performance compared to error backpropagation, and that the inversion dynamics (LCP-DI) lead to significantly better performance compared to the controller with direct linear feedback (LCP-LF).
> > > |             | Mean   | Std.  |
> > > |-------------|--------|-------|
> > > | LCP-DI      | 98.30% | 0.08% |
> > > | LCP-DI (KP) | 98.19% | 0.11% |
> > > | LCP-LF      | 98.15% | 0.04% |
> > > | BP          | 98.29% | 0.14% |
> > >
> > > We will include all results in Section 3.

---

> > > > ### Author Response · Authors · 2022-08-02
> > > > **Answer to review GGtZ (part IV)**
> > > >
> > > > **Empirically testing the energy-based dynamics.** As suggested by the reviewer, we empirically tested the energy-based dynamics of Eq. 15 on MNIST, for both an equilibrium RNN and a feedforward architecture, and we will include these results in Section 3.1. Interestingly, we found that the inversion dynamics of Eq. 14 allow for significantly bigger step sizes and hence fewer iterations for reaching the fixed point, compared to the energy-based dynamics. This is an important result for the predictive coding field, as now it is possible to leverage the dynamic inversion fixed point iterations to find the same equilibrium state with lower computational cost, thereby alleviating a current bottleneck in the field.
> > > >
> > > > To speed up the convergence of the energy-based dynamics, we initialize the network state at its non-controlled equilibrium, i.e. the feedforward network activities in a feedforward network (a technique proposed by Whittington & Bogacz, 2017) or the free equilibrium in the RNN, and we use the Adam optimizer (Kingma & Ba, 2015) on the network state $\phi$ instead of the vanilla gradient flow. The table below shows comparable performance of the energy-based dynamics compared to the inversion dynamics of Eq. 14.
> > > > |             | Mean   | Std.  |
> > > > |-------------|--------|-------|
> > > > | Feedforward | 98.39% | 0.07% |
> > > > | RNN         | 97.82% | 0.17% |
> > > >
> > > >
> > > > As initializing the network to its free equilibrium results effectively in a two-phase algorithm, we investigated whether the same fixed points are reached when initializing at the free equilibrium or at a zero. Preliminary results show that both initializations converge to the same fixed point, but that the zero initialization needs many more iterations to reach the fixed point. We will include a detailed investigation of this in the supplementary materials.

---

> > > > > ### Comment · Reviewer_GGtZ · 2022-08-08
> > > > > **Feedback on authors answer (Part IV)**
> > > > >
> > > > > Thanks a lot for the extra experiments.

---

> > > > ### Comment · Reviewer_GGtZ · 2022-08-08
> > > > **Feedback on authors answer (Part III)**
> > > >
> > > > - **Implementation details CIFAR10 experiments**.
> > > >
> > > >    + "*our general-purpose dynamic inversion algorithm of Eq. 8 is a scalable single-phase alternative to recurrent backpropagation*". I understand from the table of results you provided to reviewer 5Kjg that LCP runs $\approx 6$ times slower than RBP on an equilibrium RNN model on MNIST. How do you think this gap scales with the complexity of the architecture?
> > > >
> > > >   + "*We will clarify the motivation for this series of experiments in the main text, and make it more explicit that we backpropagate within each module*": yes please because it's a little bit dubious from reading the main.
> > > >
> > > >   + "*As pointed out by the reviewer, we can avoid the spatial backpropagation within the implicit module by augmenting the network state with extra states representing each layer of computation within the module neurons, which we leave to future work*". Right. Even if you did so, you would have to approximate the transpose Jacobian of each elementary complex transform within the recurrent block, e.g. $\partial_x {\rm normalization}^\top$. Yet this is a common concern for all the algorithms aiming to solve Jacobian computation and weight transport in the feedback pathway.
> > > >
> > > > - **New experiments on feedforward neural networks for CIFAR10 and MNIST**. Thanks for adding these experiments. The first table of results lead to compelling performance of LCP-DI and LCP-KL wrt BP on CIFAR-10, and I truly appreciate the     effort made by the authors. However, while you may match the baseline on a 5 layers-deep architecture, I expect this gap to increase with the depth of the architecture. While these results are of course very promising, I hope to see soon how LCP scales to deeper architectures!

---

> > > > > ### Author Response · Authors · 2022-08-09
> > > > > **Discussion (Part III)**
> > > > >
> > > > > * **Implementation details CIFAR10 experiments.** From the table can be seen that the total amounts of iterations is roughly equal for RBP and LCP on the MNIST experiments. Hence, the increased computational cost is mainly originating from the increased amount of matrix-vector products needed in the iterations for LCP, compared to RBP. As this is a constant ratio independent of the size of the network (LCP does not need to compute bigger matrix-vector products, just more), we expect that the ratio in computational costs will remain roughly the same when scaling up the networks, assuming that the amount of iterations needed for LCP and RBP remains equal. Whether this hypothesis is true is an interesting empirical question. If so, this would be an additional benefit of perfect control over weak nudging, as it becomes increasingly slower to send the teaching signal to the first layers as the network gets deeper (see Laborieux et al. 2021).
> > > > > * **New experiments on feedforward neural networks for CIFAR10 and MNIST.** While the gap between LCP-DI and BP is non-existent on this experiment, there is indeed a small gap between LCP-DI(KP) and BP, which could potentially grow when scaling to more complex tasks, similar to the observations of Kunin, Yanebi, Sagastuy-Brena et al. 2020. It is an exciting future direction to scale LCP to more complex tasks, while using the insights of the mentioned paper and other related work.

---

> > > ### Comment · Reviewer_GGtZ · 2022-08-08
> > > **Feedback on authors answer (Part II)**
> > >
> > > **Partial control vs complete control**. All of this makes perfect sense to me, it's very interesting. In a nutshell: with a feedforward predictive coding kind of energy function and partial control, you would conserve time locality. But whenever you do not use an energy function whose gradient evaluates to zero on the free phase (as it would be the case for a Hopfield kind of energy function), you would lose time locality. Regardless of this, I think that studying LCP in this setting goes towards unifying more theories under the same umbrella theory.
> > >
> > > I would go a little bit further, with some extra questions on future work:
> > >  + You hint at the connection of your work with Sacramento et al (2018) and that of Payeur et al (2020). Would your theory allow to formulate a variational formulation of these theories extending their application beyond feedforward architectures to any implicit model?
> > >  + Connecting to Difference Target Propagation (DTP) : likewise, would your theory allows a variational formulation of DTP applying to any kind of implicit models?
> > >  + Connecting to Contrastive Hebbian Learning (CHL): CHL is (somehow) a "hard" version of EqProp with targets clamped to the output layer ($\beta \to \infty$). What is the link between LCP and CHL?
> > >  + Finally, something I fail to understand: I can see that your theory prescribes how to run the dynamics of $\psi$ to achieve an optimal control (in the sense of the KKT conditions of the original problem as per Eq. 2), and that the resulting update minimizes the objective as per Eq. 1. However, what is your algorithm equivalent to strictly speaking? In other words, when is the update in Eq. 4 exactly equal to the gradient of the original objective of Eq. 1?
> > > + Lastly: while I do understand mathematically speaking and line by line why you need $\alpha \to 0$ to obtain your theoretical guarantees, I have yet to understand deeply the intuition of *why* you can do without $\beta \to 0$. Could you please provide a bit of intuition here?

---

> > > > ### Author Response · Authors · 2022-08-09
> > > > **Discussion (Part II - 1)**
> > > >
> > > > **Partial control vs complete control.** Many thanks for the interesting questions and future work suggestions.
> > > > * **Connecting to Sacramento et al. and Payeur et al.** Similar to equilibrium propagation, Sacramento et al. link their cortical dendritic microcircuits to gradients on the original learning objective in the limit of weak nudging of the output neurons. The least-control principle can be used to characterize the other end of the spectrum of *perfect control*, e.g. when the output neurons are clamped to their target value. When adjusting the network and microcircuit connectivity to a general recurrent one, the least-control principle can be used to describe the training of such networks at equilibrium, with some minimal changes to the microcircuit, as hinted towards in Section S6.4.1. In contrast to Sacramento et al. and our least-control principle, Payeur et al. consider a burst multiplexing model where the feedback input does not influence the forward processing of the network. Hence, it cannot directly be reformulated through the lens of the least-control principle. However, when allowing the feedback input to influence the forward processing, i.e. when the burst probabilities can influence the event rates of the output spike trains, the burst-multiplexing framework is compatible with the least-control principle, and hence can also be extended to equilibrium RNNs (see Section S6.4.1).
> > > >
> > > > * **Connecting to Difference Target Propagation.** Although the controlled equilibrium states $\phi_*$ can be seen as ‘local targets’, our least control principle results in different local targets compared to target propagation and its variants, and has a different mechanism to translate these local targets to parameter updates. As a simple example, consider a linear network that is fully invertible, i.e. has layers of equal sizes. Target propagation propagates the output target backwards through the network by multiplying it with the inverse weight matrices, to create local targets. In LCP terminology, this would be equivalent to only controlling the first hidden layer until the output target is exactly reached. In this case, the controlled equilibrium state $\phi_*$ corresponds to the local targets assigned by target propagation and its variants. However, in general, this is not an optimal control and hence the local targets are not ‘optimal’ through the lens of LCP. Instead, the least-control principle distributes the control over all layers. Besides this difference in the local target assignment, the update rule of target propagation is also different compared to the least-control principle, as target propagation compares the local target with the (uncontrolled) feedforward activity of the layer as a local error signal ($\delta_l = \phi^l_* - \phi_l^*$), whereas the least-control principle already takes into account the change in controlled activity of the previous layers to compute a local error signal ($\delta_l = \phi^l_* - W_l \sigma(\phi^{l-1}_*$). This characteristic of target propagation that it ignores the target activities of other layers is connected to the ‘block-diagonal approximation’ of the Gauss-Newton curvature matrix in Meulemans et al. 2020.
> > > >
> > > > * **Connecting to contrastive hebbian learning.** Equilibrium propagation is applicable to general energy-based models, in the weak-nudging limit $\beta \to 0$. For the special case of equilibrium energies $E=||f(\phi, \theta)||^2$, that for example appear in predictive coding, the least-control principle is the ‘hard version’, considering $\beta \to \infty$. Note that we have to remove the $1/\beta$ factor that would appear with the equilibrium propagation update. Similarly, we are currently working on extending the least-control principle to general energy-based models, where we are interested again in the ‘hard version’ of $\beta \to \infty$ instead of the weak-nudging regime of equilibrium propagation. This would cast the least-control principle as a generalization of contrastive Hebbian learning to more general learning systems and dynamics.

---

> > > > > ### Author Response · Authors · 2022-08-09
> > > > > **Discussion (Part II - 2)**
> > > > >
> > > > > * In general, the gradient of the least-control objective (eq. 3) w.r.t. $\theta$ is never equal to the gradient on the original learning objective (eq. 1) except at the minimum of the least-control objective when the assumptions of Proposition 3 and/or 4 are satisfied, and hence both gradients are zero. Instead, the least-control objective can be intuitively thought of as some bound on the original learning objective (c.f. Proposition 5), such that following gradients on the least-control objective indirectly minimizes the original learning objective (c.f. Section 2.3), without following the exact gradients on the original learning objective. When using an optimal control (formally defined in eq. 3), the updates of Theorem 1 exactly follow the gradient on the least-control objective (eq. 3), and in general not on the original learning objective.
> > > > > * The last question is related to the previous one. Intuitively speaking, gradients are a local sensitivity measure, relating a small change in parameters to a small change in the output loss, assuming a small change in network activity. Hence, to estimate gradients on the original learning objective, it is crucial to consider small changes around the *free equilibrium* $\phi^*$ for the current parameter setting, resulting in a need for *weak nudging* ($\beta \to 0$) in equilibrium propagation. Our aim however is to move away from the weak-nudging setting towards a *perfect control* setting ($\beta \to \infty$, which is equivalent to $\alpha \to 0$ as $\alpha = 1/\beta$), where the feedback control can strongly change the network activity. However, at first sight this is incompatible with gradients, as gradients assume small activity changes. Hence, we define a *surrogate objective* (eq. 3) different from the original learning objective, where the gradients take on a different meaning. Here, the gradients relate a small change in parameters towards a small change in the ‘amount of optimal control’ $\mathcal{H}(\theta)$, assuming small changes in the *controlled equilibrium activity* $\phi_*$. Hence, even though the least-control principle operates with large activity changes due to the strong influence of the controller on the network dynamics, small parameter updates result in small changes in the *controlled equilibrium activity*, making it possible to define gradients outside of the weak-nudging regime of $\beta \to 0$.

---

> > ### Comment · Reviewer_GGtZ · 2022-08-08
> > **Feedback on authors answer (Part I)**
> >
> > Dear authors,
> >
> > Many thanks for your detailed answer and apologies for the delay.
> >
> > **Comparison with Strong DFC**.
> > - *[...] that are not limited to feedforward neural computations and not even to neural computations*. What kind of systems are you thinking about, beyond "neural computation"? Just curious.
> > - **Credit assignment for general systems at equilibrium**.
> >    + Yes, I do understand that these models become feedforward after training because the apical top-down inputs vanish. I was wondering about the best terminology to be used.
> >    + *Strong-DFC can be considered as a specific instance of the least-control principle, where we use direct linear feedback connectivity from an output controller to the network to approximate an optimal control*. Then I started to think and I realized a misunderstanding of mine: I thought your theory only applied to the case where each neuron received *direct feedback* from the output layer, but in fact not! While each neuron are indeed controlled by a variable $\Psi$, only the subset of neurons read out by the classifier (the output layer) receives direct error feedback. So as far as I understood, there are two improvements of LCP over SDFC: 1/ it trains any equilibrium model (assuming you can compute at least a proxy of $\partial_\phi f^\top$ and $\partial_\phi h^\top$) 2/ but most importantly: you no longer need *direct* feedback from the output layer onto each layer of the architecture. So you do achieve spatial credit assignment in a non-trivial fashion, which is powerful. I think that what would make it clearer would be to write the equations governing the dynamics of LFC applied to a feedforward architecture *layer-wise*, where it would be clearly seen that *u* only appears in the dynamics of the output layer. Can you please confirme this? If it is the case, my apologies for the misunderstanding.
> >
> > - **A firm theoretical foundation for the 'minimizing control' idea**. I see. Yet, even for LFC, you still need to learn feedback weights in a heuristic fashion to approximate $\partial_\phi f^\top$ through Kollen-Pollack kind of things.

---

> > > ### Author Response · Authors · 2022-08-09
> > > **Discussion (Part I)**
> > >
> > > Many thanks for the detailed and constructive questions that keep helping us improve the quality of our paper.
> > >
> > > **Comparison with Strong-DFC**
> > > * *’What kind of systems are you thinking about, beyond "neural computation"?’*. One such system is the meta-learning setting we explored, where the dynamics $\dot{\phi}$ does not represent the neural computations anymore, but instead the learning of the network weights. Of course, this setting is still closely linked to neural computations, as we still consider (meta-)learning neural computations. Alternatively, our framework can be applied to physical systems that reach equilibrium and have tunable parameters such as analog hardware or other forms of physical networks (see e.g. Stern et al., Phys. Rev. X, 2021).
> > > * **Credit assignment for general systems at equilibrium.** Indeed, the ‘complete control’ $\psi$ for each neuron that serves as a credit assignment signal is not necessarily a ‘direct control’ from the output error control $u$ towards the rest of the network, but can instead be propagated through the network by internal feedback connections, e.g. through the inversion dynamics of Eq. 8, where only the output neurons receive ‘direct control’ $\partial_\phi h(\phi)^\top u$. We thank the reviewer for the suggestion of clarifying this point with a feedforward network, which we will incorporate in the paper, while also rewriting section 2.2 to clarify that $Q(\phi, \theta) u$ is not necessarily a ‘direct control’.
> > > * **A firm theoretical foundation for the 'minimizing control' idea**. If the design choice is to avoid weight transport, then indeed $\partial_\phi f(\phi, \theta)^{\top}$ needs to be approximately learned through e.g. a separate set of feedback weights. But even in this setting, the separate set of feedback weights (e.g. the direct feedback weights Q in Strong-DFC) still depend on the forward parameters $\theta$ through the feedback weight learning that aims to satisfy the column space condition. Hence, our new firm theoretical foundation for the ‘minimizing control’ idea is needed both for Strong-DFC as well as the new algorithms we propose with the least-control principle.

---

### Official Review · Reviewer_5Kjg · 2022-07-08

**Rating:** 8
**Confidence:** 5
**Soundness:** 4 excellent
**Presentation:** 4 excellent
**Contribution:** 4 excellent

**Summary:**

This work proposes a theoretical framework for optimizing equilibrium neural networks by reformulating the original constrained optimization problem into a least-control problem. The least-control problem is further solved with a two-steps iterative procedure that first drives the networks dynamics with a control signal towards minimizing fixed points verifying the initial problem constraints, then taking a gradient step to minimize the norm of the control terminal cost.

Motivations for such a principle are manifold: First, solutions are shown to coincide with the initial optimization problem under feasible conditions on the control that leave flexibility for its explicit construction. Second, credit assignement and activation dynamics are originally combined in a single dynamic that result in a spatially and temporally local explicit weight update rule. Third, this formulation can also be linked to models of free energy minimization, offering an original perspective on the learning objective operated by such models as a relaxation of the proposed least-control objective.

Several experiment are conducted to showcase the learnability of such system for supervised image classification, implicit neural representation and meta-learning. The technique is shown to have comparative performance to the current leading technique for training equilibrium model recurrent backpropagation (RBP) and can encompass meta-learning with good results compared to gradient-based methods.

**Questions:**

I have three questions, with one regarding the neuroscientific interpretation of the update rule, a second for optimization and a third for the experiments:

- In line 98 is stated that "the resulting weight update now represents a local Hebbian rule multiplying presynaptic input with the postsynaptic control signal." Could you elaborate on this? Form equation (4), the weight update consist in the Jacobian of f at equilibirum multiplying the instantaneous variation of activity driven by f. Hence, the interpretation given by the authors is not straightfoward and does not seem to be local. Could you help clarify this point?

 - Could you relax the assumption of convexity of the function $L$ in proposition 3 to accomodate for locally convex functions? For instance, the meta-learning problem proposed in section 3.2 does not consider a convex loss function with respect to model parameters and the conditions for proposition 3/4 do not hold.

 - Regarding the experiments, can you provide more details regarding the computation cost of training such a system for the considered problems? It seems to me that the fixed point scheme for finding the equilibrium of the controlled dynamics would require careful tuning for finding a trade-off between rapidity and precision of the control estimate, that is not discussed in the empirical section. A temporal comparison with RBP would be welcome to characterize the efficiency of the optimization process.



**Limitations:**

The work does not raise particular ethical or societal question. Conversely, it constitute an interesting direction of work for optimization for deep learning. The framework is however restricted to deep equilibrium model and an important development would be to extend such principle to arbitrary dynamics. Another interesting aspect of this work is the connection drawn with neuroscience-inspired models (predictive-coding and free-energy) that calls for more in-depth comparison.



**Strengths And Weaknesses:**

First, I find this works is nicely written, with a clear and rational development which makes it highly accessible. The technical problem formulation as well as the different connection with previous models are well executed and contributes to appreciating the original contribution of this work. The experiments are diverse and show the general applicability of the method.

I would only note that a lot of the paper technicality is deferred to the supplementary, and while the mathematical development is clear, on the other hand, much of the choices for solving experimentally the least-control problems are not discussed in the main text. I understand that this is a choice for exhaustivity but I find that it undermines slightly the experimental section for demonstrating the value of the least-control formulation. Globally however, I have a very positive opinion on this work as it offers a valuable theoretical contribution to the field.

---

> ### Author Response · Authors · 2022-08-02
> **Answer to reviewer 5Kjg**
>
> We thank the reviewer for his insightful comments and overall excitement for our work. Below, we address the main questions, suggestions and concerns.
>
> **Local updates.** We thank the reviewer for raising this clarification question. We start with the learning rule provided by Theorem 1: $\nabla_\theta \mathcal{H}(\theta) = - \partial_\theta f(\phi_*, \theta)^\top \psi_*$. Following the derivation in Section S6.2 (Eq. 113 - 115), we see that product of this partial derivative with the control signal results in the update $\Delta W = \psi_* \sigma(\phi_*)^\top$, which is a local update to each synapse. For feedforward networks, the weight matrix $W$ has a lower block-diagonal structure (c.f. Section S6.1) with as blocks the weight matrix $W_i$ of layer $i$, which translates to local layerwise Hebbian weight updates $\Delta W_i = \psi_{i,*} \sigma(\phi_{i-1, *})^\top$. We will rewrite the paragraph starting at line 93 to clarify this point without the need to look at the supplementary materials.
>
> **Beyond convexity assumptions for relating the loss.** We thank the reviewer for the interesting suggestion for generalizing Proposition 3 and 4 beyond convex output losses. Propositions 3 and 4 assume the convexity of L on the output units, an assumption that is usually met when training neural networks (e.g. the L2 loss, cross-entropy loss, …) but not for meta-learning, as pointed out by the reviewer. We can relax this assumption by considering local minimizers of the original learning problem instead of global ones. We will change Propositions 3 and 4 accordingly and discuss the convex case in the appendix.
>
> **Clarifying the experimental section.** We will update Section 3 to better reflect the details and motivation behind important design choices.
>
> **Comparison of the computational costs.** Encouraged by the reviewer’s comment, we include a detailed computational comparison between LCP-DI and RBP on the MNIST experiments for equilibrium RNNs. The table below shows the amount of iterations (phase length), the computation time for one single iteration (silicon time; obtained by using the NVIDIA/PyProf profiler offline), and the total training time for both algorithms. Note that RBP has two phases, whereas LCP only one. LCP and RBP require a similar total amount of iterations, however, RBP requires significantly fewer computations per iteration, as LCP needs to compute both the system states and control states every iteration, resulting in more matrix-vector products. Consequently, LCP requires roughly 6 times more compute compared to RBP on this line of experiments, when simulating on standard digital hardware. Hence, despite its desirable single-phase property, the added computational cost per iteration results in a significantly higher computational cost of LCP. We will discuss these results in the limitations paragraph of the discussion, and add the detailed results in the supplementary materials.
> |     | Avr. 1st phase length | 1st phase silicon time (ns) | Avr. 2nd phase length | 2nd phase silicon time (ns) | Total training time (sec) |
> |-----|-----------------------|-----------------------------|-----------------------|-----------------------------|---------------------------|
> | LCP | 139                   | 221634                      | -                     | -                           | 10515 +/- 411             |
> | RBP | 77                    | 28735                       | 72                    | 32479                       | 1683 +/- 360              |

---

> > ### Comment · Reviewer_5Kjg · 2022-08-05
> > **Reply to authors**
> >
> > Thank you for your answers. The computational cost induced by LCP seems also an important aspect to add to the final version in my opinion. Altogether, your rebuttal confirm my opinion about the contribution brought by this paper and I will maintain my score.

---

### Official Review · Reviewer_6XDX · 2022-07-09

**Rating:** 8
**Confidence:** 5
**Soundness:** 4 excellent
**Presentation:** 2 fair
**Contribution:** 4 excellent

**Summary:**

The submission introduces 'the least control principle' a novel paradigm for training neural networks with local learning rules. The learning algorithm is intimately related to predictive coding and energy based models. The authors show that the learning algorithm is flexible by applying it to RNNs and meta-learning tasks.

___

Post rebuttal update: The authors have addressed most of my concerns. I have accordingly modified my score.

**Questions:**

I will gladly increase my score if/when the authors address the weaknesses in the previous section.

**Strengths And Weaknesses:**

**Strengths**
* The submission is overall very interesting and well written. It brings new insight to a field under active investigation.
* The empirical demonstrations showcase the flexibility of the algorithm.
* The relationship to the energy based models and the alternative limit of beta -> infinity is very interesting.

**Minor Weaknesses**
* The evaluation of the algorithm on a well studied and popular benchmark is missing. The experiments on RNN and meta learning are great but because they are less studied examples, it is harder for the average reader to compare performance of the algorithm. Would be good for the authors to demonstrate that they can use their algorithm to train a deep net on a classic vision task like ImageNet (or even CIFAR10 with a feedforward network).
* The discussion of the limitations of the work is very narrow and implies that the method has no limitations beyond 'out-of-equilibrium' networks and explicit parameter dependence. However, for example we know that a major limitation of predictive coding is its practical inefficiency because of its equilibration phase. Doesn't least-control also suffer from the same issue? An honest discussion of practical limitations and compute requirements is missing. The compute details of the experiments is also missing. (Hardware/compute time for the different methods etc.)
* What are the consequences of alpha not being zero?
* Related to above, overall empirical demonstrations of the fate of the theorems in regions that they do not apply is lacking. For example what happens to thm 2, prop 3, prop 4 when alpha is finite or f is not strongly convex as in (11). The experiments showing that learning take place are good but it would be great to empirically show that the theorems/propositions still approximately hold if their conditions are not fully satisfied. Prop. S8 somewhat addresses this but assumes that the approximate state and optimal state are close. Is there a guarantee that this is the case? Might be a good idea overall to show one or two empirical plots exploring these issues.


**Exposition**
The following are suggestions to improve the paper. (in no particular order)
* (Line 114) 'y are at a target value'. This probably only holds when the objective is supervised (e.g. regression/classification). Might be good to clarify this to avoid confusion.
* Sometimes the presentation is a little confusing and requires multiple reads. For example in section 2.2, the paragraphs jump between describing eq. 6 and describing the generality of thm 2. I would suggest the authors reword this paragraph such that the reader does not have to go back and forth as many times. (Pehraps provide the theorem first and then give examples and expand on the specific example.) In general, one would expect the paragraph before or after a theorem or proposition to explain the theorem/proposition in more understandable terms.
* Another example of above is after Eq. 4. The authors alternate between describing their own algorithm and contrasting with backprop. I would suggest first describing eq. 4 in terms of Hebbian learning and then moving on to contrasting it with backprop.
* Might be good to mention 'predictive coding' when referring to the model of Whittington and Bogacz (line 189).
* Would be good to explicitly show the model parameter updates (eq. 4) for the specific examples of section 3. This would make it easier for the reader to see that they are indeed Hebbian.

---

> ### Author Response · Authors · 2022-08-02
> **Answer to reviewer 6XDX (part I)**
>
> We thank the reviewer for the insightful comments and constructive feedback. Below, we address the main questions and concerns raised by the reviewer.
>
> **Training feedforward networks on MNIST and CIFAR10.**
> To enable easier comparison of the least-control principle on conventional benchmarks, we train feedforward networks with different variants of the least-control principle on both MNIST and CIFAR10. We will include these results in Section 3.
>
> For MNIST, we use a fully-connected network with two hidden layers of 256 neurons, to have the same amount of learnable parameters as in the equilibrium RNN experiments on MNIST. The table below shows that all variants of the least-control principle reach competitive performance compared to error backpropagation, and that the inversion dynamics (LCP-DI) lead to significantly better performance compared to the controller with direct linear feedback (LCP-LF). Note that for feedforward neural networks, the mechanistic implementation of LCP-LF is equivalent to Strong-DFC (Meulemans et al. 2022), whereas the underlying supporting theory remains different (see also the related answer in https://openreview.net/forum?id=ttQ_3CiZqd3&noteId=pF8pJtJNVZA).
> |             | Mean   | Std.  |
> |-------------|--------|-------|
> | LCP-DI      | 98.30% | 0.08% |
> | LCP-DI (KP) | 98.19% | 0.11% |
> | LCP-LF      | 98.15% | 0.04% |
> | BP          | 98.29% | 0.14% |
>
> For CIFAR10, we use a convolutional network with 3 convolutional layers of channel sizes 96-128-256 followed by two hidden fully connected layers of 2048 neurons. Following the suggestion of reviewer GGtZ, we do not use batch normalization (for simplicity, we dispense with normalization layers altogether). To keep the architecture as simple as possible we do not use max-pooling units either, and resort to simple strided convolutions to implement downsampling. We trained this convolutional network with the inversion dynamics of Eq. 8 (LCP-DI), and for biologically plausibility, we avoid weight transport through Kolen-Pollack learning of the feedback weights (LCP-DI (KP)). The table below shows that both versions perform competitively to error backpropagation. Note that we use a simple convolutional architecture, as a proof-of-principle that the least-control principle can train more complex architectures compared to fully connected networks in a biologically plausible manner, while remaining competitive to backpropagation. We leave it to future work to scale the models to improve the results further.
> |             | Mean   | Std.  |
> |-------------|--------|-------|
> | LCP-DI      | 77.45% | 0.57% |
> | LCP-DI (KP) | 77.08% | 0.46% |
> | BP          | 77.58% | 0.14% |
>
> **Improved results for training equilibrium RNNs on MNIST.**
> To make the equilibrium RNN results on MNIST more comparable to conventional benchmark results for fully connected feedforward networks, we improved the hyperparameters for all methods for training equilibrium RNNs on MNIST. The table below shows that all methods now get test accuracies close to 98%, while LCP-DI and LCP-DI (KP) significantly improve upon the test accuracy of LCP-LF (>0.15% improvement), indicating that its more accurate optimal control improves learning.
> |             | Mean   | Std.  |
> |-------------|--------|-------|
> | LCP-DI      | 97.93% | 0.16% |
> | LCP-DI (KP) | 97.90% | 0.04% |
> | LCP-LF      | 97.75% | 0.10% |
> | RBP         | 97.87% | 0.19% |

---

> > ### Author Response · Authors · 2022-08-02
> > **Answer to review 6XDX (part II)**
> >
> > **Comparison of the computational costs.**
> > Encouraged by the reviewer’s comment, we include a detailed computational comparison between LCP-DI and RBP on the MNIST experiments for equilibrium RNNs. The table below shows the amount of iterations (phase length), the computation time for one single iteration (silicon time; obtained by using the NVIDIA/PyProf profiler offline), and the total training time for both algorithms. Note that RBP has two phases, whereas LCP only one. LCP and RBP require a similar total amount of iterations, however, RBP requires significantly fewer computations per iteration, as LCP needs to compute both the system states and control states every iteration, resulting in more matrix-vector products. Consequently, LCP requires roughly 6 times more compute compared to RBP on this line of experiments, when simulating on standard digital hardware. Hence, despite its desirable single-phase property, the added computational cost per iteration results in a significantly higher computational cost of LCP. We will discuss these results in the limitations paragraph of the discussion, and add the detailed results in the supplementary materials.
> > |     | Avr. 1st phase length | 1st phase silicon time (ns) | Avr. 2nd phase length | 2nd phase silicon time (ns) | Total training time (sec) |
> > |-----|-----------------------|-----------------------------|-----------------------|-----------------------------|---------------------------|
> > | LCP | 139                   | 221634                      | -                     | -                           | 10515 +/- 411             |
> > | RBP | 77                    | 28735                       | 72                    | 32479                       | 1683 +/- 360              |
> >
> > **Discussion on the limitations.**
> > Following the suggestion of the reviewer and based on the new results on the computational costs of the various methods, we will add a more detailed discussion on the limitations of our work in the main paper, covering the computational costs of our new method. Below we provide a preliminary version of this discussion.
> >
> > In the strong control limit considered in our work, learning signals strongly influence the state of the network, resolving the fragility to noise issues inherent to weak nudging, and enabling flexible feedback cf. Theorem 2. However, an important drawback of this regime is that reaching a controlled state might have a higher computational cost. Indeed, although our algorithms are single-phase, solving for this one phase turns out to be costlier than solving for the two phases of RBP, as revealed by our supplementary experiments.

---

> > > ### Author Response · Authors · 2022-08-02
> > > **Answer to reviewer 6XDX (part III)**
> > >
> > > **Alpha not equal to zero.**
> > > While our theory characterizes the strong control limit $\alpha \rightarrow 0$, some interesting properties still hold outside this regime. In Proposition S9, we show that under a strict alignment condition on the feedback mapping $Q$ (Eq. 76), the controlled steady state is a local minimizer of the augmented energy $F(\phi, \theta, \alpha) = ||f(\phi, \theta)||^2 + \alpha^{-1} L(\phi)$. This strict alignment condition holds for energy-based (Eq. 10) and inversion (Eq. 8) dynamics. This has two consequences. First, the update prescribed by Theorem 1 follows the gradient of some surrogate objective function (cf. the end of Section S2.6) which is closely related to the least-control objective (cf. Remark 1 in Section S2.5). Additionally, this surrogate objective (for $\alpha \neq 0$) is still a meaningful one for the original learning problem: similar results to Propositions 3 and 4 hold for this objective, which we will add to the supplementary materials. However, the feedback mapping flexibility of Theorem 2 does not hold anymore outside the strong control limit $\alpha \rightarrow 0$, as the column space condition of Eq. 7 is now replaced by the strict alignment condition of Eq. 76.
> > >
> > > We ran an additional experiment to test the robustness of our learning rule to non-zero $\alpha$ by running the energy-based dynamics of Eq. 15 (which satisfies the strict alignment condition of Proposition S9) until equilibrium. We trained a feedforward neural network with 1000 hidden neurons on a subset of 10k samples of MNIST (50 epochs, learning rate 0.2, batch size 100) and varied $\alpha$. The results are summarized in the table below:
> > > | $\alpha$ | Train loss ($\times 10^{-3}$) | Train accuracy | Test accuracy |
> > > |----------|-------------------------------|----------------|---------------|
> > > | 0        | 4.09                          | 99.99%         | 96.61%        |
> > > | 0.01     | 3.97                          | 100%           | 96.62%        |
> > > | 0.1      | 3.36                          | 100%           | 96.70%        |
> > > | 1        | 2.10                          | 100%           | 96.65%        |
> > >
> > > This confirms our theoretical results, as modifying $\alpha$ only slightly impacts learning. This experiment is limited as it is a single seed and the feedforward network is trained on a subset of MNIST, which results in lower accuracy on the test set than usually reported for this kind of network. We will add more extensive results (full MNIST, several seeds, recurrent architecture) to the appendix, along with a more thorough discussion on the impact of $\alpha$ at the end of Section S2.6.
> > >
> > > **Experiments on the robustness of the theoretical assumptions.**
> > > Some of our theoretical results require convexity of different functions: Propositions 3 and 4 assume that the loss $L$ is convex on the output units while Propostion 5 requires $||f||^2$ to be strongly convex. We relaxed the convexity assumption of Propositions 3 and 4 motivated by Reviewer 5Kjg’s comments (https://openreview.net/forum?id=ttQ_3CiZqd3&noteId=rnPywoUqrQL). The strong convexity assumption of $||f||^2$ needed in Proposition 5 does not necessarily hold in the settings we consider, as noted by the reviewer. This assumption considerably simplifies the theoretical analysis and should reflect the behavior of $||f||^2$ in the neighborhood of the controlled steady state for non-output units.
> > >
> > > Finally, we comment on the impact of not reaching an equilibrium, which is an assumption that underlies some of our theoretical results (e.g. Theorems 1 and 2, Proposition S9). Proposition S8 links the distance between the estimated update and the desired one, to the error in approximating the steady state. We checked if its behavior holds in practice. We used the same setup as for the $\alpha \neq 0$ discussion, but used a network with 3 hidden layers of 300 neurons. After random initialization of the weights, we looked at the distance in the update along the energy-based dynamics of Eq.15 (right figure of https://ibb.co/xzhLSwt). To do so, we ran those dynamics until equilibrium, and for every point in time we measured the distance to the equilibrium it will reach (x-axis) and the distance between the corresponding update and the desired one (y-axis). The update error scales almost linearly with the distance to equilibrium in state space, which confirms that the theoretical predictions of Proposition S8 qualitatively hold along this trajectory.  Additionally, we perturbed the controlled steady state with random noise (all units except input and output, drawn from $\mathcal{U}(0, 1) \times \mathcal{N}(0, 0.5)$, 1000 samples) and plotted the results on the left figure. Each point corresponds to a random sample. The linear behavior of Proposition S8 also holds here.

---

> > > > ### Author Response · Authors · 2022-08-02
> > > > **Answer to reviewer 6XDX (part IV)**
> > > >
> > > > **Writing suggestions.** We thank the reviewer for the helpful exposition suggestions. We will adjust section 2.1 to improve the explanation of Theorem 1 and link it to Hebbian updates on the one hand, and to compare to the implicit gradient and recurrent backpropagation (Eq. 5) on the other hand. We will update Section 2.2 to better structure the explanation of Theorem 2 and expanding on the flexibility of the theorem, and we will incorporate the remaining smaller suggestions in the relevant sections.

---

> > > > > ### Comment · Reviewer_6XDX · 2022-08-05
> > > > > **Rebuttal response**
> > > > >
> > > > > I thank the authors for their responses. They have addressed most of my concerns. I have reevaluated my score accordingly.

---

### Official Review · Reviewer_PoYW · 2022-07-11

**Rating:** 6
**Confidence:** 2
**Soundness:** 2 fair
**Presentation:** 2 fair
**Contribution:** 3 good

**Summary:**

This paper proposes use of the least-control principle which minimizes the amount of optimal control needed at equilibrium as a local alternative to backpropagation in recurrent neural networks. The authors establish conditions that should be met at equilibria, relate what is optimized back to the standard optimization problem, and relate their framework to constrained energy minimization. Finally, the approach is validated to get similar performance to backpropagation on a supervised learning settings based on MNIST, CIFAR-10, and a natural image dataset as well as in meta-learning experiments on Omniglot.

**Questions:**

- Could you provide an intuitive explanation of what it means to minimize the "amount of optimal control needed at equilibrium"? (i.e. from Figure 1)
- Can you provide a clear set of advantage and disadvantages of the proposed framework with respect to the recurrent backpropagation baseline?
- Is it possible to create an experiment that could empirically highlight those advantages or is it beyond the scope of available computation?
- Can you provide more clarity about the comparison between you approach and iMAML? I was wondering why you only included iMAML for comparison rather than normal MAML as well?

**Limitations:**

The authors can make the contribution clearer by having a section that really addresses a direct comparison of advantages and limitations with respect to standard recurrent backpropagation. As someone who is not an expert in this specific proposed approach, it remained very unclear to me reading through the paper what precisely the benefits would be. The authors seems to be motivated by biological plausibility or scalability as the paper does not claim that backpropagation is ever surpassed in the experiments. In light of this, it is also disappointing that potential benefits in terms of scalability could not be validated empirically.

**Strengths And Weaknesses:**

Strengths:
- The authors provide a solid theoretical foundation for their approach in Theorems 1 and 2.
- I really appreciate sections 2.3 and 2.4, which relate back to the implications for the original learning problem and alternative interpretations of the least-control principle.
- It is nice to see application in both supervised learning and meta-learning settings, but it would be preferable to consider a larger breadth of datasets and architectures to provide a stronger validation of the proposed meta-learning approach.

Weaknesses:
- Authors match but do not improve on backpropagation in terms of performance and the authors did not clearly highlight what the additional advantages of this framework would be presumably related to scalability and biological plausibility.
- The title should be refined to highlight the focus on biological plausibility and recurrent neural networks. I believe I bid on this paper thinking it would probably relate to multi-agent RL and/or game theory in RL based on the title. You probably want to give readers a better idea about exactly what topics this paper will focus on in your title.

While I think this paper is strong in a number of ways, I lean towards rejection at the moment because I feel that I am missing some basic understanding about the underlying motivation for exploring this direction. However, I look forward to getting answers to my questions from the authors and seeing what the other reviewers have to say.

---

> ### Author Response · Authors · 2022-08-02
> **Answer to reviewer PoYW (part I)**
>
> We thank the reviewer for the helpful comments. In the following, we address the main concerns and comments.
>
> **Motivation and comparison with recurrent backpropagation.** The main motivation behind our work is to provide a gradient-based theory for activity-dependent learning in the brain. First, theories for learning in the brain have specific locality constraints, as each synapse can only access local information in space and time for updating its connection strength. This creates the need for local learning rules, and preferably also single-phase learning rules to prevent the need for overhead control and memory storage of intermediate learning signals. Second, feedback learning signals presumably change neural activity, unlike standard backpropagation which relies on a side network for credit assignment and instructing synaptic change (of other neurons). Third, leveraging gradient information is crucial for enabling efficient learning in high-dimensional parameter spaces, such as neural networks, exemplified by the staggering successes of deep learning. We will change the title of our paper to “The least-control principle for local learning at equilibrium” to better reflect the motivation behind our work.
>
> The main focus of the field has been on designing local gradient-based learning rules for feedforward neural networks that require credit assignment in space, and apart from a few notable exceptions [24, 25, 30], much less progress has been made for recurrent neural networks and other (meta) learning systems that require both credit assignment through space and time. Our work contributes to closing this gap for the special case of equilibrium systems.
>
> Although there could arise some potential benefits from the single-phase characteristics of the least-control principle, our goal is not to design a learning algorithm that is better performing or has favorable scalability properties compared to recurrent backpropagation. Instead, our aim is to design a single-phase, local learning strategy for equilibrium systems requiring both credit assignment in space and time. Compared to recurrent backpropagation, the least-control principle has as benefit that it is a single-phase algorithm reaching comparable performance, and results in local learning rules for common network architectures, making it suitable as a theory for learning in the brain. Other potential benefits such as improved stability of the network dynamics due to the added control remain to be explored in future work. The downsides of the least-control principle compared to recurrent backpropagation are that it only indirectly solves the original learning objective of Eq. 1 by minimizing the surrogate least-control objective of Eq. 3, which sometimes requires more training iterations, and that we observe that it has a higher computational cost to reach the controlled equilibrium on average. We quantified the difference in computational requirements, which we describe below.
>
> **Comparison of computational costs.** Encouraged by the reviewer’s comment, we include a detailed computational comparison between LCP-DI and RBP on the MNIST experiments for equilibrium RNNs. The table below shows the amount of iterations (phase length), the computation time for one single iteration (silicon time; obtained by using the NVIDIA/PyProf profiler offline), and the total training time for both algorithms. Note that RBP has two phases, whereas LCP only one. LCP and RBP require a similar total amount of iterations, however, RBP requires significantly fewer computations per iteration, as LCP needs to compute both the system states and control states every iteration, resulting in more matrix-vector products. Consequently, LCP requires roughly 6 times more compute compared to RBP on this line of experiments, when simulating on standard digital hardware. Hence, despite its desirable single-phase property, the added computational cost per iteration results in a significantly higher computational cost of LCP. We will discuss these results in the limitations paragraph of the discussion, and add the detailed results in the supplementary materials.
> |     | Avr. 1st phase length | 1st phase silicon time (ns) | Avr. 2nd phase length | 2nd phase silicon time (ns) | Total training time (sec) |
> |-----|-----------------------|-----------------------------|-----------------------|-----------------------------|---------------------------|
> | LCP | 139                   | 221634                      | -                     | -                           | 10515 +/- 411             |
> | RBP | 77                    | 28735                       | 72                    | 32479                       | 1683 +/- 360              |

---

> > ### Author Response · Authors · 2022-08-02
> > **Answer to reviewer PoTW (part II)**
> >
> > **Intuitive explanation of minimizing the amount of optimal control needed at equilibrium.** The least-control principle considers a learning setting where a controller pushes the system (e.g. a neural network) to a controlled equilibrium where the controlled state $\phi_*$ minimizes the loss, hence ‘solves’ the task at hand (c.f. Fig 1A). However, the system requires help from the controller to reach this solution state, so it is not yet capable of solving the original learning objective of Eq. 1 without help from the controller. To reduce the dependence of the system on the controller for solving the task, we aim to minimize the amount of *optimal control* needed to reach the solution state. To understand the requirement for an *optimal* control, consider that for many systems (e.g. the recurrent network of Eq. 11), the system state $\phi$ is of a higher dimension than the system output $y=h(\phi)$ on which the loss is defined. Hence, many different controls $\psi$ can lead to the same controlled system output $y$. In our theory, this degeneracy is broken by requiring the optimal control defined in Eq. 3 which is interpreted as the *minimal amount of control needed to reach the solution state for the current parameter setting $\theta$*. Now, by updating the parameters $\theta$ to minimize the amount of optimal control needed at equilibrium, we move the free equilibrium (without help from the controller) closer to the controlled equilibrium that is a solution state, hence implicitly solving the original learning objective (c.f. Fig 1B).
> >
> > **Clarification of the comparison with iMAML.** As the least-control principle is designed for systems at equilibrium, its application to meta-learning considers an inner loop that trains the network until convergence on the inner loss (c.f. Eq. 16). Hence, we benchmark it to other meta-learning algorithms considering the same equilibrium setup. Here, iMAML represents the recurrent backpropagation equivalent for meta-learning, as it computes the meta-gradient using implicit differentiation, and approximates the intractable inverse Hessian matrix vector product by relaxing a dynamical system similar to recurrent backpropagation. We did not compare to MAML, as it is not applicable to the considered equilibrium problem setting. We will clarify section 3.2 to better reflect these motivations.

---

> > > ### Comment · Reviewer_PoYW · 2022-08-08
> > > **Re: Answer to reviewer PoTW**
> > >
> > > After reading through the author’s rebuttal and the other reviews, I have decided to increase my score. I really appreciate the author’s explanations about their motivation, the intuition behind their approach, and their reasoning for comparing to iMAML rather than MAML.  I found these explanations to be quite clear and clarifying for me. Additionally, reading through the other reviews, I feel that because this paper does not align well with my technical background, I likely have underestimated the significance of this work in my initial review.
> > >
> > > I am still not sure that I fully buy the argument the authors are making that this approach can simultaneously be more biologically plausible while not providing empirical benefits in any setting. However, I can definitely accept that this setting is beyond current practical hardware considerations. I appreciate the compute results provided because they provide more clarity, but as the authors note it seems that it did more to highlight limitations of the approach rather than benefits.
> > >
> > > I think the modified title helps a bit in terms of aligning with the positioning of this paper, but it still does not do anything to disambiguate the topic of the work from multi-agent RL or game theory. It is not a huge deal I guess, I just think that it is the reason that I became a reviewer on this paper, which was probably inappropriate given my technical background. Could be nice to add some keyword like “biological” or “neural networks” etc.

---

### Author Response · Authors · 2022-08-02
**General overview of the new results and added clarifications (part I)**

We thank all reviewers for their insightful and sharp comments that helped us improve the paper significantly. We list here a summary of the new experiments and changes we have done and are planning to do, and then respond in detail to all the specific concerns in separate answer posts for each reviewer.

New experimental results:
* **Feedforward networks on MNIST.** To enable an easier comparison with the literature, we performed a new line of experiments on standard feedforward neural networks on MNIST. We tested fully-connected networks trained with LCP-LF, LCP-DI and LCP-DI (KP) and got comparable results to standard error backpropagation for all variants.
* **Feedforward convolutional networks on CIFAR10.** To test LCP in a more challenging setting, while avoiding spatial backpropagation within specific modules for biological plausibility, we trained a convolutional feedforward neural network on CIFAR10 with LCP-DI and LCP-DI(KP), of which the latter avoids weight transport. We achieved competitive performance compared to error backpropagation.
* **Improved equilibrium RNN results on MNIST.** We improved the hyperparameters for all methods for training equilibrium RNNs on MNIST, and now get test accuracies close to 98% for all methods, while LCP-DI and LCP-DI (KP) significantly improve upon the performance of LCP-LF (>0.15% difference), indicating that its more accurate optimal control improves the learning.
* **Energy-based dynamics.** We tested the energy-based dynamics of Eq.
10 on MNIST with both feedforward and equilibrium RNN architectures, to empirically support the theoretical equivalence of the fixed points with the inversion dynamics of Eq. 8. Indeed, we find that both dynamics lead to similar performance on MNIST. Interestingly, we found that the inversion dynamics of Eq. 8 allow for significantly bigger step sizes and hence fewer iterations for reaching the fixed point, compared to the energy-based dynamics of Eq. 10, which is an important result for the predictive coding field.
* **Robustness of performance to non-zero alpha.** To empirically test the theory when the assumptions are not perfectly validated, we tested on feedforward networks that the results of LCP on MNIST are robust to a wide range of values for the controller leak rate $\alpha$.
* **Robustness of the fixed points to finite iterations.** To empirically test the theory when the assumptions are not perfectly validated, we tested on feedforward networks that the fixed points and hence parameter updates remain close to the ideal fixed point when the iterations are stopped prematurely or when there is some noise in the fixed point estimation process, as predicted by Proposition S8.
* **Comparison of the computational requirements.** We compared the computation times required for RBP and LCP-DI. Despite its desirable single-phase property, the increased amount of computations per iteration results in an increased computational cost of LCP with a factor of ~6.

New theoretical results:
* **Non-convex losses in the output space.** We generalized Propositions 3 and 4 towards losses that are non-convex in the output space. In brief, we can exchange the guarantee on a global minimum of the original learning problem when using convex losses for the guarantee on a local minimum when using non-convex losses.
* **Non-zero values for alpha.** We generalized Propositions 3 and 4 to non-zero values for alpha, still guaranteeing that we reach a minimum of the original learning objective for finite values of alpha, under the assumptions of these propositions and of Proposition S9.

---

> ### Author Response · Authors · 2022-08-02
> **General overview of the new results and added clarifications (Part II)**
>
> Additional clarifications and rewriting:
> * **Comparison with strong-DFC.** We will include a detailed comparison between Strong-DFC (Meulemans et al. 2022) and the least-control principle in the supplementary materials, and summarize the main points within relevant sections in the main paper.
> * **Additional details on the CIFAR10 experiment.** We will update supplementary sections S6.4.3 and S6.5.2 with additional details on the model choices for the CIFAR10 experiment with deep equilibrium models and the resulting updates. We will update the writing of Section 3.1.2 to reflect better the main motivation of this experiment, which is to test the validity of our new learning principle, and not necessarily the biological plausibility of the used model.
> * **Clarification on complete versus partial control.** We will clarify in the Discussion the distinction between complete and partial control, i.e. whether all the state variables are controlled or not, its link with the energy-based dynamics of Section 2.4, and the exciting future work direction to unify contrastive learning in energy-based models with the least-control principle.
> * **A detailed limitations discussion.** We will include a more detailed discussion on the limitations of our work, covering the results of comparing the computational costs.
> * **Improve Section 2.1 and 2.2.** We will update Sections 2.1 and 2.2 to incorporate the helpful suggestions of all reviewers.
> * **Improve the experimental section.** We will update Section 3 to better incorporate and motivate our various design choices to apply the least-control principle in practice.
> * **Clarification of the comparison with iMAML.** We will update Section 3.2 to clarify the reasoning behind our comparison with iMAML and not with standard MAML.

---

> > ### Comment · Reviewer_GGtZ · 2022-08-08
> > **Feedback on authors answer (overview)**
> >
> > Dear authors,
> >
> > I have answered point by point each of your answer. Thank you, it is highly appreciated and the amount of work you provided for the rebuttal is honorable.
> >
> > I was not happy about the implicit backprop happening in the CIFAR-10 experiments, but since the authors honestly acknowledged it, intend to clarify it in the main and added extra experiments on the feedforward architectures alleviating this issues, I'm happy to re-consider my rating.
> >
> > Most importantly, my initial judgment of the paper was skewed by a misunderstanding of mine that each of the neurons of the dynamical system received direct feedback from the controller, for any implementation of LFC. @authors, could you please confirm this so that I can make my final decision?
> >
> > If it is the case that LFC does not assume direct feedback from the output layer onto *each* neuron of the dynamical system, I think this is indeed a paper that would tremendously benefit the community and open the field onto new exciting research questions. But please, confirm my thinking above first. Thanks!

---

> > > ### Author Response · Authors · 2022-08-09
> > > **Discussion (overview)**
> > >
> > > Dear reviewer,
> > >
> > > Many thanks for participating in the discussion and for your detailed and constructive comments that keep helping us improve the quality of our work. We answered your new questions and comments point by point in the comments below. You are correct that, although each neuron needs a control signal $\psi$, this control signal does not need to be a ‘direct control’ from the output feedback control $u$, but can instead be propagated through internal feedback connections of the network, e.g. in a layerwise fashion for a feedforward neural network, as is the case in the inversion and energy-based dynamics of Eq. 8 and 10.

---

> > > > ### Comment · Reviewer_GGtZ · 2022-08-09
> > > > **Decision update**
> > > >
> > > > Dear authors,
> > > >
> > > > Thank you a lot for the detailed answer and sorry for the misunderstanding. I have updated my score to strong accept.
> > > > I wish I have written this paper myself!

---

### Meta-Review · Area_Chair_XW6g · 2022-08-26

**Recommendation:** Accept
**Confidence:** Certain

**Metareview:**

As summarized by reviewer 5Kjg, this work proposes a theoretical framework for optimizing equilibrium neural networks by reformulating the original constrained optimization problem into a least-control problem. The least-control problem is further solved with a two-steps iterative procedure that first drives the network dynamics with a control signal towards minimizing fixed points verifying the initial problem constraints, then taking a gradient step to minimize the norm of the control terminal cost.

Motivations for such a principle are manifold: First, solutions are shown to coincide with the initial optimization problem under feasible conditions on the control that leave flexibility for its explicit construction. Second, credit assignment and activation dynamics are originally combined in a single dynamic that result in a spatially and temporally local explicit weight update rule. Third, this formulation can also be linked to models of free energy minimization, offering an original perspective on the learning objective operated by such models as a relaxation of the proposed least-control objective.

Several experiments were conducted to showcase the learnability of such a system for supervised image classification, implicit neural representation and meta-learning. The technique is shown to have comparative performance to the current leading technique for training equilibrium model recurrent backpropagation (RBP) and can encompass meta-learning with good results compared to gradient-based methods.

Almost all reviewers, including myself, consider this work to have potentially a high impact, linking control theory with neural network learning, backed by convincing experiments. They introduce 'the least control principle' paradigm for training neural networks with local learning rules, and such learning algorithms, which are biologically inspired to be modular, are intimately related to predictive coding and energy based models. I believe this work will definitely encourage much discussion in the research community, and open up many avenues of further investigation, and for that, I strongly recommend acceptance (as a spotlight or oral presentation, or equivalent format, if applicable for this year).


**Award:**

Yes

---

### Decision · Program_Chairs · 2022-09-14

Accept